# Bilateral Information-aware Test-time Adaptation for Vision-Language Models

**Jingwei Sun**[1,*,‡]   **Jianing Zhu**[1,*]   **Jiangchao Yao**[2]   **Gang Niu**[3]   **Masashi Sugiyama**[3,4]   **Bo Han**[1,3,†]

[1]TMLR Group, Department of Computer Science, Hong Kong Baptist University
[2]Cooperative Medianet Innovation Center, Shanghai Jiao Tong University
[3]RIKEN Center for Advanced Intelligence Project
[4]The University of Tokyo

## Abstract

Test-time adaptation (TTA) fine-tunes models using new data encountered during inference, which enables the vision-language models to handle test data with covariant shifts. Unlike training-time adaptation, TTA does not require a test-distributed validation set or consider the worst-case distribution within a given tolerance. However, previous methods primarily focused on adaption-objective design, while the data tend to be fully utilized or simply filtered through a fixed low-entropy selection criteria. In this paper, we analyze the weakness of previous selection criterion and find that only selecting fixed proportion of low-entropy samples fails to ensure optimal performance across various datasets and can lead the model to becoming over-confident in wrongly classified samples, showing unexpected overfitting to atypical features and compromising effective adaptation. To improve upon them, we propose *Bilateral Information-aware Test-Time Adaptation* (BITTA), which simultaneously leverages two distinct parts of the test inputs during adaptation. Specifically, a dynamic proportion of low-entropy samples are used to learn the core representation under covariant shifts, while high-entropy samples are adopted to unlearn atypical features. This dual approach prevents the model from undesired memorization and ensures extensive optimal performance. Comprehensive experiments validate the effectiveness in various datasets and model architectures. The code is publicly available at: https://github.com/tmlr-group/BITTA.

## 1 Introduction

Vision-language models (VLMs), such as CLIP (Contrastive Language Image Pre-training) (Radford et al., 2021), have gained widespread attention in recent years as foundation models. Due to the strong prior knowledge acquired through training on massive vision and language pairs, VLMs demonstrated outstanding zero-shot generalization performance in downstream tasks (Zhang et al., 2024b; Liang et al., 2024b; Cui et al., 2024). However, in real-world applications, the inevitable distribution shift between the source and target domains (Liang et al., 2024a; Wang & Deng, 2018), caused by natural variations or data corruption (Hendrycks & Dietterich, 2019) (e.g., weather changes and digital disturbances), often leads to significant performance degradation. This poses great challenges to VLMs, especially in safety-critical fields, such as autonomous driving (Zablocki et al., 2022) and medical diagnosis (Guan & Liu, 2021). To handle this problem, researchers have proposed *Test-Time Adaptation* (TTA), which leverages the new data encountered during inference in an unsupervised way (Shu et al., 2022; 2023; Karmanov et al., 2024; Ma et al., 2024) to adapt to shifted domains.

Despite the promising results achieved by previous research, we observe that most existing TTA algorithms focus on improving optimization techniques while overlooking the adaptation data (Shu et al., 2022; Sui et al., 2024; Osowiechi et al., 2024). In test time, as the ground-truth labels are unavailable, the models are fine-tuned with an unsupervised loss on those unlabeled data to adapt to shifted distributions. Generally, as shown in the left part of Figure 1, all the input data will be

---

[*]Equal Contribution, [‡]Trainee at RIKEN AIP.
[†]Correspondence to Bo Han (bhanml@comp.hkbu.edu.hk).

Figure 1: **Comparison of previous low-entropy selection criteria and our bilateral information-aware variant.** We pay attention to bilateral information for simultaneously selecting low entropy samples for learning and high entropy samples for unlearning.

considered as the adaptation target, then a low entropy selection criterion will be used to select a fixed proportion of confident samples to minimize the learning objectives (Shu et al., 2022; Niu et al., 2022; 2023; Feng et al., 2023). The intuition is based on an assumption that low-entropy samples contain typical features, from which the model can learn representative information about the adaptation target. However, it naturally raises the two questions: *Does the fixed low-entropy selection beneficially affect adaptation on all input data? Are those high-entropy samples all useless?*

In this paper, we start by highlighting two concerning phenomenons: learning on low-entropy samples can exacerbate the errors on certain data, and the fixed selection ratio cannot adapt to all data. As shown in Figure 2, we analyze the predictions of misclassified samples under TTA with CLIP model after applying previous selection criteria (Shu et al., 2022). The results indicate that while some initially misclassified samples are correctly classified during TTA, with lower output entropy (*i.e.,* higher confidence), a significant number of misclassified samples are still incorrectly classified. More critically, these samples also exhibit lower output entropy along the process, suggesting that the model memorizes certain atypical features during optimization on the confident samples, which compromise the adaptation. Fortunately, encompassing various atypical features, the high-entropy samples which the current model is unconfident can exactly serve as a potential candidate for appropriate regularization during adaptation. Additionally, the fixed selection ratio also restricts the model's adaptation performance. The ablation experiment in Figure 9 confirms that data with different distributions have different optimal selection ratios. Through a closer look, we found that the entropy values corresponding to different optimal selection ratios exhibit high consistency, which provides an important basis for generating standardized signals to indicate the optimal selection ratio.

Based on our findings in the analysis, we propose **B**ilateral **I**nformation-aware and **T**est-**T**ime **A**daptation (BITTA) to adapt VLMs during test time. In general, it utilizes data distribution information to calculate a dynamic proportion of low-entropy samples to learn representative information and uses high-entropy samples to unlearn atypical ones (as shown in the right part of Figure 1). On the one hand, considering the scattered feature distributions in corrupted data, we utilize low-entropy samples for model learning by minimizing entropy to reduce the uncertainty of output results, enhancing the consistency between vision and text features, and improving the separability between classes. On the other hand, considering the memorization on atypical features in low-entropy samples, we conduct unlearning on high-entropy samples to prevent misleading adaptation with atypical features. To demonstrate the effectiveness of our method, we conduct extensive experiments on three commonly used corrupted datasets, including CIFAR-10-C (Krizhevsky et al., 2009), CIFAR-100-C (Krizhevsky et al., 2009), and ImageNet-C (Deng et al., 2009), to improve VLMs' robustness to common image corruptions in safety-critical fields. We also provide further discussions on more cross-domain datasets and various ablation to justify the method's rationality. Our main contributions are as follows:

- Conceptually, we reveal two critical phenomenons in TTA with VLMs from new perspective, i.e., the fixed previous low-entropy selection criteria causes unexpected overfitting on atypical features, and cannot adapt to different data distribution. (in Section 3.1)

- Empirically, we recognize high-entropy samples can function as auxiliary set to effectively offset the atypical features memorized during the adaptation process and the entropy values can serve as standardized signals to indicate optimal selection ratio. (in Section 3.2)

- Technically, we propose BITTA, which innovatively selects bilateral information with dynamic ratio and simultaneously implement learning on low-entropy samples to fit core representations and unlearning on high-entropy samples to avoid overfitting. (in Section 3.3)

- Experimentally, we conduct extensive explorations to verify the effectiveness of BITTA under different scenarios, including the significant improvement on various datasets, compatibility with diverse methods and model architectures, etc. (in Section 4)

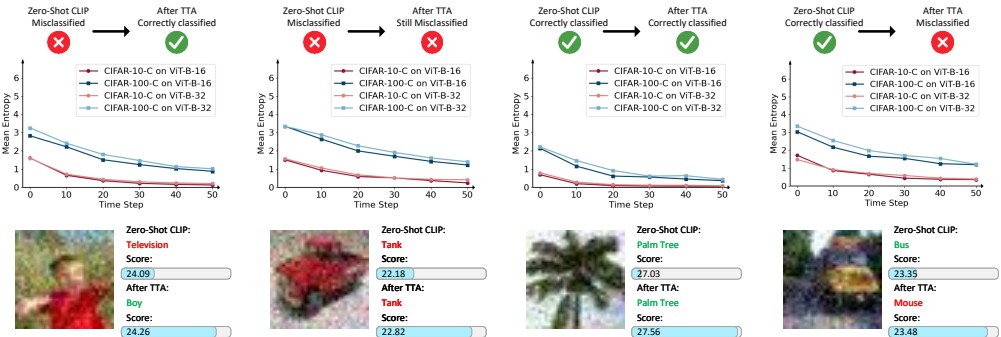

Figure 2: **Entropy dynamics on different data during adaptation.** The top line shows the entropy values when optimizing using low-entropy selection, while the bottom line shows the changes in the highest similarity score between samples and labels. It demonstrates that only learning on confident samples can induce unexpected overfitting (as decreasing entropy values) on misclassified samples.

## 2 PRELIMINARIES

In this section, we review the preliminaries of zero-shot prediction with CLIP and some commonly used techniques for test-time adaptation. In Appendix B, we provide discussions on related work.

**Zero-Shot Prediction.** The CLIP model (Radford et al., 2021) utilizes two pre-trained encoders to map images and textual descriptions into a shared feature space $\mathbb{R}^d$. For a $K$-class classification task, test images $x$ are sequentially fed into CLIP's visual encoder $E_v$ to generate the image feature $f_v$, and textual representations $\{f_{t_c}\}_{c=1}^K$ for each category are generated through the text encoder $E_t$. CLIP can perform zero-shot predictions by computing the similarity between the extracted image feature and textual representation as follows, $p(y = c|x) = \frac{\exp(\text{sim}(f_v, f_{t_c})/\tau)}{\sum_j \exp(\text{sim}(f_v, f_{t_j})/\tau)}$, where $c$ is a specific category. The pairwise similarities $\text{sim}(\cdot, \cdot)$ are calculated using cosine similarity, and $\tau$ represents the temperature parameter in the softmax function.

**Test-Time Adaptation (TTA).** TTA allows models to be fine-tuned based on new data encountered during inference, which enable the generalization of vision-language models on data with covariant shifts (e.g., common corruptions (Hendrycks & Dietterich, 2019)). TTA is generally realized by conducting entropy minimization (Niu et al., 2022; 2023; Wang et al., 2020) on low-entropy predictions to filter out the noise from test data:

$$\min -\frac{1}{\rho M} \sum_{i=1}^M \mathbb{I}[\mathcal{H}(x_i) < \tau] \sum_{c=1}^K p(y = c|x_i) \log p(y = c|x_i), \tag{1}$$

where $M$ is the batch size of test data and $K$ is the number of categories which is assumed to be known. $\mathcal{H}$ measures the output entropy and $\tau$ represents the threshold corresponding to the pre-defined selection ratio $\rho$. For cases where batch size is 1, a family of random augmentations $\mathcal{A}$ (Shu et al., 2022) is typical used to generated $M$ augmented views $\{\mathcal{A}_1(x), \mathcal{A}_2(x), ...\mathcal{A}_M(x)\}$.

## 3 METHOD

In this section, we present and discuss the key observations about unexpected overfitting and proportion adaptation problem that motivate our approach (in Section 3.1). Second, we discuss the key intuition behind the dynamic selection of bilateral information about input data during TTA (in Section 3.2). Lastly, we provide a detailed implementation of our proposed algorithm (in Section 3.3).

### 3.1 DRAWBACKS OF LOW-ENTROPY SAMPLES SELECTION.

**Overfitting on atypical feature.** We adopt the confidence selection strategy (Shu et al., 2022) to pick the top 10% of samples with the highest confidence and optimize them using the entropy minimization strategy (Shu et al., 2022; Maharana et al., 2025), and then analyze the entropy changes of four types of samples. As shown in Figure 2, the samples that remain correctly classified and those which were initially misclassified in the zero-shot CLIP but correctly classified after TTA exhibited lower entropy and higher confidence, indicating that the model has learned some typical features. This not only

(a) Atypical sample (b) Entropy change during TTA    (c) Entropy values under different selection ratios

Figure 3: **Motivation behind regularization and standardization.** (a) Visualizations of the misclassified low-entropy samples (top) and indistinguishable high-entropy samples (bottom), which both stem from noise information; (b) Entropy value change of the misclassified low-entropy samples when unlearning those high-entropy counterpart, which demonstrates the correlation between this two parts. (c) The entropy values corresponding to different selection ratios remain remarkably close.

helps correct previous errors but also boosts the model's confidence in subsequent classification. However, for the other two types of samples (i.e., those that remain misclassified or those that were originally correctly classified but misclassified after TTA), the model keeps wrong predictions with higher confidence. This suggests that the model has memorized some atypical features from selected samples (i.e., the noise information contained in the high confident incorrect predictions), which compromise the effective adaptation during inference.

**Incompatibility of fixed selection ratio.** The sample selection ratio is also a crucial and noteworthy question which previous works often overlook. Typically, a pre-set threshold is introduced to select a fixed proportion of low-entropy samples. To illustrate the drawbacks of this strategy, we conduct optimization using different selection ratios on multiple datasets and count the corresponding optimal selection ratio. As shown in Figure 9 and Table 8, a remarkable drawback of this strategy is that the pre-set ratio cannot enable optimal performance for different datasets. Even within the same dataset, the optimal selection ratios for different corruption types are inconsistent. This indicates that there are inherent differences in the effective information density of data with different distributions, and it is necessary to dynamically adjust the selection ratio to achieve the optimal performance.

## 3.2 REGULARIZATION AND STANDARDIZATION

The aforementioned overfitting problem means that some misclassified cases within low-entropy samples cause the model to unconsciously memorize some atypical features, leading to overly confident decisions when facing wrongly classified ones. Therefore, additional regularization should be considered. However, the atypical features seems to be entangled into the low-entropy samples during TTA which is hard to be filtered out regarding the sample-level selection. Fortunately, we also have a part of high-entropy samples, containing various atypical features which can be used to regularize the adaptation process. Moreover, for the incompatibility of fixed low-entropy selection ratio, we need to identify standardized characteristics which can reflect the optimal selection ratio.

**Regularization with atypical feature.** To verify whether it is feasible to utilize high-entropy samples for regularization, we visualize some misclassified samples from both high-entropy and low-entropy part in Figure 3(a). As can be seen, for the misclassified low-entropy samples, the model mistakenly identifies some noise information as key features of the wrong category. Similarly for the high-entropy part, the key features that could originally be used to distinguish categories are masked by massive indistinguishable noise. It can be seen that both the indiscriminability of high-entropy samples and the misclassification of some low-entropy samples stem from noise information. Therefore, the atypical feature from both parts is similar. More visualization of atypical samples are shown in Appendix H.3. To further validate the feasibility of using high-entropy samples to regularize the overfitting on atypical features, we conduct active unlearning on the high-entropy samples. By selecting the bottom 10% of samples with the highest entropy values, we increase the model's prediction entropy for these samples to further increase uncertainty. Meanwhile, we track the entropy changes of those misclassified low-entropy samples. As shown in Figure 3(b), the increase in entropy of high-entropy samples also triggers the increase in entropy of low-entropy misclassified samples. It demonstrates the close correlation between that two parts and the potential capability of using high-entropy samples as a representative auxiliary set. To further corroborate the rationality of utilizing high-entropy samples for unlearning, we provide the following theorem to demonstrate their indistinguishability.

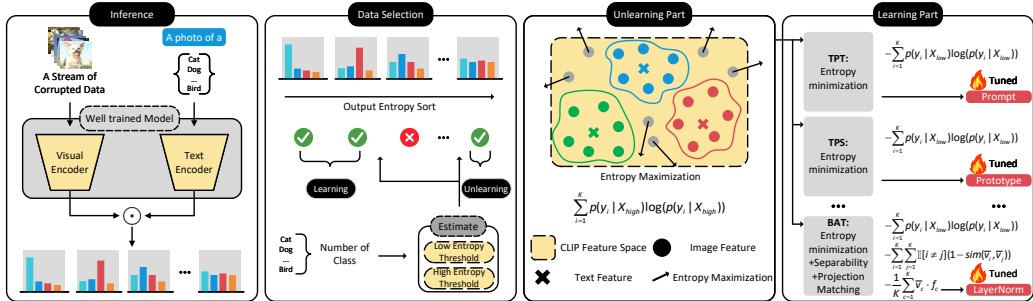

Figure 4: **Overview of BITTA:** *Left:* Given a stream of corrupted data, BITTA conduct inference using image features and text embeddings generated by the CLIP model. *Left-middle:* Estimating appropriate sample selection ratio according to the output entropy and number of categories. *Right-middle:* High-entropy samples will be used for unlearning to avoid overfitting. *Right:* Low-entropy samples will be used for learning core representations, with various methods available for this process.

**Theorem 3.1** *Assuming there exists one generation function $g(c, s)$, utilizing class and distribution information to generate image $x$. Now we have one image $x_{tgt}$ from target domain belongs to class $c_k$, we assume that it experiences a huge distribution shift $h$ from source domain $s_{src}$, i.e., $x_{tgt} = g(c_k, s_{src} + h)$, we then have that class $c_k$ can be strictly distinguished from all the other classes $c_{k'}$ when $h < \frac{D(c_k, c_{k'})}{2J_u}$, where $D(c_k, c_{k'})$ is the minimal distance between $c_k$ and $c_{k'}$ in the source domain, $J_u$ is an upper bound of the Jacobian spectrum norm of $g$.*

From this theorem, we can find that it is theoretically difficult to distinguish the samples that deviate significantly from the source domain and high-entropy samples that deviate severely from the source domain due to excessive noise exactly belong to this category of samples. Therefore, to prevent the model from falling into overfitting, it is wiser to avoid learning these unlearnable features.

**Standardization of selection ratio.** To identify common characteristics across different selection ratios, we analyzed the entropy values of the samples at these optimal selection ratios. As shown in Figure 3(c), within the same dataset, the samples exhibit similar entropy values across these different ratios. This leads us to hypothesize that the initial entropy value may reflect the optimal selection ratio. However, we also observe that the initial entropy values differ across datasets. By analyzing the initial data obtained during the adaptation process (i.e., the entropy values of the first batch), we noticed a certain regularity. Specifically, there exists a linear relationship between the initial entropy and the number of categories (which will be discussed in detail in Section 3.3). Based on this relationship, we are able to dynamically predict the selection ratio, yielding reasonable adjustments. To provide theoretical support for this prediction, we introduce the following theorem.

**Theorem 3.2** *Let $X_i$ denote a batch of data, $f$ represent the predictive function fitted on the training set data, and $\hat{Y}_i = f(X_i)$ denote the low-entropy sample selection ratio predicted by this function. Now, we have the prediction error $\{S_i = |Y_i - \hat{Y}_i|\}_{i=1-n}$ of the selection module calculated on calibration set $(X_1, Y_1)...(X_n, Y_n)$, measuring the difference between the predicted value $\hat{Y}_i$ and the ground-truth value $Y_i$, and the corresponding quantile $Q(1 - \alpha) = quantile_{(1-\alpha)(1+1/n)}(S_1, S_2, ..., S_n)$. Then, for test data $X_{n+1}$, its corresponding ground-truth $Y_{n+1}$ satisfies the marginal coverage guarantee $P(Y_{n+1} \in [\hat{Y}_{n+1} - Q(1-\alpha), \hat{Y}_{n+1} + Q(1-\alpha)]) \geq 1 - \alpha$.*

From this theorem, the ground-truth optimal low-entropy sample selection ratio has at least $1 - \alpha$ probability to fall within the aforementioned interval. Therefore, as long as the width of this interval remains at a low level when $\alpha$ takes small, the predicted result can be theoretically guaranteed to be strictly close to the true optimal value. Due to space constraints, the completed proof of Theorems 3.1 and 3.2, as well as the calculation for the interval width, can be found in Appendix C and E.

## 3.3 METHOD REALIZATION

Based on the above insights, we present the framework and learning objectives of our proposed BITTA. The overall algorithms of the method is summarized in Appendix D.

**Framework.** As shown in Figure 4, our framework consists of the following four core components: (1) given a stream of corrupted data, we adopt well-trained VLMs (e.g., CLIP), extracting image features and text embeddings to calculate cosine similarity; (2) then we estimate the required sample selection ratio during the process based on the distribution of data categories; (3) using high-entropy samples to unlearn atypical features contained in low-entropy samples; (4) simultaneously using low-entropy samples to learn core representations. Since our framework is designed via a data perspective, it is general and compatible with different specific adaptation learning parts (Shu et al., 2022; Sui et al., 2024; Maharana et al., 2025).

**Bilateral information-aware adaptation.** As aforementioned, the adaptation with low-entropy selection results in overconfidence when handling misclassified samples. Thus, we consider using the high-entropy samples to alleviate overfitting, for which the overall objective is to minimize $\mathcal{L}_{\text{learning}} + \lambda \mathcal{L}_{\text{unlearning}}$, where $\lambda > 0$ is a balanced weight, $\mathcal{L}_{\text{unlearning}}$ indicates the newly-introduced objective for high-entropy samples, which targets to prevent the model from overfitting due to memorizing excessive atypical features, $\mathcal{L}_{\text{learning}}$ follows the same setup as previous TTA algorithms, which is responsible for learning core representations. Concretely, the unlearning loss is defined as,

$$\mathcal{L}_{\text{unlearning}} = \sum_{i=1}^{K} p(y_i | X_{\text{high}}) \log(p(y_i | X_{\text{high}})). \tag{2}$$

where $X_{\text{high}}$ indicate the selected high-entropy samples among input data. With a reverse optimization direction, it is designed to increase the predictive entropy of high-entropy samples, thereby enhancing the model's uncertainty in these predictions with atypical features.

For the learning part in TTA with CLIP model, various traditional methods can be adopted. Here we adopt the advanced adaptation-objective proposed by BAT (Maharana et al., 2025) as an instance, which fine-tune the layernorm layers to adapt VLMs. It consists of the following three components:

$$\mathcal{L}_{\text{learning}} = - \sum_{i=1}^{K} p(y_i | X_{\text{low}}) \log(p(y_i | X_{\text{low}})) - \frac{1}{K} \sum_{c=1}^{K} \bar{v}_c \cdot f_c - \sum_{i=1}^{K} \sum_{j=1}^{K} \mathbb{I}[i \neq j](1 - \text{sim}(\bar{v}_i, \bar{v}_j)), \tag{3}$$

which is to reduce prediction uncertainty, enhance image-text alignment, and improve discrimination ability across different categories, respectively. $X_{\text{low}}$ is the selected low-entropy samples and $\bar{v}_c$ is computed by averaging those visual embeddings whose corresponding pseudo label is $c$.

**Dynamical estimation threshold.** As we analysed before, the sample selection ratio is also a crucial and noteworthy question which previous works often overlook. A pre-set fixed low-entropy sample selection ratio cannot enable optimal performance for different datasets. Considering the entropy values of the initial data corresponding to different optimal selection ratios are similar, we extract a subset of images from the training set of several different datasets, and find the following relationship between the category number $K$, prediction entropy $\mathcal{H}(x_i)$, and the corresponding entropy $\tau_l^n$ of the optimal selection ratio: $\frac{\tau_l^n}{Max(\mathcal{H}(x_i))} = -0.00038K + 0.83$.

Subsequently, we use $\tau_l^p = \frac{1}{M} \sum_{i=1}^{M} \mathbb{I}[\mathcal{H}(x_i) < \tau_l^n]$ to convert the entropy into the predicted optimal ratio, which yields reasonable adjustments. However, for high-entropy samples, we adapt a fixed selection ratio $\tau_h^p$ as we have observed that a fixed ratio of high-entropy samples is sufficient to achieve optimal performance, which indicates that, unlike the requirement for diverse low-entropy samples, a small amount of high-entropy samples can provide sufficient regularization signals. More detailed explanations of our dynamical estimation module can be found in Appendix E.

## 4 EXPERIMENT

In this section, we provide comprehensive verification. First, we introduce details of experimental setups (in Section 4.1). Second, we provide performance comparison and compatibility experiments (in Section 4.2). Third, we conduct extensive ablation studies to understand BITTA (in Section 4.3).

Table 1: **Comparison of TTA mean accuracy (%) across different corruption types, using two different CLIP models**. Δ highlighted the improvement in green over the top-performing baseline.

| | Method | Gaussian | Shot | Impulse | Defocus | Glass | Motion | Zoom | Snow | Frost | Fog | Brightness | Contrast | Elastic | Pixelate | JPEG | Mean |
|---|---|---|---|---|---|---|---|---|---|---|---|---|---|---|---|---|---|
| CIFAR-10-C | ViT-B-16 | 37.99 | 41.73 | 54.40 | 71.70 | 40.91 | 67.92 | 73.61 | 73.89 | 77.41 | 70.26 | 84.47 | 62.29 | 53.82 | 47.60 | 59.39 | 61.16 |
| | TPT | 39.91 | 44.91 | 58.76 | 72.25 | 43.53 | 70.03 | 74.77 | 75.48 | 78.51 | 72.71 | 85.06 | 70.84 | 57.23 | 52.29 | 61.27 | 63.84 |
| | TDA | 42.94 | 46.04 | 63.27 | 72.27 | 46.88 | 69.76 | 74.65 | 77.46 | 79.35 | 71.57 | 85.86 | 62.33 | 60.48 | 63.72 | 62.94 | 65.30 |
| | BCA | 31.51 | 34.53 | 54.17 | 66.50 | 35.64 | 64.93 | 70.59 | 73.90 | 76.27 | 68.45 | 84.02 | 55.42 | 51.52 | 55.80 | 55.95 | 58.61 |
| | DMN-ZS | 44.97 | 49.50 | 59.77 | 73.60 | 48.75 | 70.85 | 74.80 | 77.05 | 77.45 | 70.05 | 86.20 | 67.55 | 62.75 | 55.30 | 62.05 | 65.38 |
| | DPE | 34.78 | 38.66 | 57.81 | 73.14 | 44.30 | 68.59 | 75.89 | 74.26 | 78.89 | 71.17 | 86.90 | 64.43 | 57.49 | 52.37 | 58.54 | 62.48 |
| | BAT | 60.47 | 65.48 | 63.40 | 80.09 | 52.34 | 80.40 | 82.00 | 83.05 | 83.55 | 81.25 | 89.62 | 82.39 | 67.54 | 60.52 | 68.06 | 73.34 |
| | BAT+BITTA | **62.59** | **67.42** | **64.97** | **80.63** | **56.25** | **80.74** | **82.29** | **83.52** | **84.29** | **81.92** | **89.90** | **83.25** | **69.13** | **65.50** | **69.26** | **74.78** |
| | Δ | +2.12 | +1.94 | +1.57 | +0.54 | +3.91 | +0.34 | +0.29 | +0.47 | +0.74 | +0.67 | +0.28 | +0.86 | +1.59 | +1.78 | +1.20 | +1.44 |
| | ViT-B-32 | 35.58 | 40.07 | 43.16 | 69.98 | 41.50 | 64.51 | 70.19 | 70.80 | 72.34 | 66.66 | 81.38 | 64.51 | 59.66 | 48.16 | 56.58 | 59.01 |
| | TPT | 43.02 | 57.23 | 46.77 | 71.24 | 46.46 | 68.01 | 72.67 | 73.68 | 75.75 | 68.94 | 83.86 | 73.51 | 62.53 | 50.39 | 57.80 | 65.46 |
| | TDA | 47.40 | 51.38 | 52.15 | 72.68 | 54.42 | 69.19 | 75.08 | 76.77 | 78.01 | 72.27 | 84.70 | 63.33 | 66.55 | 56.54 | 60.55 | 65.40 |
| | BCA | 47.88 | 51.13 | 51.54 | 71.65 | 52.97 | 68.64 | 73.75 | 76.14 | 77.55 | 70.76 | 84.43 | 63.07 | 65.76 | 56.86 | 60.39 | 64.83 |
| | DMN-ZS | 45.90 | 48.60 | 43.85 | 70.10 | 48.80 | 68.00 | 72.20 | 72.10 | 73.05 | 67.95 | 81.40 | 67.70 | 63.10 | 53.20 | 59.05 | 62.33 |
| | DPE | 38.83 | 41.34 | 43.76 | 72.49 | 42.00 | 65.56 | 73.05 | 73.99 | 75.19 | 67.46 | 83.45 | 65.56 | 62.58 | 54.25 | 57.80 | 61.15 |
| | BAT | 49.40 | 54.79 | **52.40** | 76.16 | 53.08 | 75.06 | 76.29 | 76.81 | 78.32 | 74.83 | 86.33 | 78.83 | 66.58 | 53.06 | 60.39 | 67.49 |
| | BAT+BITTA | **52.34** | **57.61** | 52.28 | **76.82** | **55.99** | **75.79** | **76.83** | **77.64** | **78.68** | **76.79** | **86.50** | **80.41** | **67.44** | **58.45** | **61.69** | **69.02** |
| | Δ | +2.94 | +0.38 | -0.12 | +0.67 | +1.57 | +0.73 | +0.54 | +0.83 | +0.36 | +1.96 | +0.17 | +1.58 | +0.86 | +1.59 | +1.14 | +1.53 |
| CIFAR-100-C | ViT-B-16 | 19.57 | 21.39 | 25.26 | 42.46 | 20.08 | 43.19 | 47.98 | 48.44 | 49.70 | 41.69 | 57.00 | 34.37 | 29.21 | 23.93 | 32.47 | 35.79 |
| | TPT | 17.21 | 18.98 | 25.43 | 42.73 | 20.04 | 42.68 | 48.27 | 49.24 | 50.22 | 42.49 | 57.75 | 37.99 | 30.20 | 25.15 | 32.46 | 36.06 |
| | TDA | 23.36 | 24.34 | 32.60 | 42.31 | 20.17 | 42.04 | 46.93 | 48.28 | 48.28 | 41.44 | 57.00 | 31.95 | 32.95 | 33.39 | 33.41 | 37.24 |
| | BCA | 12.48 | 13.51 | 25.93 | 39.85 | 19.16 | 40.54 | 45.34 | 47.93 | 46.78 | 40.66 | 55.77 | 27.07 | 31.59 | 31.59 | 31.28 | 33.97 |
| | DMN-ZS | 19.75 | 21.95 | 29.80 | 42.50 | 18.30 | 43.40 | 48.30 | 49.10 | 51.05 | 43.85 | 56.70 | 35.80 | 31.80 | 26.45 | 31.25 | 36.67 |
| | DPE | 22.74 | 24.18 | 29.95 | 45.48 | 22.18 | 44.83 | 50.67 | 49.97 | 50.84 | 43.57 | 59.55 | 36.61 | 31.53 | 27.89 | 34.22 | 38.28 |
| | BAT | 21.40 | 25.19 | 29.87 | 50.40 | 25.70 | 48.21 | 54.85 | 52.13 | 50.69 | 48.22 | 63.51 | 43.34 | 34.78 | **31.75** | 37.17 | 41.15 |
| | BAT+BITTA | **26.71** | **28.67** | **35.19** | **50.71** | **26.06** | **48.95** | **55.24** | **52.56** | **51.44** | **48.66** | **63.67** | **46.63** | **35.32** | 30.83 | **37.79** | **42.56** |
| | Δ | +3.35 | +3.48 | +2.59 | +0.31 | +0.36 | +0.74 | +0.39 | +0.43 | +0.39 | +0.44 | +0.16 | +3.29 | +0.54 | -0.92 | 0.62 | +1.41 |
| | ViT-B-32 | 16.20 | 17.82 | 17.55 | 39.06 | 17.68 | 38.59 | 43.83 | 42.30 | 43.37 | 39.59 | 50.38 | 29.36 | 28.78 | 22.86 | 29.40 | 31.78 |
| | TPT | 14.76 | 16.37 | 15.56 | 37.55 | 18.62 | 37.56 | 42.82 | 43.01 | 44.16 | 39.35 | 51.36 | 32.17 | 29.55 | 21.73 | 28.69 | 31.55 |
| | TDA | 19.39 | 21.45 | 21.86 | 42.06 | 22.94 | 40.78 | 45.33 | 45.74 | 45.78 | 40.38 | 52.33 | 29.71 | 33.08 | 26.40 | 31.85 | 34.60 |
| | BCA | 14.39 | 15.96 | 15.88 | 40.91 | 22.46 | 39.32 | 44.55 | 44.27 | 44.47 | 39.20 | 50.05 | 28.22 | 34.39 | 26.98 | 30.39 | 32.76 |
| | DMN-ZS | 16.80 | 19.40 | 18.70 | 40.50 | 21.05 | 39.40 | 43.90 | 43.55 | 44.30 | 39.10 | 52.25 | 29.20 | 31.20 | 23.95 | 28.85 | 32.81 |
| | DPE | 18.33 | 20.02 | 19.98 | 42.13 | 19.70 | 41.70 | 47.24 | 45.27 | 46.19 | 41.09 | 53.61 | 32.42 | 31.72 | 27.31 | 31.97 | 34.58 |
| | BAT | 14.76 | 21.49 | 21.37 | 46.33 | 22.91 | 44.12 | 50.16 | 47.15 | 46.19 | 44.24 | 58.54 | 34.74 | 34.44 | 24.84 | 33.06 | 36.29 |
| | BAT+BITTA | **21.50** | **25.91** | **23.42** | **47.15** | **23.81** | **44.80** | **50.76** | **47.92** | **47.12** | **44.96** | **59.23** | **40.50** | **35.03** | **27.57** | **33.81** | **38.23** |
| | Δ | +2.11 | +4.42 | +1.56 | +0.82 | +0.87 | +0.68 | +0.60 | +0.77 | +0.93 | +0.72 | +0.69 | +5.76 | +0.59 | +0.26 | +0.75 | +1.94 |
| ImageNet-C | ViT-B-16 | 13.00 | 14.42 | 13.68 | 24.10 | 15.80 | 25.00 | 26.60 | 32.58 | 30.70 | 36.98 | 55.00 | 17.58 | 13.60 | 33.04 | 33.48 | 25.70 |
| | TPT | 8.78 | 9.42 | 9.64 | 24.48 | 16.20 | 25.12 | 23.98 | 33.78 | 32.30 | 37.78 | 55.56 | 19.10 | 14.26 | 35.88 | 34.84 | 25.41 |
| | TDA | 9.60 | 11.54 | 11.20 | 24.78 | 14.60 | 23.94 | 25.04 | 35.14 | 32.78 | 38.68 | 56.50 | 16.28 | 14.20 | 38.78 | 33.84 | 25.79 |
| | BCA | 6.74 | 7.74 | 7.18 | 26.00 | 14.96 | 24.90 | 25.88 | 33.86 | 32.94 | 37.92 | 56.14 | 17.32 | 14.34 | 39.60 | 35.80 | 25.42 |
| | DMN-ZS | 11.80 | 12.24 | 12.20 | 23.20 | 14.96 | 24.04 | 23.10 | 31.50 | 29.66 | 36.54 | 53.06 | 16.94 | 12.10 | 31.56 | 33.10 | 24.40 |
| | DPE | 10.26 | 13.46 | 17.28 | 26.36 | 17.96 | 27.46 | 24.86 | 35.20 | **33.24** | 39.60 | **56.52** | 20.16 | 15.96 | 36.78 | 35.40 | 27.37 |
| | BAT | 20.18 | 23.48 | 19.92 | 27.16 | 21.02 | 31.24 | 28.66 | 36.26 | 31.48 | 40.98 | 56.36 | 26.22 | **23.90** | 39.90 | 39.22 | 31.06 |
| | BAT+BITTA | **22.46** | **24.10** | **22.44** | **28.00** | **23.94** | **31.88** | **29.30** | **36.92** | 32.22 | **41.86** | 55.88 | **26.42** | 22.68 | **40.38** | **39.68** | **31.88** |
| | Δ | +2.28 | +0.62 | +2.52 | +0.84 | +2.92 | +0.64 | +0.64 | +0.66 | -1.02 | +0.88 | -0.64 | +0.20 | -1.22 | +0.48 | +0.46 | +0.82 |
| | ViT-B-32 | 14.30 | 14.56 | 14.54 | 23.90 | 12.00 | 21.96 | 19.96 | 25.52 | 26.36 | 29.22 | 50.80 | 16.66 | 19.12 | 31.70 | 31.22 | 23.45 |
| | TPT | 13.46 | 13.54 | 14.20 | 23.76 | 12.10 | 20.94 | 20.56 | 26.00 | 26.84 | 30.58 | 50.96 | 18.62 | 19.64 | 33.36 | 32.56 | 23.81 |
| | TDA | 11.92 | 13.12 | 12.90 | 24.40 | 11.86 | 22.94 | 22.34 | 28.42 | 25.82 | 32.58 | 51.12 | 16.74 | 20.28 | 36.12 | 33.24 | 24.25 |
| | BCA | 10.50 | 11.42 | 11.80 | 25.44 | 11.44 | 22.76 | 23.26 | 27.70 | 26.00 | 31.24 | 51.26 | 17.60 | 20.16 | 36.44 | 33.74 | 24.05 |
| | DMN-ZS | 14.00 | 14.14 | 13.36 | 26.04 | 12.50 | 23.80 | 23.36 | 28.04 | 26.26 | 32.44 | 49.76 | 17.40 | 19.30 | 33.84 | 29.96 | 24.28 |
| | DPE | 17.28 | 16.54 | 17.48 | 25.10 | 13.52 | 22.44 | 22.44 | 26.64 | 26.24 | 32.44 | **51.70** | 18.66 | 22.38 | 34.06 | 33.42 | 25.69 |
| | BAT | 19.08 | 19.70 | 19.30 | 24.86 | 19.38 | 28.68 | 25.42 | 28.56 | 26.16 | 35.46 | 50.08 | 17.10 | **27.04** | 36.70 | 36.14 | 27.58 |
| | BAT+BITTA | **22.48** | **22.34** | **22.20** | **27.22** | **20.22** | **29.26** | **25.80** | **28.82** | **27.46** | **35.94** | 50.00 | **23.96** | 26.96 | **36.78** | **36.30** | **29.05** |
| | Δ | +3.40 | +2.64 | +2.90 | +1.18 | +0.84 | +0.58 | +0.38 | +0.26 | +0.62 | +0.48 | -1.70 | +5.30 | -0.08 | +0.08 | +0.16 | +1.47 |

Figure 5: **t-SNE plots of image features from BITTA and zero-shot ViT-B-16.** The results indicate that our method produces more discriminative features that are more tightly clustered.

## 4.1 EXPERIMENTAL SETUPS

**Baselines.** For performance comparison, we adopt some commonly used method, including TPT (Shu et al., 2022), TDA (Karmanov et al., 2024), BCA (Zhou et al., 2025), DMN-ZS (Zhang et al., 2024d), DPE (Zhang et al., 2024a) and BAT (Maharana et al., 2025). For compatibility experiments, we adopt TPT (Shu et al., 2022), DiffTPT (Feng et al., 2023), CTPT (Yoon et al., 2024) and TPS (Sui et al., 2024). We provide more implementation details of each TTA method in Appendix A.

**Implementation details.** We test on CIFAR-10-C, CIFAR-100-C (Krizhevsky et al., 2009), and ImageNet-C (Deng et al., 2009) with a corruption level of 5. We assign template `a photo of a <cls>'` to the text encoder. The optimization for both vision and text encoder is based on AdamW (Loshchilov, 2017), with learning rates of 1e-3, 5e-4, and 5e-4 for CIFAR-10-C, CIFAR-100-C, and ImageNet-C, and batch sizes of 200, 200, and 64. More details can refer to the Appendix.

Table 2: **Comparison of BITTA combined with different TTA methods over CIFAR-10-C using ViT-B-16**. $\Delta$ highlighted the improvement in green over the original method without BITTA.

| Method | Gaussian | Shot | Impulse | Defocus | Glass | Motion | Zoom | Snow | Frost | Fog | Brightness | Contrast | Elastic | Pixelate | JPEG | Mean |
|---|---|---|---|---|---|---|---|---|---|---|---|---|---|---|---|---|
| TPT | 39.91 | 44.91 | 58.76 | 72.25 | 43.53 | 70.03 | 74.77 | 75.48 | 78.51 | 72.71 | 85.06 | 70.84 | 57.23 | 52.29 | 61.27 | 63.84 |
| TPT + BITTA | 42.37 | 47.13 | 60.89 | 73.62 | 45.92 | 71.67 | 75.72 | 76.45 | 79.70 | 73.71 | 86.25 | 71.03 | 59.10 | 54.94 | 62.06 | 65.37 |
| $\Delta$ | +2.46 | +2.22 | +2.13 | +1.37 | +2.39 | +1.64 | +0.95 | +0.97 | +1.19 | +1.00 | +1.19 | +0.19 | +1.87 | +2.65 | +0.79 | +1.53 |
| DiffTPT | 39.25 | 45.65 | 59.40 | 71.80 | 43.75 | 71.10 | 75.50 | 75.95 | 78.85 | 73.80 | 84.90 | 72.55 | 57.20 | 51.05 | 62.85 | 64.24 |
| DiffTPT + BITTA | 41.91 | 48.50 | 62.40 | 73.00 | 46.35 | 73.05 | 76.75 | 76.85 | 80.10 | 74.90 | 85.95 | 71.90 | 60.15 | 54.85 | 63.75 | 66.03 |
| $\Delta$ | +2.66 | +2.85 | +3.00 | +1.20 | +2.60 | +1.95 | +1.25 | +0.90 | +1.25 | +1.10 | +1.05 | -0.65 | +2.95 | +3.80 | +0.90 | +1.79 |
| CTPT | 38.18 | 42.74 | 56.73 | 71.66 | 41.58 | 68.91 | 74.18 | 74.41 | 77.05 | 70.6 | 84.55 | 63.95 | 54.98 | 49.33 | 58.79 | 61.84 |
| CTPT + BITTA | 39.40 | 44.10 | 58.31 | 72.16 | 43.04 | 69.46 | 74.67 | 74.77 | 78.05 | 71.08 | 85.15 | 64.05 | 55.66 | 50.15 | 59.12 | 62.61 |
| $\Delta$ | +1.22 | +1.36 | +1.58 | +0.50 | +1.46 | +0.55 | +0.49 | +0.36 | +1.00 | +0.48 | +0.60 | +0.10 | +0.68 | +0.82 | +0.33 | +0.77 |
| TPS | 40.11 | 44.48 | 59.73 | 71.72 | 43.05 | 70.25 | 74.32 | 74.69 | 77.67 | 73.36 | 83.78 | 77.30 | 58.35 | 52.85 | 62.59 | 64.28 |
| TPS + BITTA | 42.62 | 46.92 | 61.32 | 72.27 | 43.87 | 71.32 | 74.82 | 74.97 | 77.82 | 72.87 | 84.39 | 77.20 | 59.82 | 57.26 | 61.06 | 65.24 |
| $\Delta$ | +2.51 | +2.44 | +1.59 | +0.55 | +0.82 | +1.07 | +0.50 | +0.28 | +0.15 | -0.49 | +0.61 | -0.10 | +1.47 | +4.41 | -1.53 | +0.96 |

Table 3: **The classification accuracy of ViT-B-16 on cross-domain datasets.** The experimental results show that the BITTA also demonstrates good compatibility on these different datasets.

| Method | Flower102 | DTD | Pets | Cars | UCF101 | CalTech101 | Food101 | SUN397 | Aircraft | EuroSAT | Mean Mean |
|---|---|---|---|---|---|---|---|---|---|---|---|
| TPT | 68.98 | 47.75 | 87.79 | 66.87 | 68.04 | 94.16 | 84.67 | 65.50 | 24.78 | 42.44 | 65.10 |
| TPT+BITTA | 70.56 | 48.99 | 87.33 | 67.13 | 68.47 | 94.85 | 86.50 | 66.47 | 26.08 | 44.44 | **66.09** |
| TPS | 71.28 | 50.45 | 87.41 | 69.04 | 70.75 | 95.13 | 85.18 | 68.45 | 26.30 | 43.80 | 66.78 |
| TPS+BITTA | 72.34 | 51.68 | 88.69 | 69.13 | 71.61 | 95.47 | 86.44 | 72.82 | 25.67 | 46.68 | **68.05** |

Table 4: **Ablation study on prompt templates on ViT-B-16/ViT-B-32.** Our method shows great adaptability to different templates.

| Prompt Template on ViT-B-16 | CIFAR-10-C | CIFAR-100-C | ImageNet-C |
|---|---|---|---|
| 'a photo of a <cls>' | 74.78 | 42.56 | 31.88 |
| 'a low contrast photo of a <cls>' | 74.85 | 42.65 | 31.97 |
| 'a blurry photo of a <cls>' | 75.55 | 42.34 | 31.83 |
| 'a noisy photo of a <cls>' | 75.37 | 42.22 | 31.64 |
| Prompt Template on ViT-B-32 | CIFAR-10-C | CIFAR-100-C | ImageNet-C |
| 'a photo of a <cls>' | 69.02 | 38.23 | 29.05 |
| 'a low contrast photo of a <cls>' | 69.08 | 37.71 | 29.03 |
| 'a blurry photo of a <cls>' | 69.11 | 37.15 | 28.46 |
| 'a noisy photo of a <cls>' | 69.27 | 37.80 | 28.57 |

Table 5: **Generality on diverse architectures on ImageNet-C.** Our method can boost zero-shot performance under different architectures.

| Various Baseline Methods on ResNet101 | Gaussian | Shot | Impulse | Defocus |
|---|---|---|---|---|
| MEMO | 31.84 | 33.71 | 33.22 | 30.52 |
| MEMO+BITTA | 32.63 | 34.82 | 34.07 | 31.70 |
| SAR | 33.45 | 33.33 | 34.52 | 32.38 |
| SAR+BITTA | 33.92 | 33.94 | 34.90 | 32.35 |
| Various Baseline Methods on ViTBase | Gaussian | Shot | Impulse | Defocus |
| MEMO | 39.58 | 37.13 | 39.42 | 31.85 |
| MEMO+BITTA | 40.69 | 37.74 | 40.83 | 32.97 |
| SAR | 52.83 | 52.24 | 53.49 | 53.68 |
| SAR+BITTA | 56.27 | 56.44 | 57.69 | 58.26 |

## 4.2 MAIN RESULTS

**Comparison with different baselines.** In Table 1, we present a performance comparison with several representative TTA methods to demonstrate the effectiveness of BITTA. The results show that our method consistently outperforms others across different visual backbones. In Figure 5, we present visualization results of t-SNE plots (Van der Maaten & Hinton, 2008) which demonstrates that our method produces more tightly clustered category features, resulting in clearer distinctions between different categories. In Appendix H, we present detailed t-SNE results on VIT-B-16/VIT-B-32.

**Compatible with different methods.** As the unlearning mechanism is a general design from the perspective of data utilization in TTA, BITTA can be easily integrated into previous TTA methods, leading to performance improvements. In Table 2, we compare the performance of several methods using the ViT-B-16 backbone on the CIFAR-10-C dataset when combined with BITTA, including TPT (Shu et al., 2022), CTPT (Yoon et al., 2024) and TPS (Sui et al., 2024). We fix their learning objective for low-entropy samples and introduce high-entropy sample selection module, implementing unlearning to avoid overfitting on low-entropy samples. The results show that BITTA consistently helps these methods achieve better performance. More results are provided in Appendix F.2.

**Generalization on different datasets.** Cross-domain datasets are also commonly used in this field. To fully verify the effectiveness of our proposed method, we further supplemented experimental results on more cross-domain datasets as shown in Table 4.1, which confirm that BITTA has great generalization ability. More results on ImageNet variants are provided in Appendix F.9.

## 4.3 ABLATION AND FURTHER ANALYSIS

**Effects of different prompt templates.** Consistent with the prompt templates widely used in previous works, we use 'a photo of a <cls>' in our prior experiments. To examine the impact of different prompt templates on the experimental results, we replaced the prompt with various alternatives and reported the results in Table 4. As seen, the effects produced by different prompts are similar, suggesting that BITTA can flexibly adapt to various prompts and users can flexibly select the prompt in practical applications. Comparison with baseline methods can be found in Appendix F.4.

**Generality on diverse architectures.** To fully demonstrate the effectiveness of BITTA in other model architectures, such as CNNs and ViT, we present the effects of combining BITTA with two commonly used baseline (*i.e.*, MEMO (Zhang et al., 2022) and SAR (Niu et al., 2023)) under different

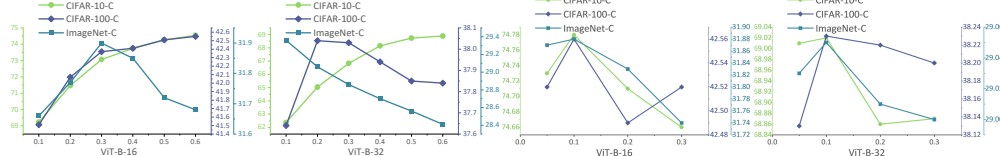

Figure 6: **Impact of varying update steps.** Continuous update lead to the performance degradation, while the mean accuracy still significantly exceed zero-shot ViT-B-16 and ViT-B-32.

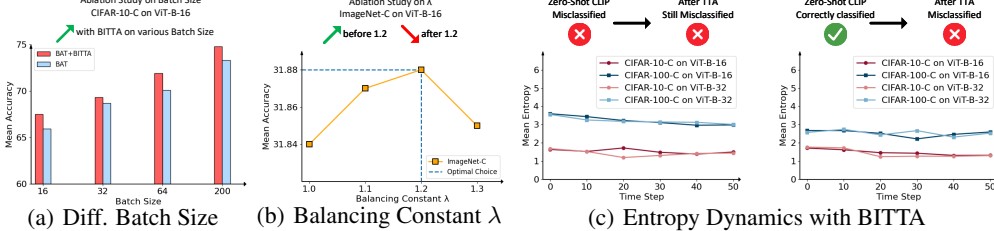

Figure 7: **Ablation on dynamical selection module.** *Left:* different low-entropy selection ratio. Optimal performance is achieved at different selection ratios for each dataset. *Right:* different high-entropy selection ratio. Fixed ratio is sufficient to achieve optimal performance.

Figure 8: **More ablation studies and further analysis.** (a) Performance on various batch sizes. Our method performs well under various batch sizes on the benchmark datasets; (b) Performance with varying balancing constant $\lambda$. Setting $\lambda$ to 1.2 is found to be optimal; (c) Entropy dynamics with BITTA. The results verify that the overfitting problems in misclassified samples can be alleviated.

architectures in Table 5. The experimental results clearly show that BITTA can significantly enhance the model performance regardless of the architecture. This outcome profoundly reveals that the overfitting problem widely exists in various architectures. More results can be find in Appendix F.8.

**Impact of varying update steps.** We perform multiple steps of parameter adaptation on a single batch. To evaluate the impact of different update steps on overall performance, we conduct ablation study by varying the number of update steps from 1 to 4. Figure 6 demonstrates that continuous updates in a single batch cause loss of prior knowledge, which in turn reduces the accuracy on subsequent data and ultimately results in performance degradation. Therefore, setting the number of update steps to 1 is a reasonable choice. More results on ImageNet-C can be found in Appendix F.3.

**Ablation on dynamical selection.** To demonstrate the necessity of this module, we show mean average accuracy across three datasets on the left side of Figure 7. We gradually increased the selection ratio of low-entropy samples from 0.1 to 0.6. The results show that for each dataset, the optimal performance is achieved under different selection ratios, while our estimation module can ensure excellent performance across all datasets. Meanwhile, on the right side, we also change the selection ratio of high-entropy samples from 0.1 to 0.3. It is found that setting the ratio to 0.1 is sufficient for all datasets to reach the optimal solution. Therefore, it is reasonable to design a dynamic selection ratio for low-entropy samples and a fixed selection ratio for high-entropy samples.

**Performance on different batch size.** In Figure 8(a), we show the average accuracy of our method using the ViT-B-16 backbone with smaller batch sizes. It can be observed that even with a low batch size, our method still significantly outperforms the baseline method, demonstrating excellent adaptability. More results and discussion can be found in Appendix F.7.

**Ablation study on $\lambda$.** We introduce $\lambda$ to balance the learning loss and the unlearning loss. As shown in Figure 8(b), we tested the performance of the model as $\lambda$ varied between 1.0 and 1.3. Empirically, setting $\lambda$ to 1.2 is found to be an appropriate choice. Although the results show the variance, it is worthy noting that the performance gap is smaller than 0.1% across different datasets, indicating a robustness of our methods regarding the $\lambda$. More results can be find in Appendix F.6.

Table 6: **The comparison with other regularization techniques.** The experimental results show that the unlearning module achieves better performance then other regularization techniques.

| Method | Gaussian | Shot | Impulse | Defocus | Glass | Motion | Zoom | Snow | Frost | Fog | Brightness | Contrast | Elastic | Pixelate | JPEG | Mean |
|---|---|---|---|---|---|---|---|---|---|---|---|---|---|---|---|---|
| BAT | 60.47 | 65.48 | 63.40 | 80.09 | 52.34 | 80.40 | 82.00 | 83.05 | 83.55 | 81.25 | 89.62 | 82.39 | 67.54 | 60.52 | 68.06 | 73.34 |
| +weight decay | 61.14 | 65.97 | 63.94 | 80.23 | 52.28 | 80.49 | 82.26 | 82.99 | 84.06 | 82.09 | 90.09 | 82.86 | 67.71 | 61.57 | 69.15 | 73.79 |
| +distribution dissolve | 60.98 | 66.50 | 64.88 | **80.69** | 55.75 | 80.58 | 82.28 | 83.02 | 84.18 | 81.89 | 89.59 | 83.14 | 68.83 | 64.21 | 69.06 | 74.37 |
| BAT+BITTA | **62.59** | **67.42** | **64.97** | 80.63 | **56.25** | **80.74** | **82.29** | **83.52** | **84.29** | **81.92** | **89.90** | **83.25** | **69.13** | **65.50** | **69.26** | **74.78** |

Table 7: **The comparison of expected calibration error (ECE ↓).** $\Delta$ highlighted the decline ratio in green over the original method without BITTA.

| | Method | Gaussian | Shot | Impulse | Defocus | Glass | Motion | Zoom | Snow | Frost | Fog | Brightness | Contrast | Elastic | Pixelate | JPEG | Mean |
|---|---|---|---|---|---|---|---|---|---|---|---|---|---|---|---|---|---|
| CIFAR-10-C | BAT (ViT-B-16) | 24.04 | 20.11 | 24.17 | 10.99 | 30.22 | 9.86 | 9.43 | 8.60 | 8.15 | 9.42 | 5.11 | 7.96 | 17.59 | 24.35 | 18.71 | 15.25 |
| | BAT+BITTA | 20.73 | 17.59 | 20.55 | 9.67 | 24.48 | 8.71 | 8.84 | 7.13 | 6.85 | 8.64 | 3.94 | 6.52 | 15.43 | 18.30 | 16.54 | 12.93 |
| | Δ | 13.77% | 12.53% | 14.98% | 12.01% | 18.99% | 11.66% | 6.26% | 17.09% | 15.95% | 8.28% | 22.89% | 18.09% | 12.28% | 24.85% | 11.59% | 15.21% |
| | BAT (ViT-B-32) | 50.04 | 34.31 | 31.14 | 12.93 | 28.51 | 13.83 | 12.78 | 11.58 | 10.48 | 12.66 | 6.16 | 9.50 | 19.03 | 25.45 | 23.41 | 20.12 |
| | BAT+BITTA | 23.47 | 21.51 | 26.90 | 11.20 | 23.90 | 11.58 | 10.57 | 9.53 | 8.66 | 10.52 | 4.79 | 7.66 | 16.09 | 21.02 | 19.38 | 15.12 |
| | Δ | 53.09% | 37.31% | 13.62% | 13.38% | 16.17% | 16.27% | 17.29% | 17.70% | 17.37% | 16.90% | 22.24% | 19.37% | 15.45% | 17.41% | 17.21% | 24.86% |
| CIFAR-100-C | BAT (ViT-B-16) | 50.81 | 47.82 | 32.37 | 13.31 | 25.07 | 13.78 | 12.06 | 12.97 | 15.35 | 17.15 | 8.85 | 16.93 | 21.31 | 20.30 | 17.43 | 23.03 |
| | BAT+BITTA | 21.18 | 20.53 | 13.59 | 11.36 | 21.56 | 11.54 | 9.75 | 10.13 | 10.93 | 13.33 | 6.79 | 12.15 | 18.34 | 17.43 | 15.49 | 14.27 |
| | Δ | 58.32% | 57.07% | 58.02% | 14.65% | 14.00% | 16.26% | 19.15% | 21.89% | 28.79% | 22.27% | 23.28% | 28.23% | 13.94% | 14.14% | 58.62% | 38.03% |
| | BAT (ViT-B-32) | 40.08 | 24.86 | 22.12 | 13.89 | 23.76 | 15.50 | 12.45 | 13.12 | 15.05 | 13.09 | 9.17 | 16.97 | 20.21 | 20.82 | 18.52 | 18.64 |
| | BAT+BITTA | 18.42 | 17.56 | 14.67 | 10.41 | 20.05 | 11.76 | 9.72 | 10.28 | 10.74 | 10.18 | 6.27 | 11.48 | 16.51 | 16.12 | 15.11 | 13.28 |
| | Δ | 54.04% | 29.36% | 33.68% | 25.05% | 15.61% | 24.13% | 21.93% | 21.65% | 28.64% | 22.23% | 31.62% | 32.35% | 18.31% | 22.57% | 18.41% | 28.73% |
| ImageNet-C | BAT (ViT-B-16) | 27.58 | 27.27 | 27.62 | 24.10 | 28.36 | 29.78 | 29.11 | 19.08 | 18.41 | 18.29 | 12.86 | 22.06 | 28.03 | 19.83 | 22.41 | 24.07 |
| | BAT+BITTA | 20.66 | 20.82 | 19.28 | 16.85 | 19.67 | 16.95 | 18.84 | 16.21 | 18.29 | 14.61 | 9.15 | 14.66 | 22.23 | 15.48 | 18.47 | 17.47 |
| | Δ | 25.09% | 23.65% | 30.20% | 30.08% | 30.64% | 43.08% | 35.28% | 15.04% | 25.62% | 20.64% | 28.85% | 33.54% | 20.69% | 21.94% | 17.58% | 27.39% |
| | BAT (ViT-B-32) | 26.99 | 25.19 | 26.00 | 21.62 | 23.81 | 19.96 | 24.63 | 20.22 | 24.02 | 19.56 | 15.33 | 27.79 | 23.05 | 18.08 | 18.49 | 22.32 |
| | BAT+BITTA | 17.94 | 17.50 | 17.41 | 15.42 | 17.17 | 14.76 | 19.57 | 15.24 | 18.37 | 14.48 | 10.16 | 16.41 | 18.26 | 14.48 | 14.97 | 16.14 |
| | Δ | 33.53% | 30.53% | 33.04% | 28.68% | 27.89% | 26.05% | 20.54% | 24.63% | 23.52% | 25.97% | 33.72% | 40.95% | 20.78% | 19.91% | 19.04% | 27.66% |

**Comparison with other Regularization Techniques** Apart from unlearning, we also try other regularization techniques such as weight decay (Fan et al., 2023; Chen et al., 2023) and distribution dissolve (Shen et al., 2024). As shown in Table 6, on CIFAR-10-C using ViT-B-16, although weight decay can achieve a certain effect, it is unable to prevent the model from learning atypical features and fails to realize more reasonable data selection, thus its overall performance is weaker than that of our method. For distribution dissolve, although it effectively enables the model to unlearn high-entropy samples, it differs fundamentally from our algorithm. Our method directly and precisely guides the model to confuse the classification of high-entropy images. In contrast, the distribution perturbation-based unlearning loss does not explicitly constrain the perturbation direction. This introduces the risk of the feature distribution incorrectly shifting from ambiguous regions (where the model fails to classify confidently) to discriminative regions (where the model can classify confidently). As a result, while this scheme achieves comparable performance to ours on certain corruptions, its average performance still remains lower. More implementation details can be find in Appendix F.10.

**Effects on mitigating overfitting.** To demonstrate BITTA can effectively mitigate overfitting, we present entropy changes during optimization process using BITTA in Figure 8(c). It can be observed that BITTA can prevent the model from being overconfident in misclassified samples (*i.e.*, the entropy of misclassified samples remains basically unchanged). This validates the superiority of BITTA. Additionally, Expected Calibration Error (ECE) is also an important metric that quantifies the calibration of a model's predicted probabilities (Yoon et al., 2024; Farina et al., 2024). We show the changes of ECE after applying BITTA in Table 7. As can be seen, BITTA also bring stable reduction on Expected Calibration Error, which also demonstrate BITTA's effects on mitigating overfitting.

**Additional Experiments and Discussion** We also provide further analysis from different views in the Appendix, including the introduction of related works(in Appendix B), time costs of conducting TTA (in Appendix F.1), effects of different learnable modules (in Appendix F.5), performance on more corruptions(in Appendix F.11), effects on non i.i.d. batches(in Appendix F.12), discussion on limitation and future work (in Appendix G.1), and visualization of prediction results (in Appendix H.5).

## 5 CONCLUSION

In this work, we propose BITTA, a new test-time adaptation framework focusing on the adaptation data. Through in-depth analysis, we identify that the previous selection criteria can induce unexpected overfitting, leading model becoming over-confident on wrongly classified samples. To better unleash the potential of adaptation, we introduce a bilateral information-aware mechanism which simultaneously utilizes two parts of the test inputs encountered during inference. With the novel unlearning part, we mitigate the memorization on atypical features and improve the model's adaptation performance. We hope this work can provide new insights for TTA researches.

ACKNOWLEDGMENTS

JWS, JNZ and BH were supported by NSFC General Program No. 62376235, RIKEN Collaborative Research Fund, and HKBU CSD Departmental Incentive Scheme. MS was supported by JST ASPIRE Grant Number JPMJAP25B1. JCY is supported by National Natural Science Foundation of China (No. 62306178) and STCSM (No. 22DZ2229005).

ETHICS STATEMENT

We hereby confirm that all datasets utilized in this work are publicly accessible and employed in strict adherence to their respective license agreements. This study involves no collection, use, or processing of human-related data (including personal information) or sensitive data (such as confidential, proprietary, or legally protected data), thereby eliminating any foreseeable risks pertaining to privacy infringement, security breaches, or fairness biases. The research is conducted exclusively for the purpose of advancing scientific knowledge in the field, with no financial, professional, or personal conflicts of interest declared by any author. Furthermore, all aspects of this work fully comply with the ethical guidelines outlined in the ICLR Code of Ethics.

REPRODUCIBILITY STATEMENT

To ensure the reproducibility of the experimental results, we summarize some critical factors that facilitate reproduction.

- **Datasets.** The datasets we used are all publicly accessible, which is introduced in Section 4.1. Following previous work, we test 5,000 images on ImageNet, and 10,000 images on CIFAR-10-C and CIFAR-100-C, with all test images remaining consistent with those used in prior work.
- **Assumption.** We set up our experiments to Test-Time Adaptation (TTA) scenario, where a well-pretrained CLIP model is available along with its full parameters. However, the ground-truth labels for the test images are not accessible, and there is no information transfer between different data streams. Developers can only optimize the model using the images available at each specific moment.
- **Environment.** All experiments are conducted on NVDIA Tesla V100-32GB GPUs with Python 3.9 and PyTorch 1.12.

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

APPENDIX

THE USE OF LARGE LANGUAGE MODELS

We use LLM as an assistant to help us polish our writing. We are grateful for the convenience that the development of LLMs has brought to research writing and call for the rational utilization of LLMs.

## A    DETAILS ABOUT CONSIDERED BASELINES AND METRICS

In this section, we provide details about the baseline and corresponding hyperparameters, as well as other related metrics that are considered in our work.

**TPT (Shu et al., 2022).**    For a given test image, 63 augmented versions of the image are generated based on random rotations, cropping, or flipping. Along with the original image, a total of 64 images are fed into the CLIP visual encoder. At initialization, the prompt template `'a photo of a <cls>'` is used to generate text embeddings. The confidence threshold is set to 10%, and the top 10% of samples with the lowest entropy values are used to optimize the prompt template. The learning rate is set to $5 \times 10^{-3}$.

**DiffTPT (Feng et al., 2023).**    Different from TPT, which utilizes conventional data augmentation techniques, e.g., random resized crops, to expand the test data, DiffTPT use the pre-trained stable diffusion to generate data with richer visual appearance variation and selecting generated data with higher prediction fidelity. Following this work, we utilize Stable Diffusion-V2 to generate the 63 augment views in the testing phase.

**TDA (Karmanov et al., 2024).**    Unlike conventional test-time adaptation (TTA) algorithms that adjust model parameters during the testing phase, TDA devises a training-free dynamic adapter, which constructs a lightweight key-value cache where a small number of pseudo-labels are taken as values and the features of the corresponding test samples as keys. Leveraging this key-value cache, TDA enables gradual adaptation to test data through progressive pseudo-label refinement without incurring any backpropagation operations, thus achieving efficient test-time adaptation in various real-world scenarios.

**BCA (Zhou et al., 2025).**    BCA analyzes the prediction process via Bayes' theorem and identify likelihood and prior as its core determinants. They claim that existing methods only adapt class embeddings to tune likelihood, while neglecting the critical role of prior. Therefore, BCA leverages the posterior of incoming test samples to dynamically refine the prior of each class embedding, which enables the model to better accommodate distribution shifts and boost prediction accuracy.

**DMN-ZS (Zhang et al., 2024d).**    DMN present a universal adaptation method. They design dual memory networks consisting of dynamic and static memory modules: the static memory module stores the knowledge of training data to support training-free few-shot adaptation, while the dynamic memory module real-timely retains historical test sample features during inference, thus mining extra data information that is not included in the training set. This unique design not only improves the model's performance in few-shot scenarios but also ensures the model's applicability when training data is unavailable. Moreover, the two memory modules adopt an identical flexible memory interaction mechanism, which can run in a training-free manner and be further optimized by integrating learnable projection layers. We utilize its zero-shot version (DMN-ZS) to conduct comparison experiment.

**DPE (Zhang et al., 2024a).**    Following CLIP, multiple context prompt templates are used for prompt ensembling. Specifically, For each class $c$, a total of $S$ text descriptions, denoted as $\mathcal{T}_c^{(i)}{}_{i=1}^{S}$. The prototypes of these descriptions in the embedding space are calculated by Eq. (4). To further improve the quality of these prototypes over time, DPE updated them online through a cumulative average with each individual sample in the test stream, as showing in Eq. (5), where $\mathbf{t} = [\mathbf{t_1}, \mathbf{t_2}, \cdots, \mathbf{t_C}]$,

$$\mathbf{t}_c = \frac{1}{S}\sum\nolimits_i \mathcal{E}_t(\mathcal{T}_c^{(i)}), \tag{4}$$

$$\mathbf{t}_c = \frac{(k-1)\mathbf{t} + \mathbf{t}^*}{||(k-1)\mathbf{t} + \mathbf{t}^*||}, \tag{5}$$

For visual prototypes, a priority queue strategy was designed to store the top-$\mathbf{M}$ image features for each class and symmetrically computed a set of that evolve over time. Using this priority queue, the class-specific visual prototype is obtained by Eq. (6), where $S_c$ donates the number of image features,

$$\mathbf{v}_c = \frac{1}{S_c} \sum_m f_c^m. \tag{6}$$

Based on these two types of prototypes, the final predictions are showing as follows, where $\mathcal{A} = \alpha exp(-\beta(1-x))$ is the affinity function, $t$ is the temperature parameter. In our experiments, the value of $\alpha$ is set to 6.0 and $\beta$ to 3.0 on CIFAR-10-C and CIFAR-100-C, while on ImageNet-C, $\alpha$ is set to 6.0 and $\beta$ to 5.0.

$$P(y = y_c|X) = \frac{\exp\left(\left(f_v^\top \mathbf{t}_c + \mathcal{A}(f_v^\top \mathbf{v}_c)\right)/t\right)}{\sum_{c'} \exp\left(\left(f_v^\top \mathbf{t}_{c'} + \mathcal{A}(f_v^\top \mathbf{v}_{c'})\right)/t\right)}. \tag{7}$$

In addition to the two types of prototypes $\mathbf{t}_c$ and $\mathbf{v}_c$, DPE also introduces learnable residual parameters $\hat{\mathbf{t}}_c$ and $\hat{\mathbf{v}}_c$, which are initialized to 0 and are gradually used to update the class prototypes:

$$\mathbf{t}_c \leftarrow \frac{\mathbf{t}_c + \hat{\mathbf{t}}_c}{\|\mathbf{t}_c + \hat{\mathbf{t}}_c\|}, \quad \mathbf{v}_c \leftarrow \frac{\mathbf{v}_c + \hat{\mathbf{v}}_c}{\|\mathbf{v}_c + \hat{\mathbf{v}}_c\|}. \tag{8}$$

The optimization of $\hat{\mathbf{t}}_c$ and $\hat{\mathbf{v}}_c$ is based on entropy minimization and alignment loss which utilizes the contrastive InfoNCE loss to bring prototypes from the same class closer together while pushing prototypes from different classes further apart. The optimization objective is shown as Eq. (12).

$$\mathcal{L}_{\text{aug}} = \mathcal{H}\left(\mathbb{P}_{\text{DPE}}\left(X_{\text{test}}\right)\right) = -\sum_{c=1}^{C} \mathbb{P}_{\text{DPE}}\left(y = y_c \mid X_{\text{test}}\right) \log \mathbb{P}_{\text{DPE}}\left(y = y_c \mid X_{\text{test}}\right), \tag{9}$$

$$\text{where } \mathbb{P}_{\text{DPE}}\left(X_{\text{test}}\right) = \frac{1}{\rho N} \sum_{n=1}^{N} \mathbb{1}\left[\mathcal{H}\left(\mathbb{P}_{\text{Proto}}\left(\mathcal{A}_n\left(X_{\text{test}}\right)\right) \le \tau\right] \mathbb{P}_{\text{Proto}}\left(\mathcal{A}_n\left(X_{\text{test}}\right)\right). \tag{10}$$

$$\mathcal{L}_{\text{align}} = \frac{1}{C} \sum_{c=1}^{C} \left(-\log \frac{\exp\left(\mathbf{t}_c^\top \mathbf{v}_c\right)}{\sum_{c'} \exp\left(\mathbf{t}_c^\top \mathbf{v}_{c'}\right)} - \log \frac{\exp\left(\mathbf{t}_c^\top \mathbf{v}_c\right)}{\sum_{c'} \exp\left(\mathbf{t}_{c'}^\top \mathbf{v}_c\right)}\right). \tag{11}$$

$$\mathbf{t}^*, \mathbf{v}^* = \text{argmin}(\mathcal{L}_{\text{aug}} + \lambda \mathcal{L}_{\text{align}}). \tag{12}$$

**BAT (Maharana et al., 2025).** Existing TTA methods have severe limitations in adapting CLIP due to their unimodal nature. To address these limitations, BAT-CLIP design a bimodal TTA method to improve CLIP's robustness to common image corruptions. Specifically, BAT optimizes the LayerNorm layers of the bimodal encoder by minimizing the output entropy while enhancing the projection matching between visual and textual features and improving the inter-class separability between class prototypes. The specific loss function are as follows:

$$\mathcal{L}_{ent} = -\sum_c p(l_c) log(p(l_c)), \tag{13}$$

$$\mathcal{L}_{pm} = \frac{1}{C} \sum \bar{v}_c \cdot \hat{z}_c, \tag{14}$$

$$\mathcal{L}_{sp} = \sum_{l \in C} \sum_{c \in C} \mathbb{I}[l \ne c](1 - cos(\bar{v}_c, \bar{v}_l)). \tag{15}$$

where $p(l_c)$ represents the likehood for class $c$, $\bar{v}_c$ and $\hat{z}_c$ represents visual and textual prototypes. The overall optimization objective is as follows, $\phi_v$ and $\phi_t$ represent the parameter of visual and text encoder's LayerNorm,

$$\underset{\phi_v, \phi_t}{\text{argmin}}\left(\mathcal{L}_{ent} - \mathcal{L}_{pm} - \mathcal{L}_{sp}\right). \tag{16}$$

**CTPT (Yoon et al., 2024).** Calibration is a crucial aspect for quantifying prediction uncertainty. However, previous works have been mainly developed to improve accuracy, overlooking the importance of calibration. CTPT discovered that well-calibrated prompts exhibit a more dispersed distribution of textual features. Therefore, it introduces Average Text Feature Dispersion (ATFD) to assess the calibration degree of the model. A higher value of ATFD indicates a more dispersed distribution of textual features, signifying better model calibration, while a lower value suggests poorer calibration. The formulation of ATFD is shown as follows:

$$\text{ATFD}\left(\mathbf{t}_{[p;y_1]}, \mathbf{t}_{[p;y_2]}, \ldots, \mathbf{t}_{[p;y_N]}\right) = \frac{1}{N}\sum_{i=1}^{N}\left\|\mathbf{t}_{\text{centroid}} - \mathbf{t}_{[p;y_i]}\right\|_2. \tag{17}$$

$\mathbf{t}_{\text{centroid}}$ is the centroid of the text features computed by:

$$\mathbf{t}_{\text{centroid}} = \frac{1}{N}\sum_{i=1}^{N}\mathbf{t}_{[p;y_i]}. \tag{18}$$

By maximize ATFD during TTA process, CTPT can contribute to better calibration.

**TPS (Sui et al., 2024).** Considering that TPT requires backpropagation through the text encoder to fine-tune the prompt, it leads to high memory and time consumption. To reduce these costs, TPS attempts to perform backpropagation only in the feature space. Specifically, TPS fine-tunes the prompt directly in the feature space by learning an offset vector $\mathbf{s}_c$ for each class prototype $\mathbf{p}_c$. The formulation is as follows:

$$\mathbf{p}'_c = \frac{\mathbf{p}_c + \mathbf{s}_c}{\|\mathbf{p}_c + \mathbf{s}_c\|_2}. \tag{19}$$

Similar to TPT, TPS selects low-entropy samples for training using confidence selection, and optimizes the offset vector $\mathbf{s}_c$ by minimizing the output entropy of the low-entropy samples. The loss function is as follows:

$$\mathcal{L} = -\sum_{c\in\mathcal{C}}\tilde{p}\left(c\mid\mathbf{x}_0, \mathbf{p}'_c\right)\log\tilde{p}\left(c\mid\mathbf{x}_0, \mathbf{p}'_c\right), \tag{20}$$

where $\tilde{p}\left(c\mid\mathbf{x}_0\right)$ is computed as follows:

$$\tilde{p}\left(c\mid\mathbf{x}_0\right) = \frac{1}{k}\sum_{i=1}^{k}p\left(c\mid\mathbf{x}'_i, \mathbf{p}'_c\right), \tag{21}$$

**Maximum Softmax Probability (MSP).** We use maximum softmax probability to discriminate which class the sample belongs to. The score is defined as follows,

$$S_{MSP}(x; f) = \max_{c}P(y = c|x; f) = \max softmax(f(x)), \tag{22}$$

$f$ represents the given well-trained model, and $c$ is one of the class $Y = \{1, \cdots, C\}$. The larger the softmax score, the higher the probability that the sample belongs to class $c$, reflecting the model's confidence in the sample.

# B  RELATED WORK

**Vision-Language Models (VLMs).**    VLMs have gained widespread attention for their ability to jointly process and understand both visual and textual information (Zhang et al., 2024b; Li et al., 2024; Hartsock & Rasool, 2024; Wang et al., 2024). CLIP (Radford et al., 2021) trains visual and language models simultaneously using contrastive learning, and its training on massive paired image-text datasets enables it to demonstrate strong zero-shot capabilities, making it suitable for a wide range of cross-modal tasks. ALIGNLi et al. (2021), on the other hand, leverages large-scale noisy data to broaden its applicability in cross-modal tasks and enhance its generalization ability. Subsequent models, such as DALL·E (Ramesh et al., 2021) and Flamingo (Alayrac et al., 2022), further enhanced task generalization by incorporating more complex architectures and training methods (Yuan et al., 2021; Singh et al., 2022; Li et al., 2022).

**Test-Time Adaptation.**    To mitigate the performance degradation caused by the distribution shift between the test and source domains, researchers have introduced test-time adaptation to ensure the reliability of models when deployed in out-of-distribution scenarios (Wang et al., 2020; Zhang et al., 2022; Gao et al., 2023; Chen et al.). These methods have been widely applied across various machine learning tasks, including image classification (Lin et al., 2022), image super-resolution (Deng et al., 2023), and object detection (Lin et al., 2025). In recent years, with the rise of VLMs, an increasing number of studies have focused on test-time adaptation for VLMs (Shu et al., 2023; Sreenivas & Biswas, 2024; Zhang et al., 2024c). TPT (Shu et al., 2022) introduces Test-Time Prompt Tuning, which learns domain-specific prompts for each sample by minimizing output entropy. TDA (Karmanov et al., 2024) constructs a lightweight visual key-value cache to store high-quality visual features and pseudo-labels during testing, which are then used as class prototypes in the subsequent classification process and effectively reduces the computational overhead associated with backpropagation. DPE (Zhang et al., 2024a) optimizes bimodal class prototypes for both text and visuals to better leverage multi-modal capabilities of VLMs. BAT (Maharana et al., 2025) not only adjusts the encoder for more effective feature extraction but also enhances the alignment between text and image features. In our work, rather than exploring better optimization objectives, we pay attention to another fundamental view, *i.e.* the data quality. Our BITTA strategy can effectively avoid the memorization of atypical features.

**Machine Unlearning.**    The core goal of machine unlearning is to adjust deep learning models to eliminate the influence of specific data points or classes. Its key application directions include defending against backdoor attacks (Graves et al., 2021), improving model fairness (Oesterling et al., 2024), and optimizing pre-training processes to enhance transfer learning performance (Sekhari et al., 2021). Various kinds of unlearning algorithm has been developed to achieve exact unlearning. (Neel et al., 2020; Sekhari et al., 2021) utilize methods based on differential privacy (DP). (Thudi et al., 2022) conducted unlearning research based on additional datasets, while (Zhang et al., 2023; Heng & Soh, 2023) proposed concept erasure strategy for diffusion models. Although existing unlearning methods cover a wide range, current mainstream unlearning techniques all require a target class as the object of unlearning. However, in this study, the object to be unlearned is features rather than classes, so there are significant difficulties in directly applying existing unlearning techniques. In the future, we aim to explore more sophisticated methods to achieve more precise unlearning of atypical features learned by the model.

# C  PROOF OF THEOREM

## C.1  PROOF OF THEOREM 3.1

First, we introduce the assumption about Theorem 3.1:

**Assumption C.1** *The generating function $g$ is a smooth invertible function with a smooth inverse everywhere.*

**Assumption C.2** *The invariant variable $c$ takes on values from a finite set: $C = \{c_k\}k \in [K]$.*

Now we prove Theorem 3.1 by the method of proof by contradiction. We assume that $x_{tgt}$ experiences a huge distribution shift, making it impossible to distinguish which class it belongs to, i.e., images of at least two differnt categories from source domain can generate $x_{tgt}$ after a certain shift $h$ or $h'$:

$$x_{tgt} = g(c_k, s_{src} + h) = g(c_{k'}, s_{src'} + h') \tag{23}$$

where

$$g(c_k, s_{src} + h) = g(c_k, s_{src}) + (\int_0^1 J_{g(c_k,.)}(s_{src} + t \cdot h)dt)h \tag{24}$$

$$g(c_{k'}, s_{src'} + h') = g(c_{k'}, s_{src'}) + (\int_0^1 J_{g(c_{k'},.)}(s_{src'} + t \cdot h')dt)h' \tag{25}$$

$J_g$ is the Jacobian matrix of $g$. Therefore,

$$g(c_k, s_{src}) - g(c_{k'}, s_{src'}) = (\int_0^1 J_{g(c_k,.)}(s_{src} + t \cdot h)dt)h - (\int_0^1 J_{g(c_{k'},.)}(s_{src'} + t \cdot h')dt)h' \tag{26}$$

We define $D(c_k, c_{k'})$ is the minimal distance between $c_k$ and $c_{k'}$, i.e., $\forall s_1, s_2 \in$ source domain $S_{src}$,

$$||g(c_k, s_1) - g(c_{k'}, s_{src'})|| \geq D(c_k, c_{k'}). \tag{27}$$

Therefore,

$$||(\int_0^1 J_{g(c_k,.)}(s_{src} + t \cdot h)dt)h - (\int_0^1 J_{g(c_{k'},.)}(s_{src'} + t \cdot h')dt)h'|| \geq D(c_k, c_{k'}) \tag{28}$$

$$\Rightarrow ||(\int_0^1 J_{g(c_k,.)}(s_{src} + t \cdot h)dt)h|| + ||(\int_0^1 J_{g(c_{k'},.)}(s_{src'} + t \cdot h')dt)h'|| \geq D(c_k, c_{k'}) \tag{29}$$

$$\Rightarrow J_u(||h|| + ||h'||) \geq D(c_k, c_{k'}) \tag{30}$$

where $J_u$ is an upper bound of the Jacobian spectrum norm. Therefore,

$$\max(||h||, ||h'||) \geq \frac{D(c_k, c_{k'})}{2J_u} \tag{31}$$

which violates the condition in Theorem 3.1 that $h < \frac{D(c_k, c_{k'})}{2J_u}$. Therefore, we can conclude that class $c_k$ can be strictly distinguished from all the other classes $c_{k'}$ when $h < \frac{D(c_k, c_{k'})}{2J_u}$.

From the above, it can be concluded that the classes are strictly separable only when the invariant variable is under a moderate amount of shift, *i.e.*, $h < \frac{D(c_k, c_{k'})}{2J_u}$. Otherwise, if the shift becomes excessive, the distributions of different classes may overlap and become difficult to distinguish. This means it is theoretically difficult to distinguish high-entropy samples due to the distance from the source domain, therefore unlearning these atypical feature helps to avoid overfitting.

## C.2 PROOF OF THEOREM 3.2

First, we introduce the assumption about Theorem 3.2:

**Assumption C.3** *Since the calibration data and test data are derived from the same dataset, we assume that $(X_i, Y_i)$ are exchangeable. The term "exchangeable" is defined as follows:*

**Definition C.1** *Let $Z_1, \ldots, Z_n \in \mathcal{Z}$ be random variables with a joint distribution. We say that the random vector $(Z_1, \ldots, Z_n)$ is* exchangeable *if, for every permutation $\sigma \in \mathcal{S}_n$, $(Z_1, \ldots, Z_n) \stackrel{d}{=} (Z_{\sigma(1)}, \ldots, Z_{\sigma(n)})$, where $\stackrel{d}{=}$ denotes equality in distribution, and $\mathcal{S}_n$ is the set of all permutations on $[n] := \{1, \ldots, n\}$.*

Then we introduce the following lemma:

**Lemma C.1** *Let $v_1, \ldots, v_{n+1} \in \mathbb{R}$. Then for any $t \in [0, 1]$, $v_{n+1} \leq quantile_t(v_1, \ldots, v_{n+1}) \iff v_{n+1} \leq quantile_{t(1+1/n)}(v_1, \ldots, v_n)$.*

Therefore,

$$Y_{n+1} \in [\hat{Y}_{n+1} - Q(1-\alpha), \hat{Y}_{n+1} + Q(1-\alpha)] \iff S_{n+1} \leq quantile_{1-\alpha}(S_1, \ldots, S_n, S_{n+1}) \tag{32}$$

From this point on, then, to establish the coverage guarantee, we only need to show that the event $S_{n+1} \leq quantile_{(1-\alpha)}(S_1...S_n, S_{n+1})$ holds with $>= 1 - \alpha$ probability.

Based on the exchangeability assumption, since the prediction error $S_1...S_n, S_{n+1}$ are obtained by applying the same function to $(X_i, Y_i)$, they are also exchangeable.

Therefore, we have

$$P(S_{n+1} \leq quantile_{1-\alpha}(S_1, \ldots, S_n, S_{n+1})) \geq \frac{(1-\alpha)(n+1)}{n+1} = (1-\alpha), \tag{33}$$

this completes the proof.

## C.3 PROOF OF LEMMA C.1

First, if $t > \frac{n}{n+1}$, the result holds trivially since $quantile_t(v_1, \ldots, v_{n+1}) = \max_{i \in [n+1]} v_i \geq v_{n+1}$, while $quantile_{t(1+1/n)}(v_1, \ldots, v_n) = +\infty$. From this point on, then, we will assume $t \leq \frac{n}{n+1}$ to avoid this trivial case.

Let $v_{(n;1)} \leq \cdots \leq v_{(n;n)}$ be the order statistics of $v_1, \ldots, v_n$, and let $v_{(n+1;1)} \leq \cdots \leq v_{(n+1;n+1)}$ be the order statistics of $v_1, \ldots, v_{n+1}$. Let $k = \lceil t(n+1) \rceil \in [n]$, we see that

$$quantile_t(v_1, \ldots, v_{n+1}) = v_{(n+1;k)} \tag{34}$$

and

$$quantile_{t(1+1/n)}(v_1, \ldots, v_n) = v_{(n;k)}, \tag{35}$$

by definition. So, we need to verify that $v_{n+1} \leq v_{(n+1;k)}$ holds if and only if $v_{n+1} \leq v_{(n;k)}$.

First, by definition of the order statistics, we must have $v_{(n+1;k)} \leq v_{(n;k)}$ (i.e., the $k$th smallest entry in the list cannot increase if we add a new value to the list). Therefore,

$$v_{n+1} \leq v_{(n+1;k)} \implies v_{n+1} \leq v_{(n;k)}. \tag{36}$$

To verify the converse, suppose that $v_{n+1} > v_{(n+1;k)}$. In this case, we must have $v_{(n+1;k)} = v_{(n;k)}$, again by definition of the order statistics, and so we have

$$v_{n+1} > v_{(n+1;k)} \implies v_{n+1} > v_{(n;k)}, \tag{37}$$

which violates that $v_{n+1} \leq v_{(n;k)}$. Therefore, we have $v_{n+1} > v_{(n+1;k)} \iff v_{n+1} > v_{(n;k)}$.

# D  ALGORITHMIC REALIZATION

This section provides a detailed description of the algorithm implementation of our proposed method. In general, we separate low-entropy and high-entropy samples to learn effective knowledge and unlearn irrelevant information, respectively. First, we predict the required sample selection ratio based on the class distribution of the dataset. Then, low-entropy samples are used to learn effective knowledge, while high-entropy samples are used to forget harmful information in order to prevent overfitting.

When estimating the sample ratio, we model the low-entropy samples using the following function: $\tau_l^n = Max(\mathcal{H}(x_i))(\alpha K + \beta)$, where $\tau_l^n$ represents the computed numerical threshold. As the first batch of data is processed, we convert this numerical threshold into a proportional threshold $\tau_l^p$ to avoid the issue of threshold decay as the optimization process progresses. For high-entropy samples, we fix the selection ratio $\tau_h^p$ at 10% to simplify the process and avoid additional parameter adjustments.

After obtaining the selection ratios for low-entropy and high-entropy samples, we fine-tune the model accordingly. For low-entropy samples, we minimize the entropy loss to reduce prediction uncertainty, enhance the consistency between image and text features, and improve class separability. For high-entropy samples, we maximize the entropy loss to unlearn harmful information, thus preventing model overfitting.

Finally, the total loss obtained from the above process is used to update the LayerNorm layer. The optimization for both the vision encoder and the text encoder is based on AdamW, with learning rates of 1e-3, 5e-4, and 5e-4 for CIFAR-10-C, CIFAR-100-C, and ImageNet-C, respectively, and batch sizes of 200, 200, and 64.

---

**Algorithm 1** Our method

---

**Input**: well-trained CLIP model with image encoder $E_v$ and text encoder $E_t$, corrupted data stream $X_{test}$ and its class number $K$, prompt template 'a photo of a <cls>'.
**Output**: Fine-tuned CLIP model.

1: **for** time step $t \leftarrow 0$ **to** $T$ **do**
2:    Input a batch of image to $E_v$.
3:    Input text description based on prompt template to $E_t$.
4:    // Estimate threshold
5:    **if** t==0 **then**
6:        Estimate selection ratio ($\tau_l^p = \frac{1}{M} \sum_{i=1}^{M} \mathbb{I}[\mathcal{H}(x_i) < \tau_l^n]$).
7:    **end if**
8:    Select low-entropy samples and high-entropy samples.
9:    // Learning Loss and Unlearning Loss
10:   Calculate learning loss according to Eq. (3)
        $(-\sum_{i=1}^{K} p(y_i|X_{\text{low}})log(p(y_i|X_{\text{low}})) - \frac{1}{K} \sum_{c=1}^{K} \bar{v}_c \cdot f_c - \sum_{i=1}^{K} \sum_{j=1}^{K} \mathbb{I}[i \neq j](1 - \text{sim}(\bar{v}_i, \bar{v}_j)))$.
11:   Calculate unlearning loss according to Eq. (2)
        $(\sum_{i=1}^{K} p(y_i|X_{\text{high}}) \log(p(y_i|X_{\text{high}})))$.
12:   Calculate total loss ($\mathcal{L}_{\text{learning}} + \lambda \mathcal{L}_{\text{unlearning}}$).
13:   Optimize LayerNorm layers of $E_v$ and $E_t$
14: **end for**

---

# E MORE DETAILS ABOUT THRESHOLD ESTIMATE MODULE

## E.1 SELECTION RATIO OF LOW-ENTROPY SAMPLES

In Section 3.3, we discussed the design of the threshold estimate module. Here, we provide a detailed explanation for the reasoning behind this design. During our preliminary exploration, we found that the optimal selection ratio for low-entropy samples varies across different datasets and corruption types. In Figure 9, we show the model's performance at various low-entropy sample selection ratios. The results indicate that the optimal selection ratio for low-entropy samples differs across datasets. In Table 8, we further investigate the optimal selection ratios corresponding to different types of damage within the same dataset. The results show that even within the same dataset, the selection ratios for different corruption types are inconsistent, posing a significant challenge for real-world applications. Without a unified selection ratio, the model may fail to adapt optimally when encountering unknown data in real-world scenarios. Therefore, we explore how to standardize the selection ratio for low-entropy samples, aiming to enhance the model's adaptability and robustness.

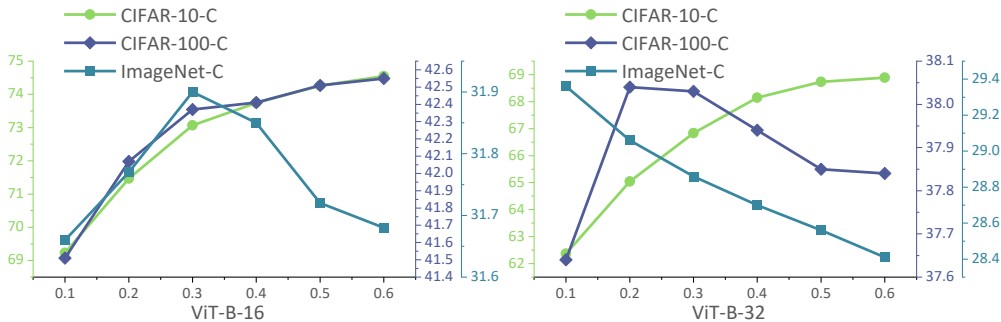

Figure 9: **The model's performance at various low-entropy sample selection ratios.** Optimal performance is achieved at different selection ratios for each dataset.

Table 8: **The optimal ratio of various corruption types**. Even within the same dataset, the optimal selection ratios for different corruption types are inconsistent.

| ViT-B-16 | Gaussian | Shot | Impulse | Defocus | Glass | Motion | Zoom | Snow | Frost | Fog | Brightness | Contrast | Elastic | Pixelate | JPEG |
|---|---|---|---|---|---|---|---|---|---|---|---|---|---|---|---|
| CIFAR-10-C | 0.6 | 0.6 | 0.7 | 0.8 | 0.6 | 0.8 | 0.8 | 0.8 | 0.8 | 0.8 | 0.8 | 0.8 | 0.7 | 0.6 | 0.6 |
| CIFAR-100-C | 0.3 | 0.3 | 0.3 | 0.6 | 0.5 | 0.6 | 0.5 | 0.6 | 0.6 | 0.5 | 0.5 | 0.5 | 0.4 | 0.6 | 0.6 |
| ImageNet-C | 0.1 | 0.1 | 0.1 | 0.1 | 0.1 | 0.3 | 0.2 | 0.2 | 0.1 | 0.2 | 0.1 | 0.3 | 0.3 | 0.4 | 0.4 |

| ViT-B-32 | Gaussian | Shot | Impulse | Defocus | Glass | Motion | Zoom | Snow | Frost | Fog | Brightness | Contrast | Elastic | Pixelate | JPEG |
|---|---|---|---|---|---|---|---|---|---|---|---|---|---|---|---|
| CIFAR-10-C | 0.5 | 0.5 | 0.6 | 0.6 | 0.4 | 0.6 | 0.6 | 0.5 | 0.6 | 0.6 | 0.6 | 0.6 | 0.5 | 0.6 | 0.6 |
| CIFAR-100-C | 0.3 | 0.2 | 0.2 | 0.2 | 0.3 | 0.2 | 0.3 | 0.1 | 0.3 | 0.2 | 0.3 | 0.3 | 0.2 | 0.3 | 0.3 |
| ImageNet-C | 0.2 | 0.2 | 0.1 | 0.1 | 0.1 | 0.1 | 0.1 | 0.1 | 0.1 | 0.3 | 0.1 | 0.1 | 0.1 | 0.1 | 0.1 |

To identify common characteristics across different selection ratios, we analyzed the entropy values of the samples at these optimal selection ratios. As shown in Figure 10, within the same dataset, the samples exhibit similar entropy values across different ratios. This leads us to hypothesize that the initial entropy value may reflect the optimal selection ratio. However, we also observed that the initial entropy values differ across datasets. By analyzing the initial data obtained during the adaptation process (i.e., the entropy values of the first batch), we noticed a certain regularity. Specifically, the ratio of entropy corresponding to the optimal selection ratio to the maximum entropy value in that batch exhibits a decreasing trend, and this relationship closely resembles a linear function. We extracted a subset of images from the training set to fit this pattern and derive the following proportional relationship:

$$\frac{\tau_l^n}{Max(\mathcal{H}(x_i))} = -0.00038K + 0.83, \tag{38}$$

where $K$ is the total category numbers. Subsequently, considering that the entropy values of the low-entropy samples decrease over the adaptation process, we use $\tau_l^p = \frac{1}{M} \sum_{i=1}^{M} \mathbb{I}[\mathcal{H}(x_i) < \tau_l^n]$ to convert the entropy threshold into a ratio. This approach helps avoid the use of a fixed numerical threshold, which could otherwise introduce a large amount of atypical features later on and exacerbate model overfitting.

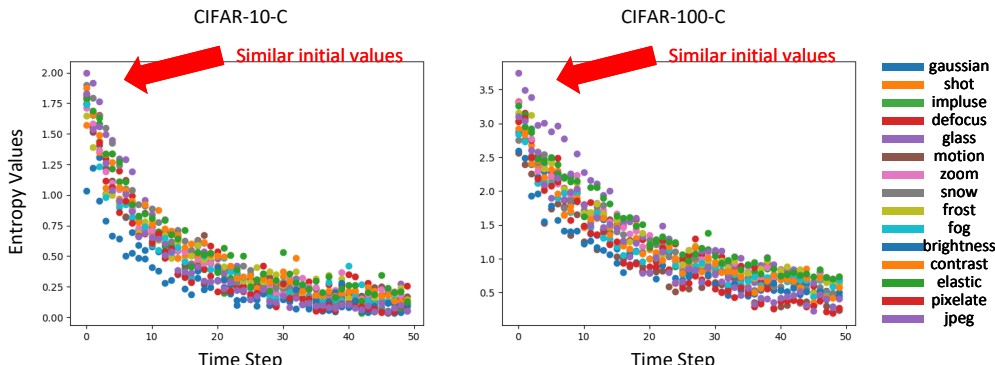

Figure 10: **The entropy values corresponding to different selection ratios.** We found that the entropy values remain very close across different ratios.

## E.2 SELECTION RATIO OF HIGH-ENTROPY SAMPLES

Similar to the selection ratio for low-entropy samples, we also explored the selection ratio for high-entropy samples. As shown in Figure 11, unlike the selection ratio for low-entropy samples, we found that selecting a fixed proportion of high-entropy samples is sufficient to help the model achieve optimal performance. Therefore, we standardize the selection ratio for high-entropy samples to 0.1.

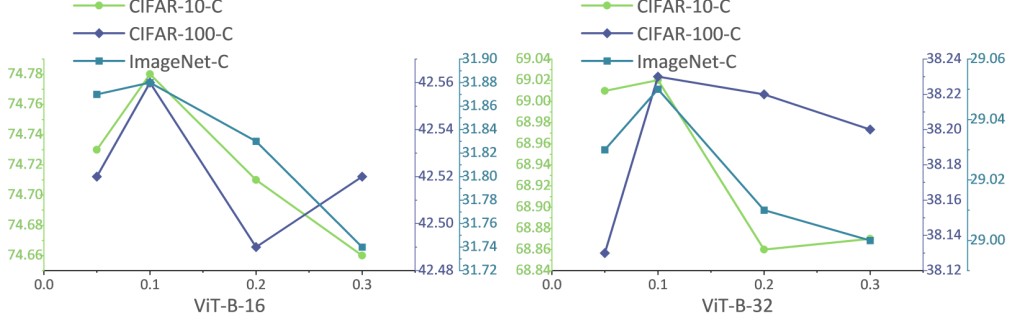

Figure 11: **The model's performance at various high-entropy sample selection ratios.** Unlike the selection ratio for low-entropy samples, optimal performance is achieved at the same selection ratios for each dataset.

## E.3 RELIABILITY OF DYNAMICALLY SELECTION MODULE

As mentioned above, we construct a dynamic selection module for low-entropy samples, which can adaptively adjust the sample selection ratio according to differences in data distributions. To verify the superiority of the fitted predictive function, we set $\alpha = 0.1$, reselected approximately 2000 images from the training set of the ImageNet-C dataset as the calibration set, and based on the ViT-B-16 backbone, calculated the error between the predicted selection ratio by the aforementioned predictive function and the groud-truth ratio in Table 8. Sort these errors, we further solved the

quantile described in Theorem 3.2 (the result is 0.028). Subsequently, we conducted a visual analysis of the matching between the prediction intervals and the true values on the test set, with the results shown in Figure 12. It can be observed that although the width of the prediction intervals is relatively narrow, almost all true values still fall within the prediction intervals. This result fully confirms the superiority of the predictive function fitted in this paper.

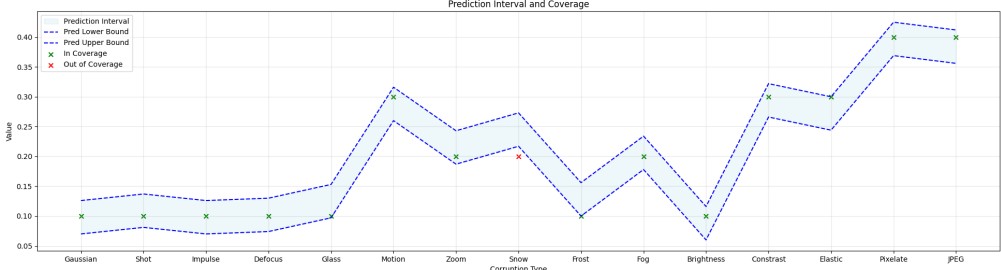

Figure 12: **Visualization of the predicted interval.** Green crosses represent ground-truth values that are correctly within the interval, whilst the red crosses represent the values that are outside of the interval. It can be seen that almost all ground-truth values are correctly within the interval, which demonstrates the superiority of the fitted prediction function.

# F  ADDITIONAL EXPERIMENT RESULTS

In this section, we provide more experiment results from different perspectives to characterize our proposed algorithms.

## F.1  TIME COSTS

We measured the average time costs taken by several different methods to test a single image on the CIFAR-10-C, CIFAR-100-C, and ImageNet-C datasets, with the results shown in Table 9. As can be seen, our method demonstrates significant competitiveness in terms of time costs.

Table 9: **Time costs of various methods in seconds.**

| ViT-B-16 | CIFAR-10-C | CIFAR-100-C | ImageNet-C |
|---|---|---|---|
| TPT | 0.0540 | 0.0612 | 0.2736 |
| DPE | 0.0498 | 0.0575 | 0.1002 |
| BAT | 0.0086 | 0.0096 | 0.0289 |
| Ours | 0.0088 | 0.0097 | 0.0292 |
| **ViT-B-32** | **CIFAR-10-C** | **CIFAR-100-C** | **ImageNet-C** |
| TPT | 0.0525 | 0.0534 | 0.2586 |
| DPE | 0.0496 | 0.0562 | 0.0966 |
| BAT | 0.0032 | 0.0040 | 0.0238 |
| Ours | 0.0034 | 0.0041 | 0.0240 |

## F.2  COMBINED WITH DIFFERENT METHODS

In the main text, we compared the performance of several methods combined with BITTA on the CIFAR-10-C dataset, including TPT (Shu et al., 2022), CTPT (Yoon et al., 2024), and TPS (Sui et al., 2024), all based on the ViT-B-16 backbone. Here, we further present the results on the ImageNet-C dataset using the same backbone. Consistent with the previous setup, we fix the optimization objective for low-entropy samples and introduce high-entropy sample selection module to implement unlearning to avoid overfitting on low-entropy samples. The results in Table 10 demonstrate that BITTA consistently improves the performance of these methods, showcasing the robustness of our proposed approach and its applicability across different datasets.

Table 10: **Comparison of BITTA combined with different TTA methods over ImageNet-C using ViT-B-16**. Δ highlighted the improvement in green over the original method without BITTA.

| Method | Gaussian | Shot | Impulse | Defocus | Glass | Motion | Zoom | Snow | Frost | Fog | Brightness | Contrast | Elastic | Pixelate | JPEG | Mean |
|---|---|---|---|---|---|---|---|---|---|---|---|---|---|---|---|---|
| TPT | 8.78 | 9.42 | 9.64 | 24.48 | 16.20 | 25.12 | 23.98 | 33.78 | **32.30** | 37.78 | 55.56 | 19.10 | 14.26 | 35.88 | **34.84** | 25.41 |
| TPT + BITTA | **12.08** | **13.90** | **12.86** | **24.88** | **16.54** | **25.62** | **24.08** | **33.84** | 32.12 | **38.50** | **55.72** | **20.16** | **14.60** | **36.06** | 34.74 | **26.38** |
| Δ | +3.30 | +4.48 | +3.22 | +0.40 | +0.34 | +0.50 | +0.10 | +0.06 | -0.18 | +0.72 | +0.16 | +1.06 | +0.34 | +0.18 | -0.10 | +0.97 |
| CTPT | 9.56 | 10.78 | 10.70 | **24.62** | 16.10 | 25.36 | 23.60 | 33.84 | 32.32 | **37.98** | **56.28** | 18.02 | 14.26 | 35.06 | 34.48 | 25.53 |
| CTPT + BITTA | **12.62** | **14.36** | **13.84** | 24.44 | **16.26** | **25.50** | **23.88** | **34.12** | **32.44** | 38.06 | 56.00 | **18.62** | **14.39** | **35.14** | **34.74** | **26.29** |
| Δ | +3.06 | +3.58 | +3.14 | -0.18 | +0.16 | +0.14 | +0.28 | +0.28 | +0.12 | +0.08 | -0.28 | +0.60 | +0.13 | +0.08 | +0.26 | +0.76 |
| TPS | 9.52 | 10.48 | 9.22 | 23.76 | 15.80 | 24.40 | 23.10 | 33.82 | 31.98 | 38.26 | **54.98** | 22.70 | 14.70 | 35.66 | 34.62 | 25.53 |
| TPS + BITTA | **11.27** | **12.30** | **11.40** | **24.38** | **16.48** | **24.94** | **23.64** | **34.38** | **32.08** | **38.36** | 54.84 | 21.08 | **15.36** | **36.80** | **34.72** | **26.14** |
| Δ | +1.75 | +1.82 | +2.18 | +0.62 | +0.68 | +0.54 | +0.54 | +0.56 | +0.10 | +0.10 | -0.14 | -1.62 | +0.66 | +1.14 | +0.10 | +0.61 |

## F.3 ABLATION STUDY ON UPDATE STEPS

We perform multiple steps of parameter adaptation on a single batch. To evaluate the impact of different update steps on overall performance, we conduct ablation study by varying the number of update steps from 1 to 4. In Figure 13, we illustrate the average accuracy across 15 different corruption types on the benchmark datasets using VIT-B-16 visual backbone. Continuous updates on a single batch can lead to over-fitting, causing the loss of prior knowledge, which in turn reduces the accuracy on subsequent data and ultimately results in performance degradation. This indicates that a single-step update allows the model to strike a good balance between retaining original prior knowledge and adapting to new data. The impact of the number of update steps on the VIT-B-32 visual backbone can be found in Figure 14. As can be seen, similar to the observations with the ViT-B-16 backbone, the model's performance gradually declines as the number of update steps increases. This phenomenon suggests that continuous updates lead to significant loss of prior knowledge of the model, which in turn affects its generalization ability and performance. On the other hand, when the number of update steps is lower, the model is better able to retain its original prior knowledge, resulting in more stable performance. Therefore, based on the experimental results, we believe that setting the number of update steps to 1 is a reasonable choice, as it helps balance the retention of prior knowledge and the improvement of model adaptability.

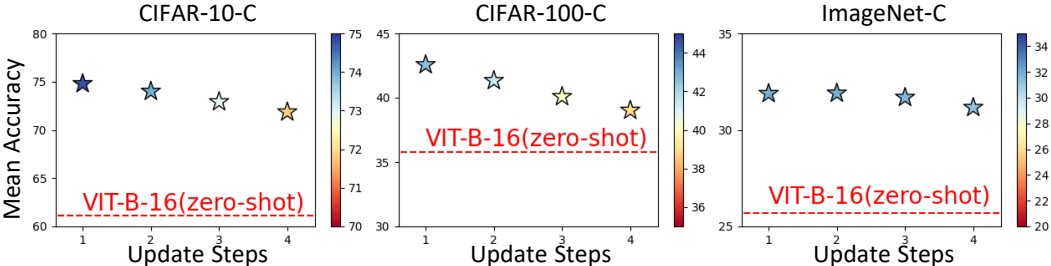

Figure 13: **Ablation study on update steps across ViT-B-16.** Continuous update lead to the performance degradation, while the mean accuracy still significantly exceed zero-shot ViT-B-16.

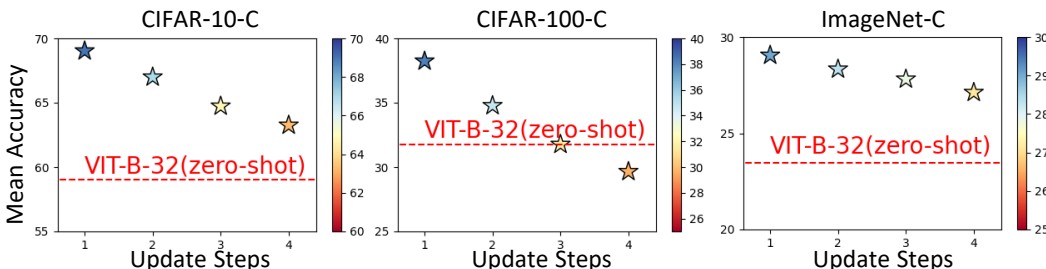

Figure 14: **Ablation study on update steps across ViT-B-32.** Similar to ViT-B-16, continuous update lead to the performance degradation on ViT-B-32. Therefore, it is reasonable to set update steps to 1.

### F.4 ABLATION STUDY ON PROMPT TEMPLATE

In Section 4, we have demonstrated the impact of different prompt templates on model performance using the ViT-B-16 and ViT-B-32 backbone. To further validate the universality and adaptability of our method, this section provides detailed results on the comparison with baseline methods. The relevant experimental results are shown in Table 11 and Table 12. As indicated by the data in the table, our method outperforms baseline methods under various prompt templates, exhibiting strong adaptability and robustness to different prompt templates. This suggests that regardless of the prompt template used, the model is able to effectively adjust and optimize its performance, showcasing its flexibility. Therefore, in practical applications, users can flexibly select and adjust the prompt templates based on specific task requirements and scenario needs to achieve optimal model performance.

Table 11: **Detailed results of ablation study on prompt templates on ViT-B-16.** Our method outperforms baseline methods across different prompt templates on ViT-B-16.

| Prompt Template | Method | CIFAR-10-C | CIFAR-100-C | ImageNet-C |
|---|---|---|---|---|
| 'a photo of a <cls>' | ViT-B-16 | 61.16 | 35.79 | 25.70 |
| | TPT | 63.84 | 36.06 | 25.41 |
| | DPE | 62.48 | 38.28 | 27.37 |
| | BAT | 73.34 | 41.15 | 31.06 |
| | Ours | **74.78** | **42.56** | **31.88** |
| | Δ | **+1.44** | **+1.41** | **+0.82** |
| 'a low contrast photo of a <cls>' | ViT-B-16 | 61.88 | 35.45 | 21.06 |
| | TPT | 64.47 | 36.56 | 24.44 |
| | DPE | 62.96 | 37.97 | 28.86 |
| | BAT | 73.59 | 40.81 | 31.30 |
| | Ours | **74.85** | **42.65** | **31.97** |
| | Δ | **+1.26** | **+1.84** | **+0.67** |
| 'a blurry photo of a <cls>' | ViT-B-16 | 62.47 | 35.08 | 26.27 |
| | TPT | 65.72 | 35.91 | 24.76 |
| | DPE | 63.52 | 38.14 | 27.91 |
| | BAT | 74.32 | 41.27 | 31.24 |
| | Ours | **75.55** | **42.34** | **31.83** |
| | Δ | **+1.23** | **+1.07** | **+0.59** |
| 'a noisy photo of a <cls>' | ViT-B-16 | 63.02 | 35.18 | 25.80 |
| | TPT | 67.21 | 36.98 | 25.64 |
| | DPE | 63.76 | 38.43 | 31.21 |
| | BAT | 74.09 | 39.61 | 30.98 |
| | Ours | **75.37** | **42.22** | **31.64** |
| | Δ | **+1.28** | **+2.61** | **+0.66** |

Table 12: **Detailed results of ablation study on prompt templates on ViT-B-32.** Similar to the results on ViT-B-16, Our method also shows great adaptability across different templates on ViT-B-32.

| Prompt Template | Method | CIFAR-10-C | CIFAR-100-C | ImageNet-C |
|---|---|---|---|---|
| 'a photo of a <cls>' | ViT-B-32 | 59.01 | 31.78 | 23.45 |
| | TPT | 63.46 | 31.55 | 25.41 |
| | DPE | 61.15 | 34.58 | 25.69 |
| | BAT | 67.49 | 36.29 | 27.58 |
| | Ours | **69.02** | **38.23** | **29.05** |
| | Δ | **+1.53** | **+1.94** | **+1.47** |
| 'a low contrast photo of a <cls>' | ViT-B-32 | 60.76 | 32.12 | 23.95 |
| | TPT | 65.03 | 31.78 | 25.96 |
| | DPE | 62.77 | 34.73 | 25.91 |
| | BAT | 65.96 | 35.99 | 27.56 |
| | Ours | **69.08** | **37.71** | **29.03** |
| | Δ | **+3.12** | **+1.72** | **+1.47** |
| 'a blurry photo of a <cls>' | ViT-B-32 | 57.89 | 31.46 | 23.58 |
| | TPT | 62.21 | 31.34 | 25.62 |
| | DPE | 60.85 | 36.08 | 27.64 |
| | BAT | 66.66 | 35.74 | 26.70 |
| | Ours | **69.11** | **37.15** | **28.46** |
| | Δ | **+2.45** | **+1.41** | **+1.76** |
| 'a noisy photo of a <cls>' | ViT-B-32 | 59.79 | 31.54 | 23.47 |
| | TPT | 64.17 | 31.42 | 25.52 |
| | DPE | 61.87 | 36.58 | 28.78 |
| | BAT | 66.10 | 35.97 | 26.84 |
| | Ours | **69.27** | **37.80** | **28.57** |
| | Δ | **+3.17** | **+1.83** | **+1.73** |

## F.5 EFFECTS OF DIFFERENT LEARNABLE MODULES

Recall that in our method, we optimize the LayerNorm parameters of vision and text encoders by introducing the learning loss and unlearning loss. In Figure 15, we illustrate the results of optimizing only the LayerNorm layers of a single modality encoder and optimizing the entire model. As can be seen, optimizing only the LayerNorm of the unimodal encoder leads to a decrease in accuracy, especially when only the text encoder is optimized. This is because the image feature distribution of corrupted data exhibits a greater discrepancy from the source domain, making the regularization of image features more significant. At the same time, optimizing the entire model causes catastrophic forgetting of the model's prior knowledge, resulting in extremely poor performance. This highlights the correctness and necessity of optimizing the LayerNorm layers of both image and text encoders.

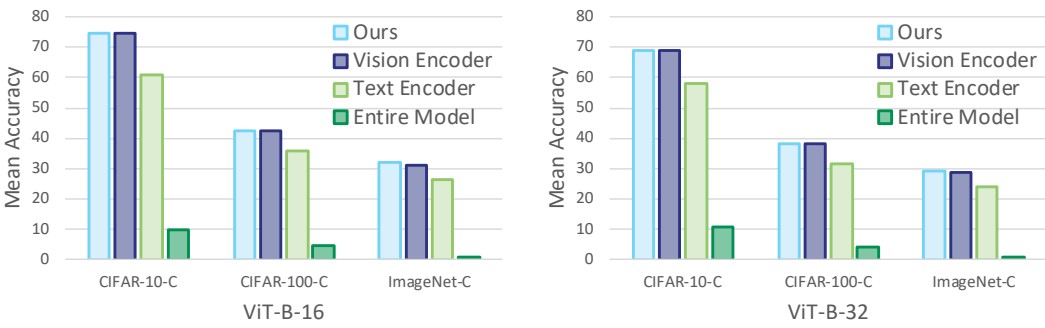

Figure 15: **Ablation study on learnable modules.** Optimizing the LayerNorm layers of both encoders shows the best performance.

## F.6 ABLATION STUDY ON $\lambda$

In the main text, we show the ablation study on $\lambda$ on ImageNet-C with ViT-B-16. In this paragraph, we show more results on the other datasets and visual backbones in Figure 16. Similar to Figure 8(b), setting $\lambda$ to 1.2 is found to be an appropriate choice and the performance gap is still smaller than 0.1% across different datasets, indicating that our methods show great robustness regarding the $\lambda$ across various datasets and visual backbones.

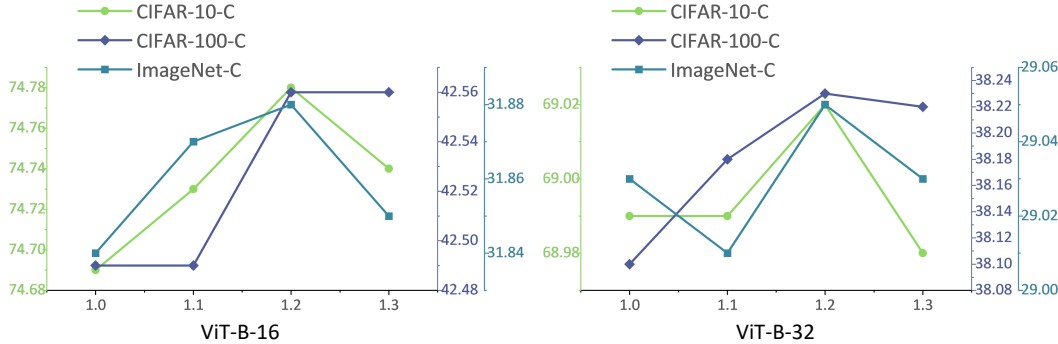

Figure 16: **Performance with varying balancing constant $\lambda$.** Setting $\lambda$ to 1.2 is found to be optimal on ViT-B-16 and ViT-B-32.

### F.7 ABLATION STUDY ON BATCH SIZE

In the main text, we conducted an in-depth investigation into the impact of batch size on performance. Here, we show more results on additional datasets and visual backbones. In Figure 17, we show the average accuracy of our method using the ViT-B-16 backbone with smaller batch sizes. It can be observed that even with a low batch size, our method still significantly outperforms the baseline method, demonstrating excellent adaptability. More results on VIT-B-32 visual backbone can be found in Figure 18. As can be seen, although there is some fluctuation in the model's adaptation performance under different batch sizes, the overall performance still significantly outperforms baseline methods. In the experiments, we observed that larger batch sizes help the model balance stability and efficiency during the learning process, while smaller batch sizes may lead to instability in certain cases.

Based on these experimental results, we can conclude that, although the choice of batch size does have some impact on the model's adaptation performance, our model demonstrates good adaptability across different batch sizes. In other words, regardless of whether using small or large batch sizes, our method consistently performs better than the traditional methods, proving its flexibility and stability in real-world applications. Therefore, batch size can be flexibly adjusted based on specific application scenarios and computational resources to optimize model performance.

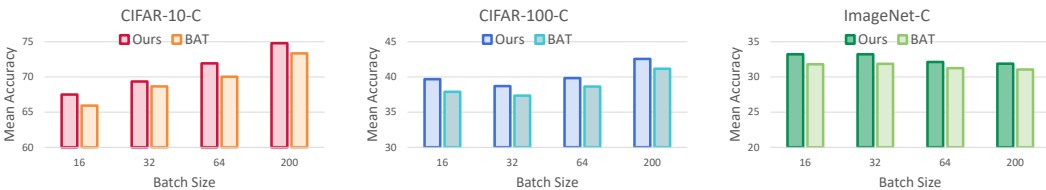

Figure 17: **Ablation study on various batch sizes across ViT-B-16.** Our method perform well under various batch sizes.

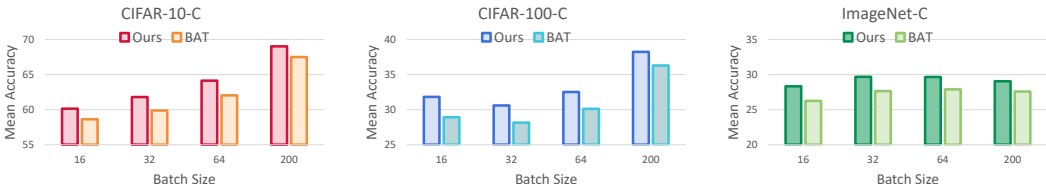

Figure 18: **Ablation study on batch sizes across ViT-B-32.**

## F.8 Generality on Diverse Architectures

In the main text, we have presented some results when combining BITTA with different architectures (such as CNN and ViT) and different methods (such as MEMO and SAR). In this section, we further present the experimental results on the ImageNet-C dataset. The data in Table 13 convincingly demonstrate that BITTA can be compatible with a variety of architectures, fully showcasing its outstanding generalization ability. At the same time, this result also further confirms the widespread existence of the overfitting problem in the research field of the Test Time Adaptation. The excellent performance of BITTA undoubtedly provides valuable and profound insights as well as practical solutions for effectively addressing this problem.

Table 13: **Comparison of BITTA combined with different TTA methods over ImageNet-C using ViT-B-16**. Δ highlighted the improvement in green over the original method without BITTA.

| ResNet101 | Gaussian | Shot | Impulse | Defocus | Glass | Motion | Zoom | Snow | Frost | Fog | Brightness | Contrast | Elastic | Pixelate | JPEG | Mean |
|---|---|---|---|---|---|---|---|---|---|---|---|---|---|---|---|---|
| MEMO | 31.84 | 33.71 | 33.22 | 30.52 | 30.09 | 44.24 | 52.02 | 49.75 | 44.20 | 58.52 | 69.12 | 33.45 | 57.68 | 60.18 | 54.76 | 45.55 |
| MEMO + BITTA | 32.63 | 34.82 | 34.07 | 31.70 | 32.11 | 45.54 | 52.95 | 50.96 | 44.97 | 59.57 | 69.37 | 37.69 | 58.85 | 60.99 | 55.94 | 46.81 |
| Δ | +0.79 | +1.11 | +0.85 | +1.18 | +2.02 | +1.30 | +0.93 | +1.21 | +0.77 | +1.05 | +0.25 | +4.24 | +1.17 | +0.81 | +1.18 | +1.26 |
| SAR | 33.45 | 33.33 | 34.52 | 32.38 | 32.58 | 45.19 | 52.34 | 50.07 | 45.25 | 59.46 | 69.21 | 39.18 | 58.63 | 60.78 | 55.35 | 46.78 |
| SAR + BITTA | 33.92 | 33.94 | 34.90 | 32.35 | 32.70 | 45.45 | 52.84 | 51.39 | 45.77 | 59.98 | 69.89 | 40.20 | 58.88 | 60.26 | 55.51 | 47.20 |
| Δ | +0.47 | +0.61 | +0.38 | -0.03 | +0.12 | +0.26 | +0.50 | +1.32 | +0.52 | +0.52 | +0.68 | +1.02 | +0.25 | -0.52 | +0.16 | +0.42 |

| ViTBase | Gaussian | Shot | Impulse | Defocus | Glass | Motion | Zoom | Snow | Frost | Fog | Brightness | Contrast | Elastic | Pixelate | JPEG | Mean |
|---|---|---|---|---|---|---|---|---|---|---|---|---|---|---|---|---|
| MEMO | 39.58 | 37.13 | 39.42 | 31.85 | 25.37 | 40.55 | 34.86 | 26.88 | 33.21 | 53.26 | 65.87 | 55.74 | 35.63 | 55.25 | 58.76 | 42.22 |
| MEMO + BITTA | 40.69 | 37.74 | 40.83 | 32.97 | 26.63 | 41.49 | 35.86 | 28.69 | 34.21 | 54.93 | 67.84 | 56.21 | 36.74 | 56.85 | 59.47 | 43.41 |
| Δ | +1.11 | +0.61 | +1.41 | +1.12 | +1.26 | +0.94 | +1.00 | +1.81 | +1.00 | +1.67 | +1.97 | +0.47 | +1.11 | +1.60 | +0.71 | +1.19 |
| SAR | 52.83 | 52.24 | 53.49 | 53.68 | 48.76 | 56.12 | 49.75 | 62.54 | 46.73 | 65.82 | 73.21 | 66.43 | 51.24 | 64.31 | 62.75 | 57.33 |
| SAR + BITTA | 56.27 | 56.44 | 57.69 | 58.26 | 55.31 | 61.45 | 56.35 | 64.39 | 62.30 | 72.56 | 77.49 | 69.17 | 62.52 | 70.33 | 65.58 | 63.07 |
| Δ | +3.44 | +4.20 | +4.20 | +4.58 | +6.55 | +5.33 | +6.60 | +1.85 | +15.57 | +6.74 | +4.28 | +2.74 | +11.28 | +6.02 | +2.83 | +5.74 |

## F.9 Results on ImageNet variants

Considering that evaluating the robustness of VLMs against corrupted images is of great significance, especially in safety-critical fields such as autonomous driving, where images are often affected by factors like blurriness and weather changes, we focus our research on the corruption benchmark in the main text. To fully verify the effectiveness of our proposed method on other datasets, we add more experiments on ImageNet variants in Table 14. The experimental results show that the BITTA model also demonstrates good compatibility on these different datasets.

Table 14: The classification accuracy of ViT-B-16 on ImageNet variants

| Method | ImageNet | ImageNet-A | ImageNet-R | ImageNet-S | ImageNet-V2 | Mean |
|---|---|---|---|---|---|---|
| TPT | 68.98 | 54.77 | 77.06 | 47.94 | 63.45 | 62.44 |
| TPT+BITTA | 70.31 | 55.78 | 78.03 | 48.93 | 64.77 | 63.56 |
| TPS | 70.10 | 60.80 | 80.28 | 49.59 | 64.83 | 65.12 |
| TPS+BITTA | 72.37 | 62.90 | 81.41 | 50.45 | 65.79 | 66.58 |

## F.10 Details about other Regularization Techniques

For weight decay, we perform normalization on each batch according to the prediction entropy and assign a weight between 0 and 1 to each image. For distribution dissolve, we replace the entropy-based unlearning loss in our original framework with a distribution dissolution loss. Specifically, we aim to ensure that the feature distribution of high-entropy images extracted by the adapted CLIP model, denoted as $E_v(X_{\text{high}})$, deviates from the original high-entropy image distribution $E'_v(X_{\text{high}})$. We re-run BITTA by updating the unlearning loss for high-entropy samples to: $L_{unlearning} = -(E_v(X_{\text{high}}) - E'_v(X_{\text{high}}))^2$.

## F.11 RESULTS ON OTHER CORRUPTION TYPES

In addition to the above analysis, to further verify the effectiveness of BITTA across different corruption types, we conduct experiments on extral corruption types on ImageNet-C and ImageNet-C-Bar with ViT-B-16. The results indicate that BITTA also achieves universal performance improvements on these corruption types.

Table 15: The classification accuracy of ViT-B-16 on other corruption types

| Dataset | ImageNet-C | | | |
|---------|------------|------------|---------|----------|
| Corruption | speckle noise | gaussian blur | spatter | saturate |
| BAT | 33.60 | 22.64 | 37.78 | 49.52 |
| BAT+BITTA | 35.74 | 25.58 | 38.54 | 50.96 |
| Dataset | ImageNet-C-Bar | | | Mean |
| Corruption | blue noise sample | brownish noise | caustic refraction | |
| BAT | 38.28 | 50.32 | 38.40 | 38.65 |
| BAT+BITTA | 39.96 | 51.22 | 38.96 | **40.14** |

## F.12 RESULTS ON NON I.I.D BATCHES

For the robustness for non i.i.d. batches, we conduct the following experiments to demonstrate the effectiveness of our method. Using 10,000 test images from CIFAR-10 with a batchsize of 200, we simulate a highly imbalanced class scenario by specifying that only classes 0-4 appear in the first 25 epochs and only classes 5-9 appear in the last 25 epochs. The test results are shown in the table below. As can be seen, our method still achieve stable performance improvement. Therefore, BITTA is robust to non i.i.d. batches.

Table 16: The classification accuracy of ViT-B-16 on non i.i.d batches.

| Corruption | Gaussian | Shot | Impulse | Defocus | Glass | Motion |
|------------|----------|------|---------|---------|-------|--------|
| BAT | 57.91 | 61.61 | 61.15 | 76.91 | 45.47 | 77.03 |
| BAT+BITTA | 60.29 | 65.82 | 63.75 | 78.80 | 47.91 | 78.53 |

# G  DISCUSSION

## G.1  LIMITATION AND FUTURE WORK

In this paper, we have clearly identified the overfitting problem caused by the low-entropy sample selection strategy that has been widely adopted in previous related works. Meanwhile, we have conducted an in-depth discussion and revealed the potential advantages and possibilities of using high-entropy samples for regularization processing. However, it is undeniable that the research findings presented in this paper still have certain limitations.

Firstly, although the current implementation of the "unlearning" operation by using the sample-level method has been able to achieve relatively satisfactory results, there is still room for further exploration. Specifically, is it possible to further reduce the fine-grained level to achieve a more accurate learning and unlearning process? This question points out a new exploration direction for future researches.

In addition, the dynamic sample selection strategy we proposed is mainly constructed based on empirical observations. In future researches, the focus can be placed on providing more precise and reliable theoretical analyses for how to select an appropriate proportion of samples, so as to further improve and optimize this strategy.

## G.2  BROADER IMPACT

As a fundamental task in the field of computer vision, the robustness of image classification tasks in the face of various kinds of corruptions is of paramount importance for the successful deployment of reliable deep learning systems in real-world scenarios. Especially in safety-critical fields such as autonomous driving, error predictions caused by noise are highly likely to lead to extremely serious consequences. In view of this, it is particularly necessary and urgent to enhance the robustness of models on out-of-distribution data.

Our research highlights a crucial yet long-overlooked issue in existing Test Time Adaptation methods: these methods typically rely solely on low-entropy samples to optimize the model. Such a one-sided optimization approach is likely to inadvertently cause the model to overfit.

To address this problem, the method we proposed ingeniously uses high-entropy samples to regularize the TTA process. Specifically, by leveraging the atypical features contained in the high-entropy part, it effectively eliminates the adverse effects of misclassified low-entropy samples. A series of comprehensive and in-depth experiments have fully verified the effectiveness and good compatibility of this method, strongly indicating that the sample selection strategy we designed is a highly promising new method that can effectively prevent the model from overfitting. This achievement undoubtedly provides a new perspective and profound insights for future research in the field of TTA, and is expected to drive further development and innovation in this field.

# H VISUALIZATION ANALYSIS

## H.1 COMPARSION OF LOW-ENTROPY SELECTION CRITERIA AND BITTA

In Figure 1, we illustrate the difference between our bilateral information-aware test-time adaptation strategy. Here, we further provide a visual comparison between low-entropy selection strategy and BITTA. It can be seen that on different datasets, our method comprehensively leads the low-entropy selection strategy, which demonstrates the effectiveness of BITTA.

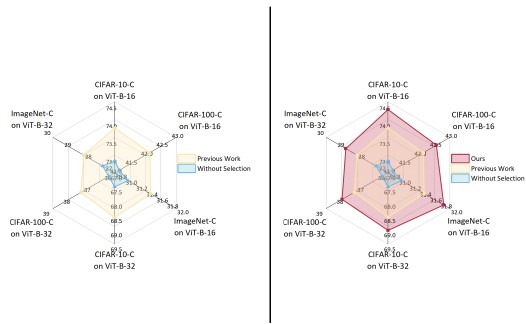

Figure 19: **Performance comparison of low-entropy selection criteria and BITTA.** It can be seen that BITTA effectively improve TTA performance across different datasets and model architectures.

## H.2 VISUALIZATION OF CORRUPTED IMAGE

In Figure 20, we present visualization examples of 15 different types of corrupted data to illustrate the impact of various corruption types on image quality. Additionally, each corruption type includes 5 different severity levels. To more clearly demonstrate the effects of different levels of corruption on the images, we further present the effects caused by each severity level in Figure 21. From these figures, it can be observed that, at the highest level of corruption, the distinguishability of the image significantly decreases, posing a great challenge for classification tasks. Therefore, the classification of corrupted images is not only an urgent issue that needs to be addressed but also a highly difficult research problem.

## H.3 MORE ATYPICAL SAMPLES VISUALIZATION

In Figure 3(a), we presented the atypical samples from low-entropy part and high-entropy part. Here, we further showcase more of these atypical samples. As can be seen from Figure 22, the model made extremely unreasonable judgments on these samples from both low-entropy part and high-entropy part, resulting in a complete mismatch between the image features and the pseudo-labels. This shows that the confidence selection strategy will cause the model to learn atypical features, and high-entropy samples becomes an ideal candidate to compensate for the atypical samples in the low-entropy part.

## H.4 T-SNE VISUALIZATION

In addition to the t-SNE visualization results presented in Section 4, we further showcase the t-SNE plots for all 15 corruption types in the CIFAR-10-C dataset in Figure 23-24. A comparison with the zero-shot CLIP method reveals that the image features generated by our method exhibit higher aggregation and clearer distinction between classes. This indicates that our approach significantly improves the zero-shot performance of CLIP when handling corrupted data.

## H.5 COMPARISON OF CLASSIFICATION RESULTS

In Figure 25-27, we visualize some examples in CIFAR-10-C, CIFAR-100-C and ImageNet-C, where we made correct predictions but not for zero-shot CLIP. This demonstrates the effectiveness of our method and shows that our method can well improve the classification accuracy of corrupted data.

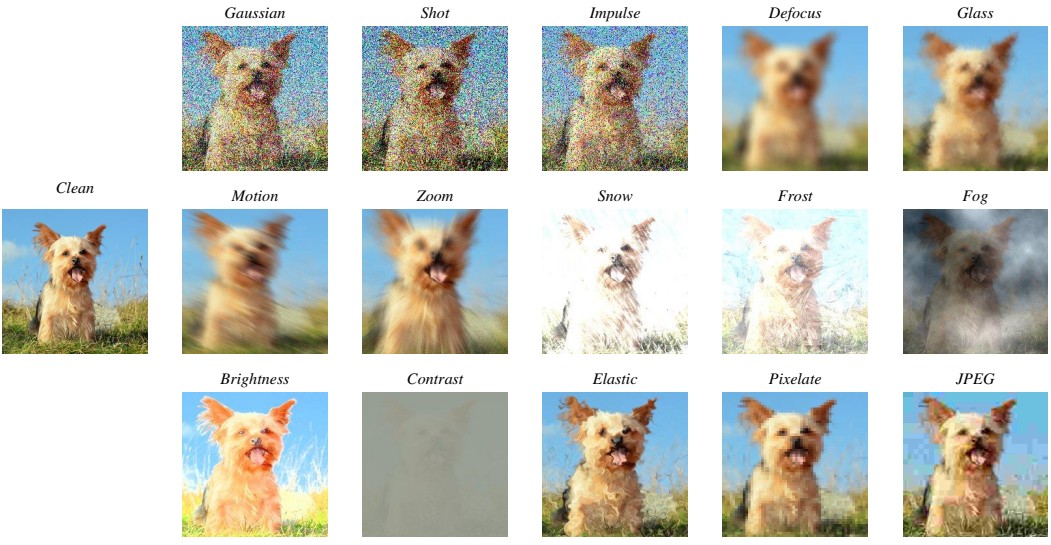

Figure 20: Visualization of a clean sample and its corrupted versions under different types of corruption.

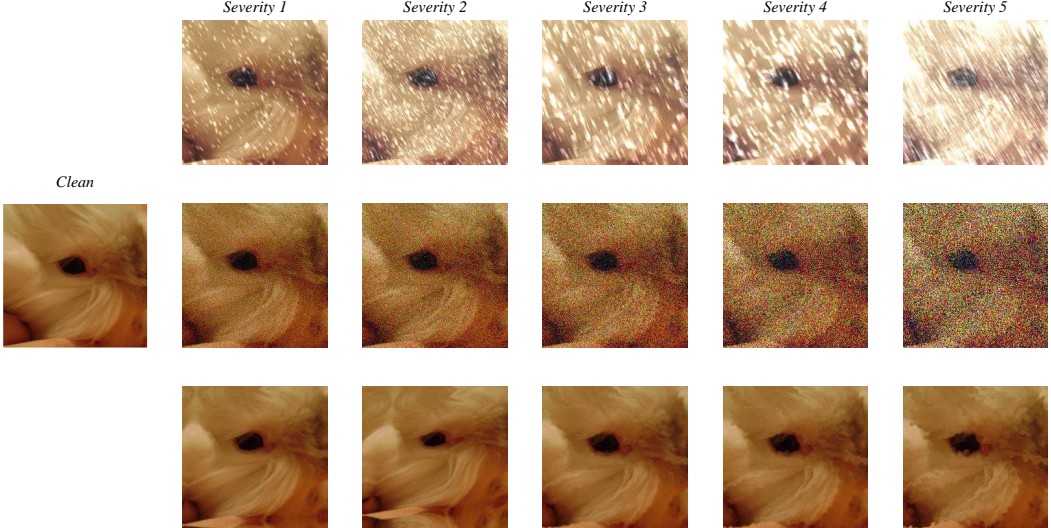

Figure 21: Visualization of various severity levels, including Snow, Gaussian Noise, and Elastic Transform.

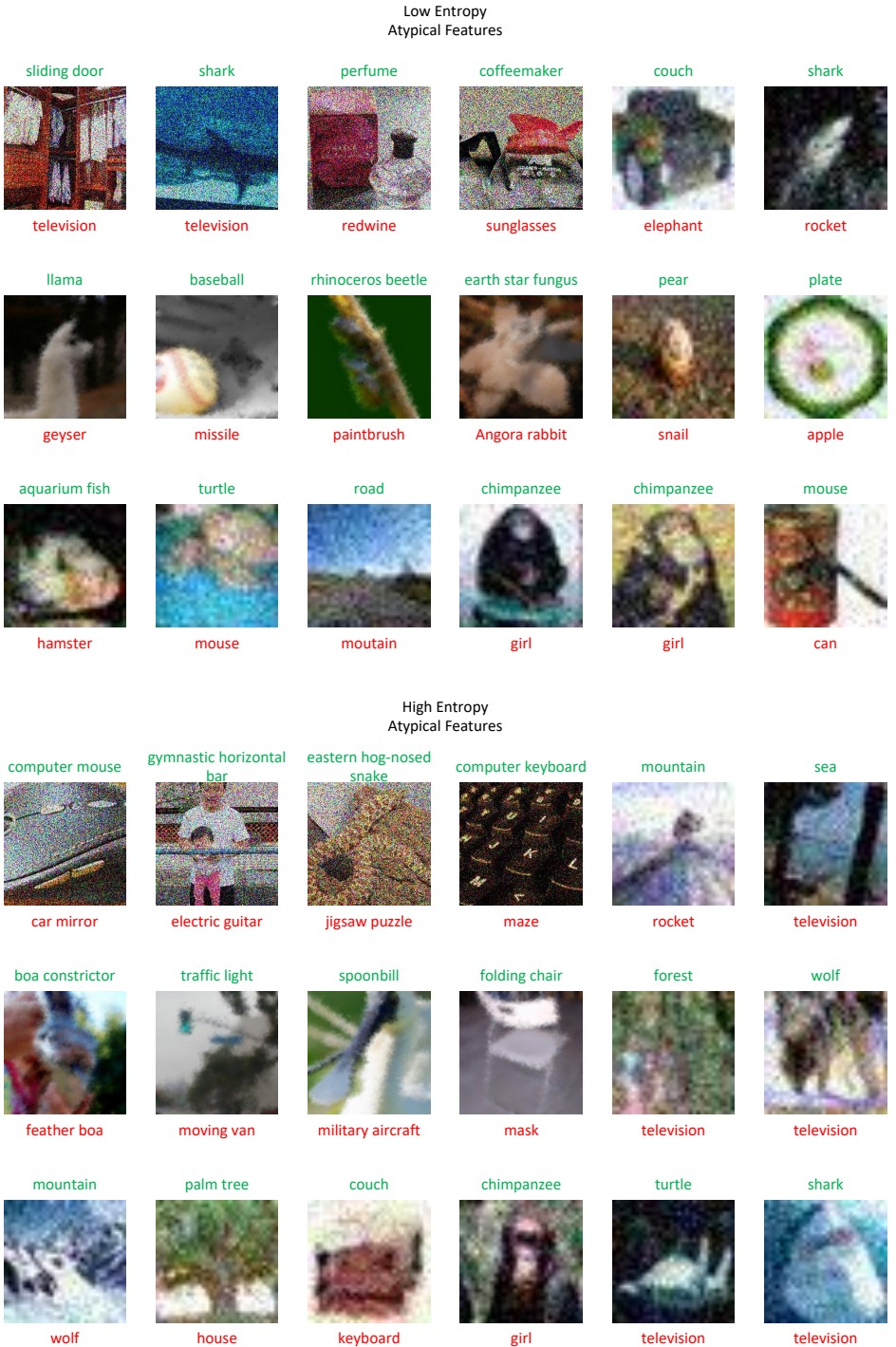

Figure 22: More Visualization of atypical samples from low-entropy part and high-entropy part.

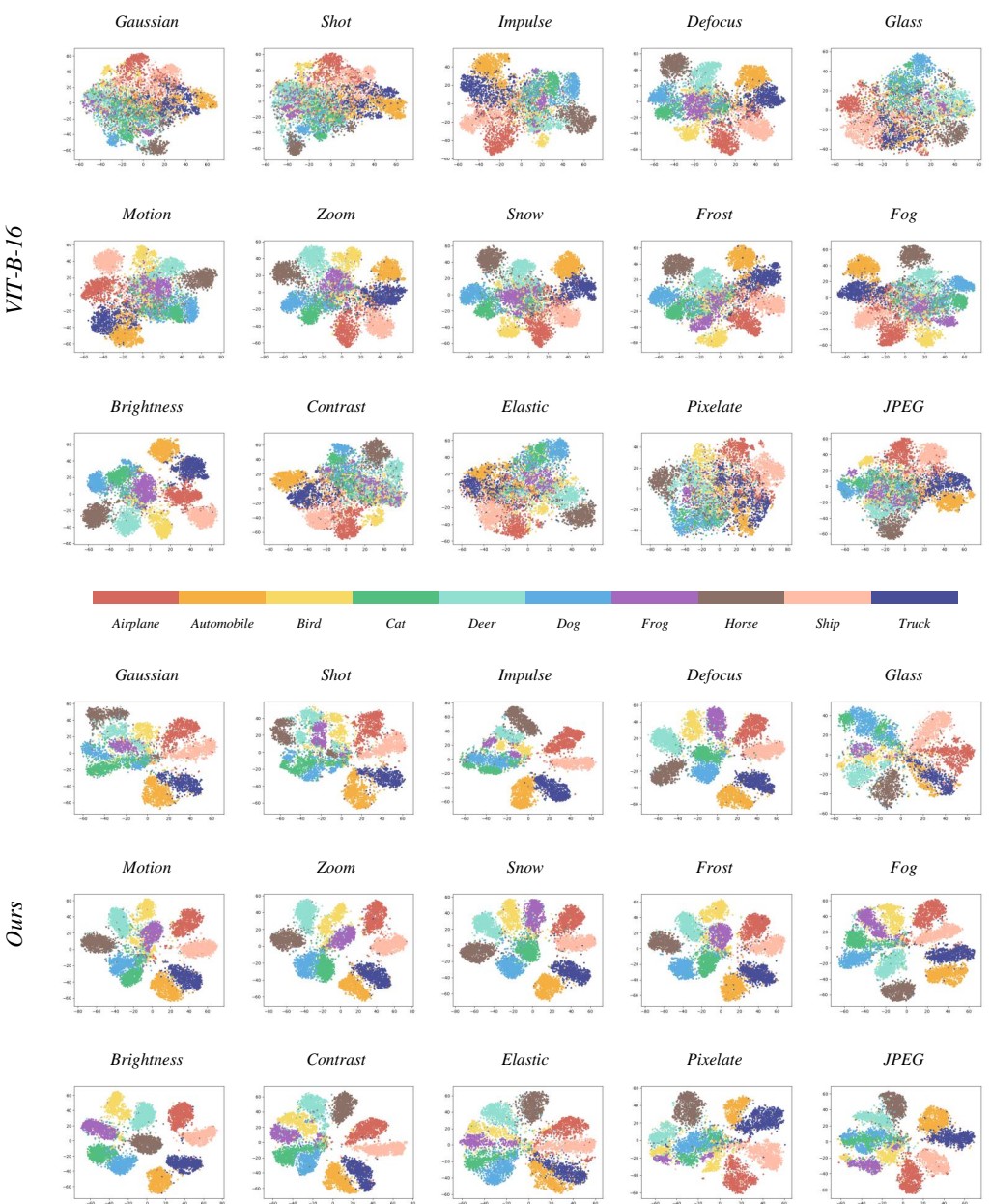

Figure 23: The t-SNE plots of our method and Zero-Shot CLIP (VIT-B-16) for CIFAR-10-C.

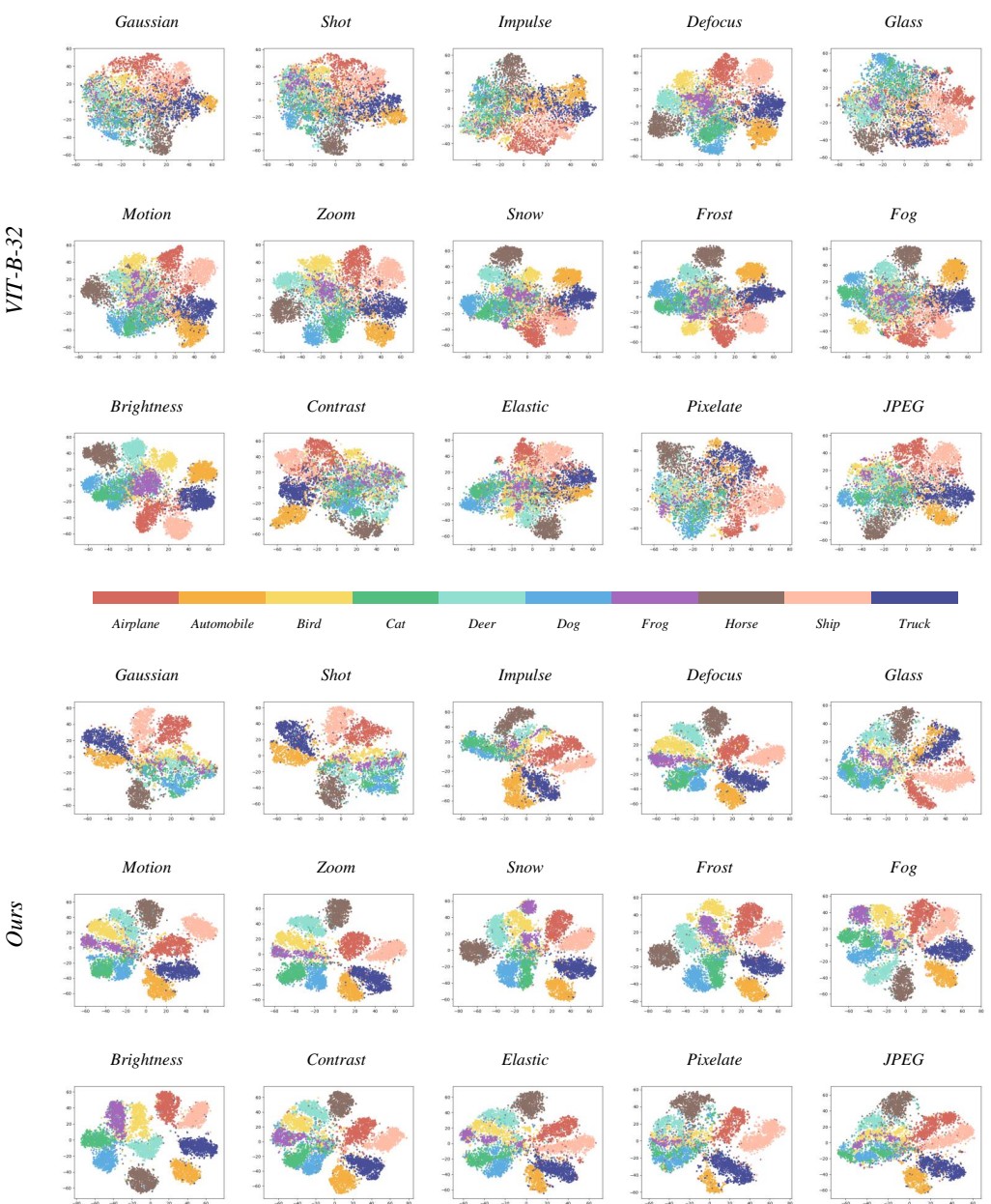

Figure 24: The t-SNE plots of our method and Zero-Shot CLIP (VIT-B-32) for CIFAR-10-C.

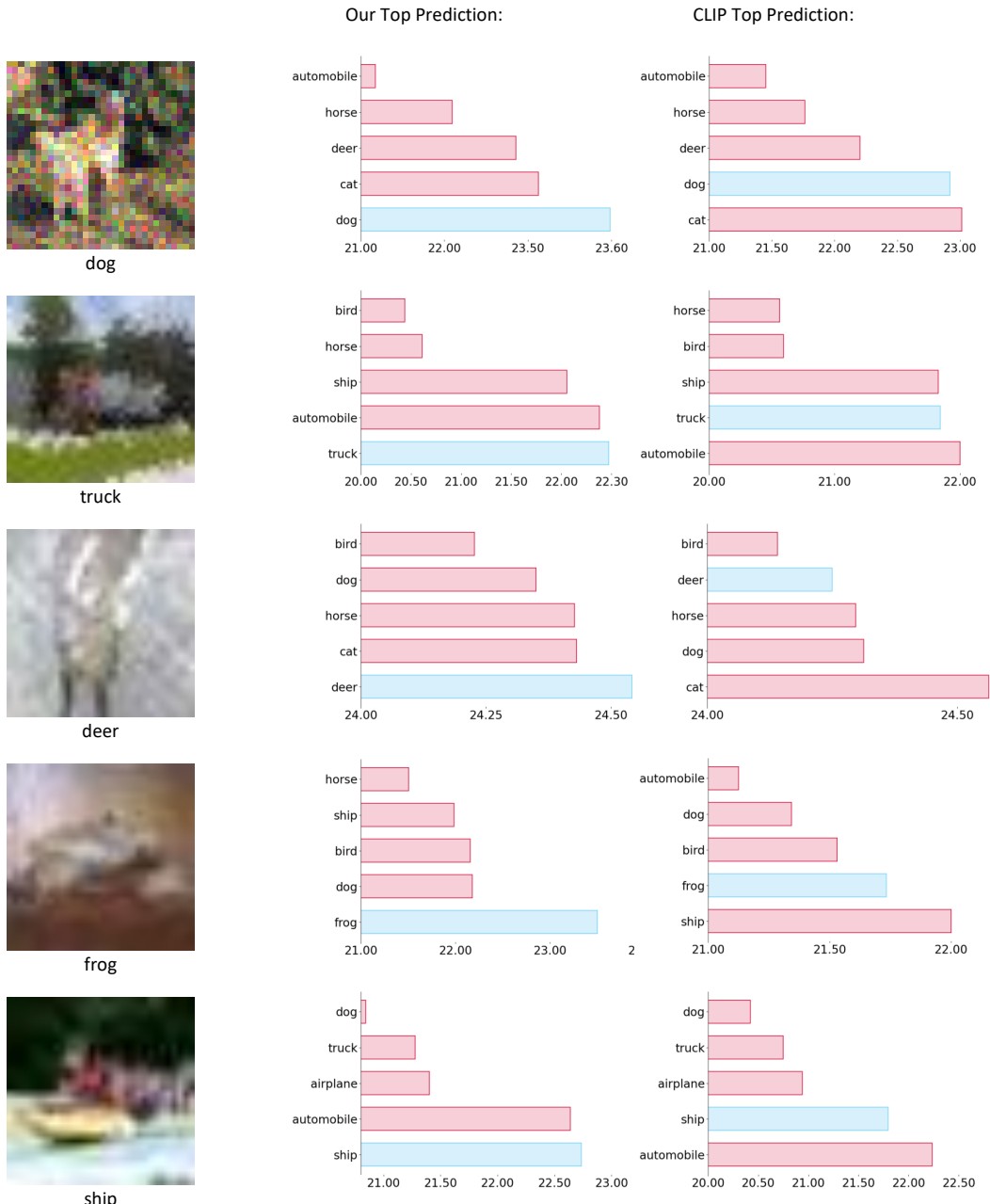

Figure 25: Comparison of classification results between zero-shot CLIP and our method on CIFAR-10-C.

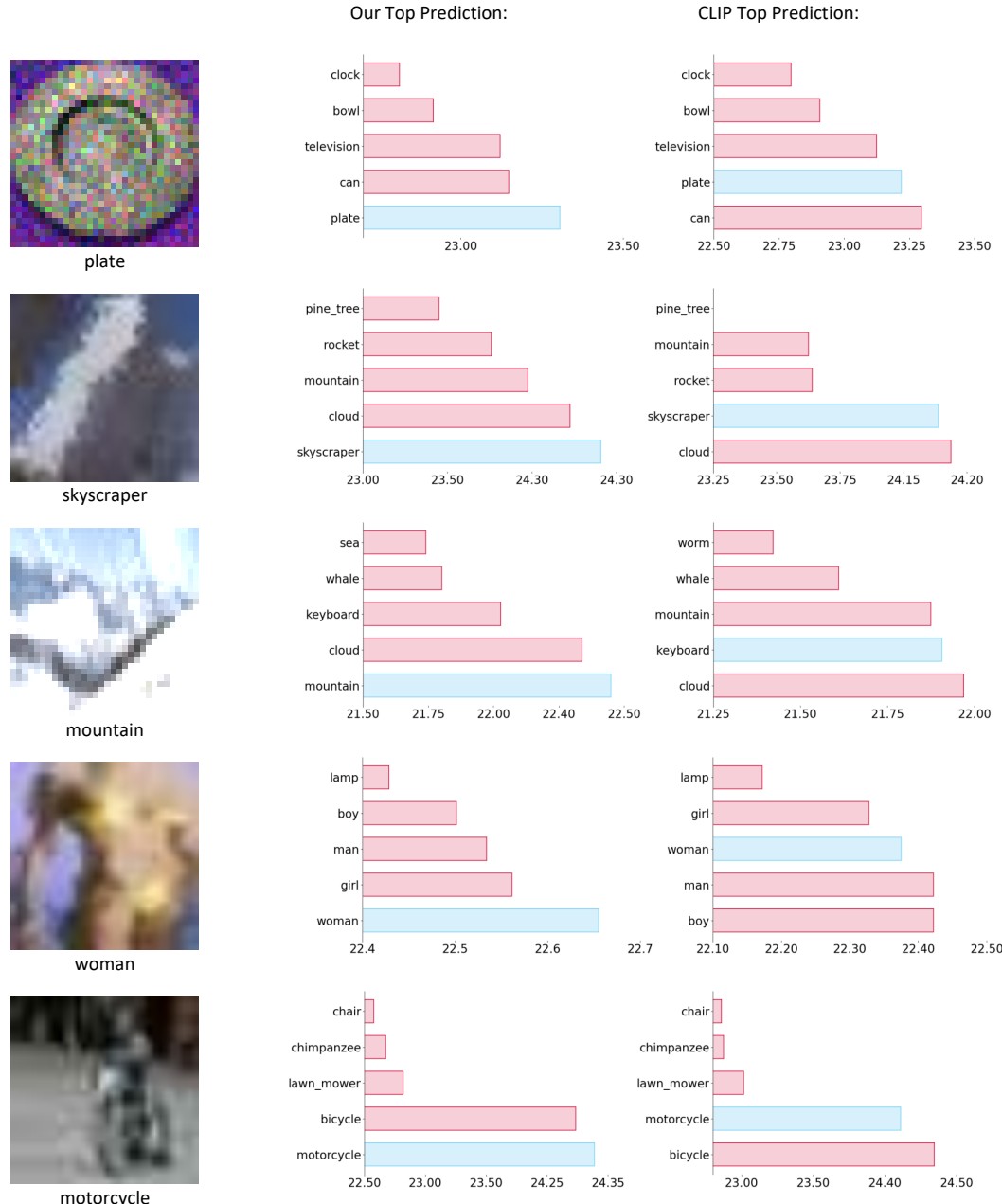

Figure 26: Comparison of classification results between zero-shot CLIP and our method on CIFAR-100-C.

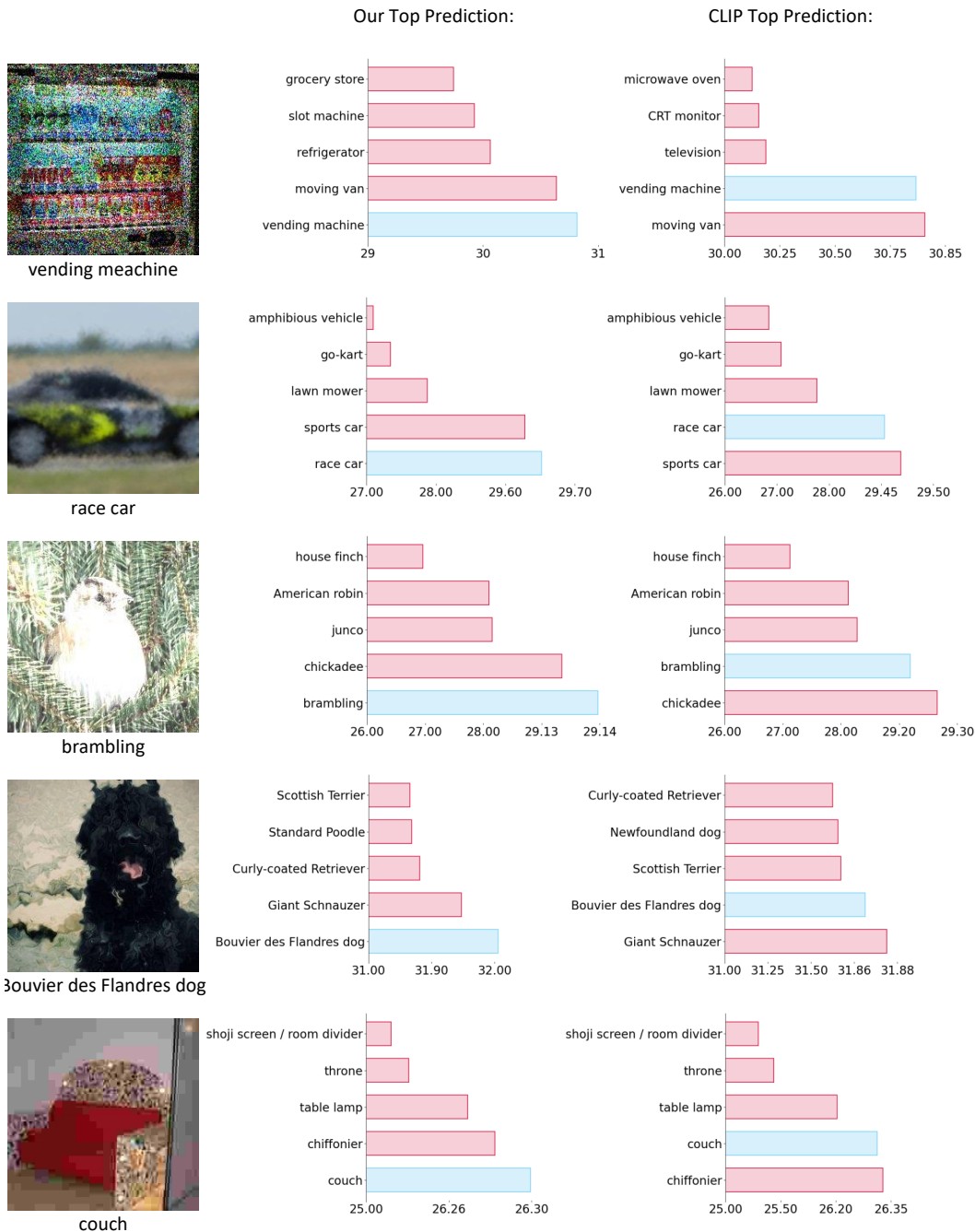

Figure 27: Comparison of classification results between zero-shot CLIP and our method on ImageNet-C.

