# OpenReview forum: "Bilateral Information-aware Test-time Adaptation for Vision-Language Models"
_ICLR.cc/2026/Conference — ICLR 2026 Poster_

### Official Review · Reviewer_y1BL · 2025-10-21

**Soundness:** 1
**Presentation:** 2
**Contribution:** 2
**Rating:** 2
**Confidence:** 4

**Summary:**

The paper tackles a known failure mode in VLM TTA: selecting only low-entropy samples for entropy minimization (EM) can overfit "atypical features" and amplify overconfidence on misclassified cases. The paper argues that the high-entropy samples which the current model is unconfident can serve as a potential candidate for regularization. Thus, BITTA proposes a bilateral strategy: (i) learn with low-entropy samples using a standard TTA objective, and (ii) unlearn with a small set of high-entropy samples by maximizing their predictive entropy. The intent is to curb memorization of atypical features while preserving core representations. BITTA is used as plug-in to multiple TTA algorithms, yielding gains on multiple benchmarks.

**Strengths:**

- Clear diagnosis of a common TTA pitfall (confidence selection → overfitting to atypical cues).

- Simple, compatible design: bilateral learning + unlearning that can wrap around existing TTA learners.

- Dynamic selection ratio that reflects dataset/noise distribution rather than a fixed value.

**Weaknesses:**

- What the method assumes. The paper assumes two things:
(1) Low-entropy and high-entropy samples both carry spurious (atypical) cues; and
(2) those cues are stronger or more frequent in the high-entropy group.
This is why the method adds an “unlearning” branch on high-entropy samples.

- What is missing. The paper does not make this assumption clear; it does not define what counts as an “atypical feature,” and never measures how much atypicality exists in each entropy group (low / medium / high).

- Why this matters. Without a clear definition and measurement regarding “atypical feature,” we cannot tell whether the gains of BITTA come from truly removing reliance on "atypical" features or from a generic regularization effect. That weakens the main motivation for the bilateral design and the authors' claims.

 - What evidence would resolve this. Provide a formal definition an operational test for “atypical features”, and report their level by entropy bin. Then show that the unlearning process for high-entropy samples reduces these atypical signals—especially for misclassified low-entropy cases.

**Questions:**

- Regarding "atypical feature" assumption, what are "atypical features"? Would it be possible to provide an operational definition of “atypical features”?

- Would it be possible to quantify their prevalence/strength of "atypical features" by entropy bins (low/medium/high) to substantiate the "atypical" feature assumption?

- Would it be possible to show that the unlearning process for high-entropy samples can reduce these atypical signals—especially for misclassified low-entropy cases?

---

> ### Author Response · Authors · 2025-11-20
> **Rebuttal by Authors**
>
> Dear Reviewer y1BL,
>
> Thank you for your valuable comments. In response to your questions, we will address them point by point, and our replies are as follows:
>
> >1. Regarding "atypical feature" assumption, what are "atypical features"? Would it be possible to provide an operational definition of “atypical features”?
>
> **The noise information contained in the incorrect prediction samples is what we claim to be atypical feature.** For the misclassified images of low-entropy part, the model **mistakenly identifies some noise information as key features of the wrong category**. In the high-entropy part, due to the influence of noise, the key features that could originally be used to distinguish categories are masked, **leaving only indistinguishable noise information.** It can be seen that both the indiscriminability of high entropy samples and the misclassification of some low-entropy samples **stem from noise information.** Therefore, we define this kind of noise information as atypical feature. As atypical features are sample-level information that is difficult to separate independently from individual samples, we cannot provide a operational definition. However, **we will supplement the explanations mentioned above in the revised version to help readers better understand.**
>
> >2. Would it be possible to quantify their prevalence/strength of "atypical features" by entropy bins (low/medium/high) to substantiate the "atypical" feature assumption?
>
> **We have detailedly verified the existence of atypical features through the change of entropy in Figure 2:** learning on low-entropy samples reduces the prediction entropy of misclassified samples, indicating that the model becomes overconfident in such samples and further extracts inaccurate features (i.e., atypical features) for image classification, ultimately leading to model overfitting, this result directly confirms the existence of atypical features.
>
> For the proposed method, **the experimental results in Figure 8(c) show that it does not reduce the prediction entropy of misclassified samples.** This means that after regularization by BITTA, the model no longer uses such noisy information for classification, effectively avoiding overfitting and fully demonstrating the effectiveness of the proposed method.
>
> In summary, **through the comparative analysis of Figure 2 and Figure 8(c),** we have clearly confirmed that existing TTA methods have the risk of utilizing atypical features (i.e., noisy information) for classification, while our method successfully suppresses this phenomenon, **providing a new optimization  direction for TTA researches.**
>
> >3. Would it be possible to show that the unlearning process for high-entropy samples can reduce these atypical signals—especially for misclassified low-entropy cases?
>
> **We have detailedly clarified why high-entropy samples are used to suppress the model's learning of atypical features in Figure 3b:** the unlearning process of high-entropy samples increases the entropy of misclassified samples in the low-entropy part, indicating that **there is a correlation between the feature representations of these two types of samples.** Thus, the atypical features learned by the model on low-entropy samples can be effectively eliminated through the unlearning regularization of high-entropy samples.

---

> ### Comment · Reviewer_y1BL · 2025-11-26
>
> Thank you for the detailed response. However, I remain unconvinced because the reply does not provide a **formal** definition of “atypical features,” nor a **link between atypicality, entropy**, and the need for bilateral learning/unlearning. To make your “atypical feature” explanation convincing (and falsifiable), I believe the revision should include the following minimal items:
>
> 1. **Definition.**
>  Provide a formal definition of “atypical features” (not just “noise in incorrect predictions”), e.g., a **specific feature component/subspace or logit contribution** that can be identified.
>
> 2. **Measurable proxy.**
>  Introduce a computable atypicality score A(x), and report its prevalence/strength by entropy bins (low/mid/high), explicitly including misclassified low-entropy samples.
>
> 3. **Causal validation of unlearning.**
>  Show that the unlearning branch reduces A(x) (feature-level atypical signal), not merely alters entropy, and demonstrate that high-entropy unlearning reduces atypicality for misclassified low-entropy cases, as **theoretical** analysis, not empirical results.
>
> Without these, the current evidence remains compatible with generic regularization/calibration effects rather than the claimed “bilateral removal of atypical features.”

---

> > ### Author Response · Authors · 2025-11-26
> > **Further Clarification**
> >
> > Dear Reviewer y1BL,
> >
> > We deeply appreciate your emphasis on the need for clearer definitions. However, we would like to clarify that **not all interpretability-related concepts can be perfectly formalized.** We believe the theoretical framework presented in our manuscript is sufficiently rigorous to enable readers to fully understand our proposed method. In fact, due to the extremely high dimensionality of deep neural networks, researchers often face inherent challenges in formulating every conceptual detail mathematically.
> >
> > For instance, WCA (ICML 2024) [1] analyzes images by distinguishing between "discriminative regions" and "non-discriminative regions" but does not provide a formal mathematical definition for these concepts. Similarly, SAR (ICLR 2023) [2] argues that samples with large gradients hinder the adaptation process yet lacks a formal metric to quantify this phenomenon. Despite the absence of formalized definitions, these works effectively facilitate readers' in-depth understanding of their proposed methods through empirical experiments and qualitative analyses.
> > In light of this, we contend that **an overemphasis on formal mathematical definitions is unnecessary for the core contribution and comprehensibility of our work.**
> >
> > [1] Visual-Text Cross Alignment: Refining the Similarity Score in Vision-Language Models.
> > [2] Towards Stable Test-Time Adaptation in Dynamic Wild World

---

> > > ### Comment · Reviewer_y1BL · 2025-11-26
> > >
> > > Thank you for the response. I would like to emphasize that my comments are intended to help improve the overall quality and clarity of the paper. While the proposed methodology may indeed work empirically, I still find it difficult to understand why it should work. In particular, I would like a clearer and more principled explanation of the relationship between **entropy** and **atypical features**. To substantively strengthen this point, I believe a more rigorous theoretical analysis would be highly beneficial.
> > >
> > > I think one possible direction is to view the proposed “typical/atypical” features through the lens of **robust/non-robust features**. If such a connection is plausible, then well-established frameworks—most notably [1] and its follow-up works—could be leveraged (even if some relaxation is needed) to (i) explicitly define and distinguish the typical vs. atypical feature subspaces at the feature level like robust vs. non-robust features, and (ii) derive a concrete analysis linking these features to entropy, in a manner analogous to prior robust/non-robust feature analyses. Probably, it may be helpful to refer to Section 2 of [1] and Section 3.1 of [2] as potential starting points for formalization.
> > >
> > > [1] Adversarial Examples Are Not Bugs, They Are Features
> > >
> > > [2] Removing Undesirable Feature Contributions Using Out-of-Distribution Data

---

### Official Review · Reviewer_uPAE · 2025-10-29

**Soundness:** 2
**Presentation:** 3
**Contribution:** 2
**Rating:** 4
**Confidence:** 4

**Summary:**

The paper proposes BITTA, a bilateral information-aware TTA framework for VLMs (e.g., CLIP): low-entropy samples are used for learning (entropy minimization) and high-entropy samples for unlearning (entropy maximization), with a heuristic to dynamically set the low-entropy selection ratio. Experiments on CIFAR-10/100-C and ImageNet-C show consistent but modest gains and some compatibility with existing TTA methods.

**Strengths:**

1. Clear identification of a failure mode of fixed low-entropy selection (overconfidence on errors).
2. Simple, general plug-in that works with multiple TTA baselines and backbones.
3. Sensible diagnostics (entropy dynamics, t-SNE) and reasonable ablations (batch size, steps, λ).
3. Low computational overhead; easy to implement.

**Weaknesses:**

1. Technical novelty is limited; the idea (minimize on confident, maximize on uncertain) is incremental, and the theory is high-level with strong assumptions.
2. ImageNet generalization is under-evaluated: key variants (e.g., ImageNet-R/A/V2/Sketch) are not systematically included in the main results.
3. Missing important baselines (e.g., DiffTPT [1], DMN-ZS [2]), weakening the empirical case.
4. Gains are often modest with occasional regressions; clearer analysis of when it helps/hurts is needed.

[1] Diverse data augmentation with diffusions for effective test-time prompt tuning
[2] Dual Memory Networks: A Versatile Adaptation Approach for Vision-Language Models

**Questions:**

See weaknesses.

---

> ### Author Response · Authors · 2025-11-20
> **Rebuttal by Authors [1/3]**
>
> Dear Reviewer uPAE,
>
> Thank you for your valuable comments. In response to your questions, we will address them point by point, and our replies are as follows:
>
> >1. Technical novelty is limited; the idea (minimize on confident, maximize on uncertain) is incremental, and the theory is high-level with strong assumptions.
>
> We believe that the technical contributions of BITTA are not limited. **Although it is simple, it is highly effective.** We believe that the value of an algorithm does not necessarily lie in excessive complexity, and there is no need to pursue overly intricate designs. In previous researches on Test-Time Adaptation (TTA), there are also many examples of **simple but effective** methods: for instance, in PromptAlign (NeurIPS 2023) [1], the authors achieved better performance merely by aligning with the mean and variance of the source domain; in DiffTPT (ICCV 2023) [2], the authors introduced an additional diffusion model to generate more augmented views, which is also simple but yields positive results; in SAR (ICLR2023) [3], the authors found that switching from fine-tuning all parameters to only fine-tuning batch normalization layers led to better performance. **Although these methods are also simple, their superiority remains undiminished.**
>
> Moreover, compared to the method itself, we consider **the new insights BITTA brings to the TTA field to be more significant.** We have confirmed that high-entropy samples, which were often overlooked in previous studies, can indeed be utilized and this new insight is a key contribution of BITTA.
>
> In addition, we believe that the theoretical part of this paper is also an important contribution—these theories can better support and help understand the proposed method. **Due to the extremely high dimensionality of deep neural networks, researchers can usually only abstract them into ideal hypothetical conditions for analysis.** For example, WCA (ICML2024) [4] assumes the network as a linear system for analysis, while BoostAdapter (NeurIPS 2024) [5] conducts theoretical derivation under the low noise condition. **Although these theories are all based on specific assumptions, they all effectively help readers gain a deeper understanding of the proposed methods.** Based on this, we believe that the theoretical analysis in this paper is valuable.
>
> [1] Align your prompts: Test-time prompting with distribution alignment for zero-shot generalization. NeurIPS2023
>
> [2] Diverse data augmentation with diffusions for effective test-time prompt tuning. ICCV2023
>
> [3] Towards Stable Test-time Adaptation in Dynamic Wild World. ICLR2023
>
> [4] Visual-Text Cross Alignment: Refining the Similarity Score in Vision-Language Models. ICML2024.
>
> [5] BoostAdapter: Improving Vision-Language Test-Time Adaptation via Regional Bootstrapping. NeurIPS2024
>
> >2. ImageNet generalization is under-evaluated: key variants (e.g., ImageNet-R/A/V2/Sketch) are not systematically included in the main results.
>
> Thank you for your advice. However, **we have reported the performance of the proposed method on ImageNet variant datasets in Appendix F.9**, which demonstrates the effectiveness of our method on cross-domain datasets. For your convenience, we show the results in the following Table:
>
> Method|ImageNet|ImageNet-A|ImageNet-R|ImageNet-S|ImageNet-V2|Mean
> |-|-|-|-|-|-|-|
> TPT|68.98|54.77|77.06|47.94|63.45|62.44
> TPT+BITTA|70.31|55.78|78.03|48.93|64.77|63.56
> TPS|70.10|60.80|80.28|49.59|64.83|65.12
> TPS+BITTA|72.37|62.90|81.41|50.45|65.79|66.58

---

> > ### Author Response · Authors · 2025-11-20
> > **Rebuttal by Authors [2/3]**
> >
> > >3. Missing important baselines (e.g., DiffTPT [1], DMN-ZS [2]), weakening the empirical case.
> >
> > Thank you for recommending two such excellent works to us. DiffTPT introduces a diffusion model for better image augmentation, and this method can be well compatible with our BITTA. To this end, we supplement compatibility experiments between BITTA and DiffTPT, and the results on CIFAR-10-C using ViT-B-16 are shown in the table below. As can be seen, the model performance has achieved universal improvement after incorporating BITTA.
> >
> > Corruption|Gaussian|Shot|Impulse|Defocus|Glass|Motion|Zoom|Snow|Frost|Fog|Brightness|Constrast|Elastic|Pixelate|JPEG|Mean
> > |-|-|-|-|-|-|-|-|-|-|-|-|-|-|-|-|-|
> > DiffTPT|39.25|45.65|59.40|71.80|43.75|71.10|75.50|75.95|78.85|73.80|84.90|72.55|57.20|51.05|62.85|64.24
> > DiffTPT+BITTA|41.91|48.50|62.40|73.00|46.35|73.05|76.75|76.85|80.10|74.90|85.95|71.90|60.15|54.85|63.75|66.03
> > GAP|2.66|+2.85|+3.00|+1.20|+2.60|+1.95|+1.25|+0.90|+1.25|+1.10|+1.05|-0.65|+2.95|+3.80|+0.90|+1.79
> >
> >
> > DMN-ZS is similar to the DPE method used in Table 1, both utilizing feature information from historical images to better assist detection tasks. We supplement comparative experiments between our method and DMN-ZS, and the results are shown below. As can be seen, our method outperforms these memory-related methods (including DPE and DMN-ZS), which fully confirms the superiority of BITTA.
> >
> > CIFAR-10-C using ViT-B-16:
> >
> > Corruption|Gaussian|Shot|Impulse|Defocus|Glass|Motion|Zoom|Snow|Frost|Fog|Brightness|Constrast|Elastic|Pixelate|JPEG|Mean
> > |-|-|-|-|-|-|-|-|-|-|-|-|-|-|-|-|-|
> > DPE|34.78|38.66|57.81|73.14|44.30|68.59|75.89|74.26|78.89|71.17|86.90|64.43|57.49|52.37|58.54|62.48
> > DMN-ZS|44.97|49.50|59.77|73.60|48.75|70.85|74.80|77.05|77.45|70.05|86.20|67.55|62.75|55.30|62.05|65.38
> > Ours|62.59|67.42|64.97|80.63|56.25|80.74|82.29|83.52|84.29|81.92|89.90|83.25|69.13|65.50|69.26|74.78
> >
> > CIFAR-100-C using ViT-B-16:
> >
> > Corruption|Gaussian|Shot|Impulse|Defocus|Glass|Motion|Zoom|Snow|Frost|Fog|Brightness|Constrast|Elastic|Pixelate|JPEG|Mean
> > |-|-|-|-|-|-|-|-|-|-|-|-|-|-|-|-|-|
> > DPE|22.74|24.18|29.95|45.48|22.18|44.83|50.67|49.97|50.84|43.57|59.55|36.61|31.53|27.89|34.22|38.28
> > DMN-ZS|19.75|21.95|29.80|42.50|18.30|43.40|48.30|49.10|51.05|43.85|56.70|35.80|31.80|26.45|31.25|36.67
> > Ours|26.71|28.67|35.19|50.71|26.06|48.95|55.24|52.56|51.44|48.66|63.67|46.63|35.32|30.83|37.79|42.56
> >
> > ImageNet-C using ViT-B-16:
> >
> > Corruption|Gaussian|Shot|Impulse|Defocus|Glass|Motion|Zoom|Snow|Frost|Fog|Brightness|Constrast|Elastic|Pixelate|JPEG|Mean
> > |-|-|-|-|-|-|-|-|-|-|-|-|-|-|-|-|-|
> > DPE|10.26|13.46|17.28|26.36|17.96|27.46|24.86|35.20|33.24|39.60|56.52|20.16|15.96|36.78|35.40|27.37
> > DMN-ZS|11.80|12.24|12.20|23.20|14.96|24.04|23.10|31.50|29.66|36.54|53.06|16.94|12.10|31.56|33.10|24.40
> > Ours|22.46|24.10|22.44|28.00|23.94|31.88|29.30|36.92|32.22|41.86|55.88|26.42|22.68|40.38|39.68|31.88
> >
> > CIFAR-10-C using ViT-B-32:
> >
> > Corruption|Gaussian|Shot|Impulse|Defocus|Glass|Motion|Zoom|Snow|Frost|Fog|Brightness|Constrast|Elastic|Pixelate|JPEG|Mean
> > |-|-|-|-|-|-|-|-|-|-|-|-|-|-|-|-|-|
> > DPE|38.83|41.34|43.76|72.49|42.00|65.56|73.05|73.99|75.19|67.46|83.45|65.56|62.58|54.25|57.80|61.15
> > DMN-ZS|45.90|48.60|43.85|70.10|48.80|68.00|72.20|72.10|73.05|67.95|81.40|67.70|63.10|53.20|59.05|62.33
> > Ours|52.34|57.61|52.28|76.82|55.99|75.79|76.83|77.64|78.68|76.79|86.50|80.41|67.44|58.45|61.69|69.02
> >
> > CIFAR-100-C using ViT-B-32:
> >
> > Corruption|Gaussian|Shot|Impulse|Defocus|Glass|Motion|Zoom|Snow|Frost|Fog|Brightness|Constrast|Elastic|Pixelate|JPEG|Mean
> > |-|-|-|-|-|-|-|-|-|-|-|-|-|-|-|-|-|
> > DPE|18.33|20.02|19.98|42.13|19.70|41.70|47.24|45.27|46.19|41.09|53.61|32.42|31.72|27.31|31.97|34.58
> > DMN-ZS|16.80|19.40|18.70|40.50|21.05|39.40|43.90|43.55|44.30|39.10|52.25|29.20|31.20|23.95|28.85|32.81
> > Ours|21.50|25.91|23.42|47.15|23.81|44.80|50.76|47.92|47.12|44.96|59.23|40.50|35.03|27.57|33.81|38.23
> >
> > ImageNet-C using ViT-B-32:
> >
> > Corruption|Gaussian|Shot|Impulse|Defocus|Glass|Motion|Zoom|Snow|Frost|Fog|Brightness|Constrast|Elastic|Pixelate|JPEG|Mean
> > |-|-|-|-|-|-|-|-|-|-|-|-|-|-|-|-|-|
> > DPE|17.28|16.54|17.48|25.10|13.52|24.78|22.44|28.06|26.64|33.32|51.70|18.66|22.38|34.06|33.42|25.69
> > DMN-ZS|14.00|14.14|13.36|26.04|12.50|23.80|23.36|29.04|26.26|32.44|49.76|17.40|19.30|33.84|29.96|24.35
> > Ours|22.48|22.34|22.20|27.22|20.22|29.26|25.80|28.82|27.46|35.94|50.00|23.96|26.96|36.78|36.30|29.05
> >
> > We have supplemented the above results into our main text in the revised version, thank you for your advice.

---

> > > ### Author Response · Authors · 2025-11-20
> > > **Rebuttal by Authors [3/3]**
> > >
> > > >4. Gains are often modest with occasional regressions; clearer analysis of when it helps/hurts is needed.
> > >
> > > Thank you for your comment. We believe that the phenomenon of limited gains or even slight performance regressions on some corruption types does not affect the stability of the overall performance improvement of our method. In fact, as shown in the following table, **almost all methods may encounter limited gains or occasional performance regressions when facing different corruption types.** Therefore, we think the stable average performance improvement brought by BITTA is significant.
> > >
> > > DPE (NeurIPS 2024) vs TPT (NeurIPS 2022) on CIFAR-10-C|TPT vs CLIP on CIFAR-100-C|BAT (ICCV 2025) vs DPE (NeurIPS 2024) on ImageNet-C
> > > |-|-|-|
> > > Gaussian,Shot,Impluse,Average|Gaussian,Shot,Motion,Average|Frost,Brightness,Contrast,Average
> > > -5.13%,-6.25%,-0.95%,-1.36%|-2.36%,-2.41%,-0.51%,-0.23%|-1.76%,-0.16%,-1.56%,1.89%
> > >
> > > In addition, to better demonstrate that the performance improvement brought by our method is stable, **we select another set of random seeds and re-run the experiments.** The experimental results on the CIFAR-10-C dataset using ViT-B-16 are shown below. As can be seen, **our method still achieves a stable performance lead.**
> > >
> > > Corruption|Gaussian|Shot|Impulse|Defocus|Glass|Motion|Zoom|Snow|Frost|Fog|Brightness|Constrast|Elastic|Pixelate|JPEG|Mean
> > > |-|-|-|-|-|-|-|-|-|-|-|-|-|-|-|-|-|
> > > BAT|61.93|65.63|63.90|80.59|53.18|80.51|82.93|83.39|83.55|82.29|88.69|82.98|69.65|60.24|69.30|73.91
> > > BAT+BITTA|64.33|68.23|65.56|81.20|56.47|81.26|83.11|83.91|84.61|82.21|89.32|84.13|71.04|65.22|70.56|75.41
> > > GAP|+2.40|+2.60|+1.66|+0.61|+3.29|+0.75|+0.18|+0.52|+1.06|-0.08|+0.63|+1.15|+1.39|+4.98|+1.26|+1.50
> > >
> > > We also conduct the following statistics on the CIFAR-10-C dataset using ViT-B-16, dividing the corruptions into four different types: Noise(i.e., gaussian noise, shot noise, impulse noise), Blur(i.e., defocus blur,  glass blur, motion blur, zoom blur), Weather(i.e., snow, frost, fog), Digital Categories(i.e., brightness, contrast, elastic_transform, pixelate, jpeg_compression)
> > >
> > > |Corruption Type|Noise|Blur|Weather|Digital Categories|
> > > |-|-|-|-|-|
> > > |Improvement|+1.87|+1.27|+0.63|+1.78|
> > >
> > > As shown in the results, the gain of BITTA on weather-related corruptions (0.63) is slightly lower than that on other types of noise corruptions, **but it still maintains a positive improvement.** We believe the reason for this phenomenon may be that weather noise usually causes distortions of global image features—for example, fog reduces image contrast and details, while snow leads to global brightness deviations. Such changes destroy the normal semantic feature distribution learned during the CLIP pre-training phase, distorting the global statistical information (such as texture, edges, and color distribution) relied on by the model, which significantly increases the difficulty of semantic alignment. Therefore, the performance improvement of BITTA on such noise data is relatively weaker. Similar phenomenon is also found in other datasets and visual backbones.
> > >
> > > **We hope the above analysis can help you better understand our method.** If you feel that the clarifications and improvements have strengthened the work, we would be grateful if you could consider updating your score. Please don’t hesitate to let us know if there are any other points we can clarify or further improve. If you have any further questions, please feel free to raise them at any time. **We will try our best to address your concerns until the end of the discussion period. Thank you again for your valuable time and comments.**

---

> > > > ### Author Response · Authors · 2025-11-26
> > > > **[Invitation to rolling discussion] Need further clarification?**
> > > >
> > > > Dear Reviewer uPAE,
> > > >
> > > > Given that the discussion deadline is approaching, we are eager to know if our responses have addressed your concerns. We look forward to further discuss with you. If you have any remaining questions, please feel free to raise them, we will try our best to address your concerns until the end of the discussion period.

---

### Official Review · Reviewer_q3aD · 2025-10-29

**Soundness:** 3
**Presentation:** 3
**Contribution:** 3
**Rating:** 4
**Confidence:** 4

**Summary:**

The article addresses the problem of test-time adaptation of a vision-language model given a stream of test data. In particular, the model selects samples with low entropy, assuming that the model's confidence reflects certainty in its predictions, and optimizes the model to further minimize entropy. Deviating from previous works, the approach selects samples of high entropy as well. For the latter, it is assumed that they have atypical features that need to be unlearned, something achieved by maximizing their entropy. The tradeoff between the two objectives aims to prevent overfitting and overconfidence while still learning from the stream of data. Experiments across domains and corruptions show the effectiveness of the approach (BITTA), outperforming recent competitors.

**Strengths:**

1. The use of samples with high entropy/low confidence is sound and very interesting. It is interesting because it has not been explored by previous approaches, focusing mostly on learning from high-confidence samples. While it is unclear what type of atypical information is included in samples with high entropy, maximizing the entropy of the latter can still act as a regularizer for avoiding overconfidence, thus improving the results.

2. The paper shows several analyses concerning multiple aspects of the approach, such as the impact of batch-size and update steps (Fig. 6, Fig. 8.a), hyperparameters (Fig. 8.b), selection module (Fig. 7), and changes in the entropy dynamic (Fig. 8.c). Moreover, experiments showcase the generality of the approach to other architectures/methods (e.g., Tab. 5 and Tab. 8). Overall, these analyses support the design choices and provide insights on the potential of the method, how the latter works, and the tradeoffs to keep in mind when applying it.

3. The appendix contains several details regarding design choices (e.g., the threshold estimate module of Appendix E), and the supplementary material includes the code. All in all, these additions make the submission transparent and provide strong support for its reproducibility.

**Weaknesses:**

1. The setting is very similar to that of Episodic TTA (e.g., [a,b,c]), where the model is updated as the stream of target data becomes available. Some of these competitors are missing in the experimental results, and including them would make the comparisons more comprehensive.

2. Related to the previous point, these methods test with a batch size of 1 (i.e., [a,c]), assuming no priors on the batch constitution. On the other hand, BITTA is very sensitive to the batch-size (e.g., results of Fig. 8.a and the adaptive threshold of 294-306). Open questions are whether the model would be i) effective for extremely low batch sizes and ii) robust to non i.i.d. batches (e.g., samples of the same class, as in [d,e]).

3. While it is intuitive that maximizing the entropy of low confident examples can both prevent overfitting and reduce overconfidence, the fact that high entropy samples share atypical features with low entropy ones (65-70) is not intuitive and not clarified with the qualitative examples of Fig. 3.a and H.3. It would be helpful to either clarify the meaning of atypical features or provide evidence of these shared spurious factors, or down weigh the related statements.

4. Lines 270-274 indicate that minimizing entropy on high-confidence samples leads to overconfidence and maximizing entropy on low-confidence ones reduces overfitting. This is also suggested by Fig. 2, Fig. 3.b, and Fig. 8.c at the level of entropy. To further analyze the phenomenon of overconfidence, it could be interesting to show the expected calibration error [f], an analysis reported by related TTA works exploring overconfidence and potential solutions (e.g., [g,h]).

5. While maximizing entropy is a good unlearning strategy, it would have been interesting to explore other alternatives (e.g., [i]) to justify this design choice. Note that different unlearning choices could lead to different effects in terms of overfitting and overconfidence. Moreover, related works do not discuss how the paper relates to the machine unlearning literature and related approaches that used unlearning for downstream tasks (e.g., debiasing [j]).


**References** ([a,b,f] already in the manuscript):

[a] Karmanov, Adilbek, et al. "Efficient test-time adaptation of vision-language models." CVPR 2024.\
[b] Zhang, Ce, et al. "Dual prototype evolving for test-time generalization of vision-language models." NeurIPS 2024.\
[c] Zhou, Lihua, et al. "Bayesian test-time adaptation for vision-language models." CVPR 2025.\
[d] Gong, Taesik, et al. "Note: Robust continual test-time adaptation against temporal correlation." NeurIPS 2022.\
[e] Niu, Shuaicheng, et al. "Towards stable test-time adaptation in dynamic wild world." ICLR 2023.\
[f] Guo, Chuan, et al. "On calibration of modern neural networks." ICML, 2017.\
[g] Yoon, Hee Suk, et al. "C-TPT: Calibrated test-time prompt tuning for vision-language models via text-feature dispersion." ICLR 2024. \
[h] Farina, Matteo, et al. "Frustratingly easy test-time adaptation of vision-language models." NeurIPS 2024.\
[i] Fan, Chongyu, et al. "Salun: Empowering machine unlearning via gradient-based weight saliency in both image classification and generation." ICLR 2024.\
[j] Chen, Ruizhe, et al. "Fast model debias with machine unlearning." NeurIPS 2023.

**Questions:**

Following from the weaknesses above:
1. How does the method compare with other TTA approaches working online?
2. Is the method robust to the batch composition?
3. Is it possible to clarify the meaning of atypical features?
4. Is BITTA reducing the overconfidence of the model/improving its calibration?
5. How does BITTA relate to approaches for machine unlearning?

---

> ### Author Response · Authors · 2025-11-20
> **Rebuttal by Authors [1/3]**
>
> Dear Reviewer q3aD,
>
> Thank you for your valuable comments. In response to your questions, we will address them point by point, and our replies are as follows:
>
> >1. The setting is very similar to that of Episodic TTA (e.g., [a,b,c]), where the model is updated as the stream of target data becomes available. Some of these competitors are missing in the experimental results, and including them would make the comparisons more comprehensive.
>
> Thank you for recommending these excellent works to us. TDA[a] utilizes dynamic cache to achieve training-free adaptation. DPE[b] evolve text and visual prototype at the same time to achieve better performance, **which we have already compared with in Table 1.** BCA[c] continuously updating class embeddings to adapt likelihood, and uses the posterior of incoming samples to continuously update the prior for each class embedding. We supplement comparisons with these methods in the table below. As can be seen, our method still outperforms these methods , which fully confirms the superiority of BITTA.
>
> CIFAR-10-C using ViT-b-16:
>
> Corruption|Gaussian|Shot|Impulse|Defocus|Glass|Motion|Zoom|Snow|Frost|Fog|Brightness|Constrast|Elastic|Pixelate|JPEG|Mean
> |-|-|-|-|-|-|-|-|-|-|-|-|-|-|-|-|-|
> DPE|34.78|38.66|57.81|73.14|44.30|68.59|75.89|74.26|78.89|71.17|86.90|64.43|57.49|52.37|58.54|62.48
> TDA|42.94|46.04|63.27|72.27|46.88|69.76|74.65|77.46|79.35|71.57|85.86|62.33|60.48|63.72|62.94|65.30
> BCA|31.51|34.53|54.17|66.50|35.64|64.93|70.59|73.90|76.27|68.45|84.02|55.42|51.52|55.80|55.95|58.61
> Ours|62.59|67.42|64.97|80.63|56.25|80.74|82.29|83.52|84.29|81.92|89.90|83.25|69.13|65.50|69.26|74.78
>
> CIFAR-100-C using ViT-b-16:
>
> Corruption|Gaussian|Shot|Impulse|Defocus|Glass|Motion|Zoom|Snow|Frost|Fog|Brightness|Constrast|Elastic|Pixelate|JPEG|Mean
> |-|-|-|-|-|-|-|-|-|-|-|-|-|-|-|-|-|
> DPE|22.74|24.18|29.95|45.48|22.18|44.83|50.67|49.97|50.84|43.57|59.55|36.61|31.53|27.89|34.22|38.28
> TDA|23.36|24.34|32.60|42.31|20.17|42.04|46.93|48.48|48.28|41.44|57.00|31.95|32.95|33.39|33.41|37.24
> BCA|12.48|13.51|25.93|39.85|19.16|40.54|45.34|47.93|46.78|40.66|55.77|27.07|31.59|31.59|31.28|33.97
> Ours|26.71|28.67|35.19|50.71|26.06|48.95|55.24|52.56|51.44|48.66|63.67|46.63|35.32|30.83|37.79|42.56
>
> ImageNet-C using ViT-b-16:
>
> Corruption|Gaussian|Shot|Impulse|Defocus|Glass|Motion|Zoom|Snow|Frost|Fog|Brightness|Constrast|Elastic|Pixelate|JPEG|Mean
> |-|-|-|-|-|-|-|-|-|-|-|-|-|-|-|-|-|
> DPE|10.26|13.46|17.28|26.36|17.96|27.46|24.86|35.20|33.24|39.60|56.52|20.16|15.96|36.78|35.40|27.37
> TDA|9.60|11.54|11.20|24.78|14.60|23.94|25.04|35.14|32.78|38.68|56.50|16.28|14.20|38.78|33.84|25.79
> BCA|6.74|7.74|7.18|26.00|14.96|24.90|25.88|33.86|32.94|37.92|56.14|17.32|14.34|39.60|35.80|25.42
> Ours|22.46|24.10|22.44|28.00|23.94|31.88|29.30|36.92|32.22|41.86|55.88|26.42|22.68|40.38|39.68|31.88
>
> The above results and those on ViT-b-32 have already be supplemented into our main text in the revised version, you can find them in the revised version. Thank you for your advice.
>
> >2. Related to the previous point, these methods test with a batch size of 1 (i.e., [a,c]), assuming no priors on the batch constitution. On the other hand, BITTA is very sensitive to the batch-size (e.g., results of Fig. 8.a and the adaptive threshold of 294-306). Open questions are whether the model would be i) effective for extremely low batch sizes and ii) robust to non i.i.d. batches (e.g., samples of the same class, as in [d,e]).
>
> Thank you for your question. **As we introduce in Section 2.** For the low-batch size case (e.g., the batch size is 1), a common operation is to augment the input image to construct an input set with a batch size of 64. In this way, BITTA can be applied to this augmented image set. We choose those high-confidence augmented views to learn feature representation and those low-confidence augmented views to unlearn atypical feature. Therefore, BITTA can be well conducted in the low-batch size case.
>
> Secondly, for the robustness for non i.i.d. batches, we supplement the following experiments to demonstrate the effectiveness of our method. Using 10,000 test images from CIFAR-10 with a batchsize of 200, we simulated a highly imbalanced class scenario by specifying that only classes 0-4 appear in the first 25 epochs and only classes 5-9 appear in the last 25 epochs. The test results are shown in the table below. As can be seen, our method still achieve stable performance improvement. Therefore, BITTA is robust to
> non i.i.d. batches.
>
> Corruption|Gaussian|Shot|Impulse|Defocus|Glass|Motion
> |-|-|-|-|-|-|-|
> BAT|57.91%|61.61%|61.15%|76.91%|45.47%|77.03%
> BAT+BITTA|60.29%|65.82%|63.75%|78.8%|47.91%|78.53%

---

> > ### Author Response · Authors · 2025-11-20
> > **Rebuttal by Authors [2/3]**
> >
> > >3. While it is intuitive that maximizing the entropy of low confident examples can both prevent overfitting and reduce overconfidence, the fact that high entropy samples share atypical features with low entropy ones (65-70) is not intuitive and not clarified with the qualitative examples of Fig. 3.a and H.3. It would be helpful to either clarify the meaning of atypical features or provide evidence of these shared spurious factors, or down weigh the related statements.
> >
> > **The noise information contained in the incorrect prediction samples is what we claim to be atypical feature.** For the misclassified images of low-entropy part, the model **mistakenly identifies some noise information as key features of the wrong category**. In the high-entropy part, due to the influence of noise, the key features that could originally be used to distinguish categories are masked, **leaving only indistinguishable noise information.** It can be seen that both the indiscriminability of high entropy samples and the misclassification of some low-entropy samples **stem from noise information.** Therefore, we can use the atypical feature from high-entropy part to eliminate the influence of the atypical feature learned from low-entropy part.
> >
> > >4. Lines 270-274 indicate that minimizing entropy on high-confidence samples leads to overconfidence and maximizing entropy on low-confidence ones reduces overfitting. This is also suggested by Fig. 2, Fig. 3.b, and Fig. 8.c at the level of entropy. To further analyze the phenomenon of overconfidence, it could be interesting to show the expected calibration error [f], an analysis reported by related TTA works exploring overconfidence and potential solutions (e.g., [g,h]).
> >
> > Thank you for your advice, Expected Calibration Error (ECE) is an important metric that quantifies the calibration of a model’s predicted probabilities. We supplement the following experiment on CIFAR-10-C using ViT-B-16 to analyse the ECE before and after applying BITTA. As can be seen, BITTA also bring stable reduction on Expected Calibration Error.
> >
> > Corruption|Gaussian|Shot|Impulse|Defocus|Glass|Motion|Zoom|Snow|Frost|Fog|Brightness|Constrast|Elastic|Pixelate|JPEG|Mean
> > |-|-|-|-|-|-|-|-|-|-|-|-|-|-|-|-|-|
> > BAT|24.04|20.11|24.17|10.99|30.22|9.86|9.43|8.60|8.15|9.42|5.11|7.96|17.59|24.35|18.71|15.25
> > BAT+BITTA|20.73|17.59|20.55|9.67|24.48|8.71|8.84|7.13|6.85|8.64|3.94|6.52|15.43|18.30|16.54|12.93
> > Drop|13.77%|12.53%|14.98%|12.01%|18.99%|11.66%|6.26%|17.09%|15.95%|8.28%|22.89%|18.09%|12.28%|24.85%|11.59%|15.21%
> >
> > The above results and those on ViT-b-32 and other datasets are already supplemented into our main text in the revised version, you can find them in our revised version, thank you for your advice.

---

> > > ### Author Response · Authors · 2025-11-20
> > > **Rebuttal by Authors [3/3]**
> > >
> > > >5. While maximizing entropy is a good unlearning strategy, it would have been interesting to explore other alternatives (e.g., [i]) to justify this design choice. Note that different unlearning choices could lead to different effects in terms of overfitting and overconfidence. Moreover, related works do not discuss how the paper relates to the machine unlearning literature and related approaches that used unlearning for downstream tasks (e.g., debiasing [j]).
> > >
> > > Thank you for your advice. Firstly, To explore other alternatives, gain insights from the weight saliency in [i]. We use weight decay as an alternative. To clearly compare the differences between our method and weight decay, we have supplemented the following experiment: On the CIFAR-10-C dataset, we first sort the prediction entropy of 200 images in each batch in descending order, then calculate the weight as $weight=1-\frac{(prediction\_entropy-min\_entropy)}{(max\_entropy - min\_entropy)}$, i.e., assign a weight between 0 and 1 to each image according to the magnitude of its prediction entropy. The experimental results of this scheme are shown below. It can be observed that although weight decay can achieve a certain effect, it is unable to prevent the model from learning atypical features and fails to realize more reasonable data selection (compared with our dynamic selection module), thus **its overall performance is weaker than that of our method.**
> > >
> > > Corruption|Gaussian|Shot|Impulse|Defocus|Glass|Motion|Zoom|Snow|Frost|Fog|Brightness|Constrast|Elastic|Pixelate|JPEG|Mean
> > > |-|-|-|-|-|-|-|-|-|-|-|-|-|-|-|-|-|
> > > BAT|60.47|65.48|63.40|80.09|52.34|80.40|82.00|83.05|83.55|81.25|89.62|82.39|67.54|60.52|68.06|**73.34**
> > > BAT+weight decay|61.14|65.97|63.94|80.23|52.28|80.49|82.26|82.99|84.06|82.09|90.09|82.86|67.71|61.57|69.15|**73.79**
> > > BAT+BITTA|62.59|67.42|64.97|80.63|56.25|80.74|82.29|83.52|84.29|81.92|89.90|83.25|69.13|65.50|69.26|**74.78**
> > >
> > > Secondly, for the unlearning literature and related approaches that used unlearning for downstream tasks, we have supplemented the related works into the revised version.
> > >
> > > **We hope the above analysis can help you better understand our method.** If you feel that the clarifications and improvements have strengthened the work, we would be grateful if you could consider updating your score. Please don’t hesitate to let us know if there are any other points we can clarify or further improve. If you have any further questions, please feel free to raise them at any time. **We will try our best to address your concerns until the end of the discussion period. Thank you again for your valuable time and comments.**

---

> > > > ### Comment · Reviewer_q3aD · 2025-11-25
> > > >
> > > > I thank the authors for the thorough response. I find most of my concerns to be addressed with the exception of alternative unlearning strategies (point 3) as weight decay is a regularization one (as also indicated in Tab. 6). As I do not deem this issue crucial, I will raise my score accordingly.

---

> > > > > ### Author Response · Authors · 2025-11-26
> > > > > **Further Clarification**
> > > > >
> > > > > Dear Reviewer q3aD,
> > > > >
> > > > > **We’re glad to hear that most of your concerns have been addressed, and your recognition of our work means a lot to us.**
> > > > >
> > > > > For the alternative unlearning strategies, we need to illustrate that it is technically hard to conduct more sophisticated unlearning method in the high-entropy part as **existing unlearning technique always need a target class to unlearn, while in our study, we need to unlearn feature instead of class.** Therefore, we choose to compare with another regularization technique (i.e., weight decay), and we have supplemented the illustration into related works. We hope this can fully address your concern.
> > > > >
> > > > > **If you have any other questions, please don’t hesitate to let us know, we will try our best to address your concerns.**
> > > > >
> > > > > Thank you again for your valuable time and comments.

---

### Official Review · Reviewer_obJ8 · 2025-11-01

**Soundness:** 3
**Presentation:** 3
**Contribution:** 2
**Rating:** 4
**Confidence:** 5

**Summary:**

This paper proposes BITTA, a method for test-time adaptation (TTA) that simultaneously performs learning on low-entropy (confident) samples and unlearning on high-entropy (uncertain) samples. Empirical evaluation is performed on several corruption and domain shift benchmarks, showing small but consistent improvements over prior TTA baselines.

**Strengths:**

S1. **Conceptually simple but reasonable intuition.** The idea of balancing learn and unlearn at test time is intuitive, and connects nicely to existing observations that confident-only updates lead to confirmation bias.

S2. **Dynamic adaptation ratio.** Instead of fixing the proportion of confident vs. uncertain samples, BITTA adjusts it adaptively based on batch entropy values. This is a lightweight heuristic that makes the algorithm more flexible.

S3. **Well-written and structured.** The methodology section is easy to follow, and the figures illustrating the bilateral update flow are clear.

**Weaknesses:**

W1. **Marginal improvement magnitude.** Most gains over strong baselines are within +0.3--1.0 percent points, often within the variance range reported in prior TTA studies. It would be great if the authors report the performance of average and variance of each experiment with multiple times.

W2. **Comparison of previous unlearning methods.** The paper states that high-entropy samples trigger *unlearning* to mitigate the overconfidence. I believe that there are a number of research in terms of unlearning works, so it is important to compare them with the proposed method that authors proposed in this paper. Sorry for not referring several unlearning methods to compare due to lack of expertise about unlearning domain.

W3. **No computational analysis.** BITTA claims to be lightweight, but no runtime or memory cost comparison is reported against other baselines.

W4. **Incremental novelty.** I agree that the authors address an important problem in the TTA domain. However, the proposed method is conceptually simple and lacks substantial novelty. If the approach had demonstrated a larger performance gap over prior TTA methods, its impact could have been justified despite the simplicity. Unfortunately, the observed improvements are rather marginal (as described in W1), which limits the overall significance of the contribution.

**Questions:**

BITTA is a clean and well-written paper that presents a modest yet reasonable enhancement to entropy-based test-time adaptation. The idea of learning from confident samples and unlearning uncertain ones is conceptually sound and practically implementable. However, the contribution remains incremental, and the performance gains are minor. I recommend **rejection** at this stage, but I believe the idea has potential. If the authors further refine their method and demonstrate a larger improvement over existing baselines, the work could be strong enough for acceptance in a future submission.

---

> ### Author Response · Authors · 2025-11-20
> **Rebuttal by Authors [1/2]**
>
> Dear Reviewer obJ8,
>
> Thank you for your valuable comments. In response to your questions, we will address them point by point, and our replies are as follows:
>
> >1. Marginal improvement magnitude. Most gains over strong baselines are within +0.3--1.0 percent points, often within the variance range reported in prior TTA studies. It would be great if the authors report the performance of average and variance of each experiment with multiple times.
>
> We sincerely appreciate your comments. To demonstrate that the performance improvement brought by our method is stable rather than within the fluctuation range of previous TTA methods, **we select another set of random seeds and re-run the experiments.** The experimental results on the CIFAR-10-C dataset using ViT-B-16 are shown below. As can be seen, **our method still achieves a stable performance lead.**
>
> Corruption|Gaussian|Shot|Impulse|Defocus|Glass|Motion|Zoom|Snow|Frost|Fog|Brightness|Constrast|Elastic|Pixelate|JPEG|Mean
> |-|-|-|-|-|-|-|-|-|-|-|-|-|-|-|-|-|
> BAT|61.93|65.63|63.90|80.59|53.18|80.51|82.93|83.39|83.55|82.29|88.69|82.98|69.65|60.24|69.30|73.91
> BAT+BITTA|64.33|68.23|65.56|81.20|56.47|81.26|83.11|83.91|84.61|82.21|89.32|84.13|71.04|65.22|70.56|75.41
> GAP|+2.40|+2.60|+1.66|+0.61|+3.29|+0.75|+0.18|+0.52|+1.06|-0.08|+0.63|+1.15|+1.39|+4.98|+1.26|+1.50
>
>
> Apart from that, as shown in the following table, compared with existing methods, the performance gain brought by our method is also more stable. In contrast to the performance fluctuations of some existing methods, our proposed BITTA achieves stable performanve improvement across almost all corruption types and datasets. Therefore, we believe that the performance improvement of BITTA is significant.
>
> DPE (NeurIPS 2024) vs TPT (NeurIPS 2022) on CIFAR-10-C|TPT vs CLIP on CIFAR-100-C|BAT (ICCV 2025) vs DPE (NeurIPS 2024) on ImageNet-C
> |-|-|-|
> Gaussian,Shot,Impluse,Average|Gaussian,Shot,Motion,Average|Frost,Brightness,Contrast,Average
> -5.13%,-6.25%,-0.95%,-1.36%|-2.36%,-2.41%,-0.51%,-0.23%|-1.76%,-0.16%,-1.56%,1.89%
>
> >2. Comparison of previous unlearning methods. The paper states that high-entropy samples trigger unlearning to mitigate the overconfidence. I believe that there are a number of research in terms of unlearning works, so it is important to compare them with the proposed method that authors proposed in this paper. Sorry for not referring several unlearning methods to compare due to lack of expertise about unlearning domain.
>
> **We fully understand your consideration regarding comparing more unlearning algorithm.** In fact, it is technically hard to conduct more sophisticated unlearning method in the high-entropy part as **existing unlearning technique always need a target class to unlearn, while in our study, we need to unlearn feature instead of class.** Although directly utilizing existing unlearning technique is difficult, we consider that the following direction may be worth trying. We consider that future work could explore unlearning at a lower fine-grained level. Since BITTA performs unlearning at the sample level, could it be extended to the pixel level? For example, distinguishing which pixels in an image belong to typical features that should be learned and which belong to atypical features that should be unlearned. **We have discussed this in Appendix G.1, and we hope BITTA can offer a new perspective for TTA research.**
>
> >3. No computational analysis. BITTA claims to be lightweight, but no runtime or memory cost comparison is reported against other baselines.
>
> Concerning the issue of computational analysis, **we have reported the additional time cost caused by introducing BITTA in Appendix F.1.** For your convenience, we show the results in the following table. As can be seen, introducing BITTA only adds an **extra 0.0001-0.0002 seconds**, which does not increase the algorithm complexity and thus will not incur additional inference costs.
>
> ViT-B-16|CIFAR-10-C|CIFAR-100-C|ImageNet-C
> |-|-|-|-|
> TPT|0.0540|0.0612|0.2736
> DPE|0.0498|0.0575|0.1002
> BAT|0.0086|0.0096|0.0289
> Ours|0.0088|0.0097|0.0292
> ViT-B-32|CIFAR-10-C|CIFAR-100-C|ImageNet-C
> TPT|0.0525|0.0534|0.2586
> DPE|0.0496|0.0562|0.0966
> BAT|0.0032|0.0040|0.0238
> Ours|0.0034|0.0041|0.0240

---

> > ### Author Response · Authors · 2025-11-20
> > **Rebuttal by Authors [2/2]**
> >
> > >4. Incremental novelty. I agree that the authors address an important problem in the TTA domain. However, the proposed method is conceptually simple and lacks substantial novelty.
> >
> > **Thank you for your recognition that we address an important problem in the TTA domain.** However, we don't agree our method lacks substantial novelty. **Although it is simple, it is highly effective.** We believe that the value of an algorithm does not necessarily lie in excessive complexity, and there is no need to pursue overly intricate designs. In previous researches on Test-Time Adaptation (TTA), there are also many examples of **simple but effective** methods: for instance, in PromptAlign (NeurIPS 2023) [1], the authors achieved better performance merely by aligning with the mean and variance of the source domain; in DiffTPT (ICCV 2023) [2], the authors introduced an additional diffusion model to generate more augmented views, which is also simple but yields positive results; in SAR (ICLR2023) [3], the authors found that switching from fine-tuning all parameters to only fine-tuning batch normalization layers led to better performance. **Although these methods are also simple, their superiority remains undiminished.**
> >
> > Moreover, compared to the method itself, we consider **the new insights BITTA brings to the TTA field to be more significant.** We have confirmed that high-entropy samples, which were often overlooked in previous studies, can indeed be utilized and this new insight is a key contribution of BITTA.
> >
> > [1] Abdul Samadh J, Gani M H, Hussein N, et al. Align your prompts: Test-time prompting with distribution alignment for zero-shot generalization[J]. Advances in Neural Information Processing Systems, 2023, 36: 80396-80413.
> >
> > [2] Feng C M, Yu K, Liu Y, et al. Diverse data augmentation with diffusions for effective test-time prompt tuning[C]//Proceedings of the IEEE/CVF International Conference on Computer Vision. 2023: 2704-2714.
> >
> > [3] Niu S, Wu J, Zhang Y, et al. Towards Stable Test-time Adaptation in Dynamic Wild World[C]//The Eleventh International Conference on Learning Representations.
> >
> > **Your recognition of our work means a lot to us.** If you feel that the clarifications and improvements have strengthened the work, we would be grateful if you could consider updating your score. Please don’t hesitate to let us know if there are any other points we can clarify or further improve. If you have any further questions, please feel free to raise them at any time. **We will try our best to address your concerns until the end of the discussion period. Thank you again for your valuable time and comments.**

---

> > > ### Author Response · Authors · 2025-11-26
> > > **[Invitation to rolling discussion] Need further clarification?**
> > >
> > > Dear Reviewer obJ8,
> > >
> > > Given that the discussion deadline is approaching, we are eager to know if our responses have addressed your concerns. We look forward to further discuss with you. If you have any remaining questions, please feel free to raise them, we will try our best to address your concerns until the end of the discussion period.

---

> > > > ### Comment · Reviewer_obJ8 · 2025-11-26
> > > > **Response by Reviewer obJ8**
> > > >
> > > > Thank you for kindly addressing my concerns. However, I believe that there are existing unlearning methods that operate without supervision[1], and applying them naively to the TTA setup would not be significantly difficult. Since I consider a comparison with such unlearning methods essential to demonstrate the efficacy of this paper, I have decided to retain my score.
> > > >
> > > > [1] Shen, Shaofei, et al. "Label-agnostic forgetting: A supervision-free unlearning in deep models." arXiv preprint arXiv:2404.00506 (2024).

---

> ### Author Response · Authors · 2025-11-27
> **Further clarification**
>
> Dear Reviewer obJ8,
>
> We sincerely appreciate for sharing us such outstanding work, where the LAF method achieves unsupervised unlearning by perturbing the original feature distribution. Inspired by this approach, we replace the entropy-based unlearning loss in our original framework with a **distribution dissolution loss.** Specifically, we aim to ensure that the feature distribution of high-entropy images extracted by the adapted CLIP model, denoted as $E_v{(X_{high})}$, deviates from the original high-entropy image distribution $E_v'{(X_{high})}$.
>
> We re-run BITTA by updating the unlearning loss for high-entropy samples to: $L_{unlearning}=-(E_v{(X_{high})}-E_v'{(X_{high})})^2$
>
> Notably, unlike LAF, which trains a separate VAE to assist CNNs in extracting image feature distributions, we directly leverage CLIP. This design choice is motivated by the fact that CLIP’s prior knowledge, acquired through training on massive image datasets, enables accurate feature distribution extraction—consistent with the practice in [1], where CLIP is also used directly for this purpose.
>
> Additionally, we supplement an alternative **weight decay-based scheme**: On the CIFAR-10-C dataset, we first sort the prediction entropy of 200 images in each batch in descending order, then calculate the weight as $weight=1-\frac{(prediction\_entropy-min\_entropy)}{(max\_entropy - min\_entropy)}$, i.e., assign a weight between 0 and 1 to each image according to the magnitude of its prediction entropy, replacing our original adaptation scheme that selected a certain proportion of low-entropy and high-entropy samples.
>
> Experimental results on CIFAR-10-C with ViT-B-16 for both schemes are presented below.
>
> Corruption|Gaussian|Shot|Impulse|Defocus|Glass|Motion|Zoom|Snow|Frost|Fog|Brightness|Constrast|Elastic|Pixelate|JPEG|Mean
> |-|-|-|-|-|-|-|-|-|-|-|-|-|-|-|-|-|
> BAT|60.47|65.48|63.40|80.09|52.34|80.40|82.00|83.05|83.55|81.25|89.62|82.39|67.54|60.52|68.06|**73.34**
> BAT+weight decay|61.14|65.97|63.94|80.23|52.28|80.49|82.26|82.99|84.06|82.09|90.09|82.86|67.71|61.57|69.15|**73.79**
> BAT+distribution dissolve|60.98|66.50|64.88|80.69|55.75|80.58|82.28|83.02|84.18|81.89|89.59|83.14|68.83|64.21|69.06|**74.37**
> BAT+BITTA|62.59|67.42|64.97|80.63|56.25|80.74|82.29|83.52|84.29|81.92|89.90|83.25|69.13|65.50|69.26|**74.78**
>
> As observed:
>
> For the weight decay approach: While it yields modest performance gains, **it fails to prevent the model from learning atypical features and lacks the capability for adaptive data selection** (in contrast to our dynamic selection module). Consequently, its overall performance is inferior to our proposed method.
>
> For the distribution dissolution approach: Although it effectively enables the model to "forget" high-entropy samples, it differs fundamentally from our entropy-based unlearning algorithm. Our method directly and precisely guides the model to confuse the classification of high-entropy images. In contrast, the distribution perturbation-based unlearning loss does not explicitly constrain the direction of perturbation. **This introduces the risk of the feature distribution incorrectly shifting from ambiguous regions (where the model fails to classify confidently) to discriminative regions (where the model can classify confidently).** As a result, while this scheme achieves comparable performance to ours on certain noise types, its average performance across all corruption types remains lower than our method.
>
> The above results have already be supplemented into our main text in the revised version, you can find them in Table 6. Thank you for your advice. We hope these supplemented results can fully address your concerns. If you feel that the clarifications and improvements have strengthened the work, we would be grateful if you could consider updating your score. **Thank you again for your valuable time and comments.**
>
>
> [1] Align Your Prompts: Test-Time Prompting with Distribution Alignment for Zero-Shot Generalization

---

### Official Review · Reviewer_c2Yw · 2025-11-09

**Soundness:** 3
**Presentation:** 3
**Contribution:** 3
**Rating:** 6
**Confidence:** 5

**Summary:**

This paper addresses the problem in test-time adaptation of vision–language models whereby updating solely on low-entropy samples leads to inadvertent overfitting and sample-selection ratios that lack cross-domain robustness. The proposed method minimizes entropy for low-entropy samples while simultaneously strengthening image–text alignment and inter-class separability, and maximizes predictive entropy for high-entropy samples to suppress memorization of atypical features, thereby striking a balance between adaptation and robustness. The authors also provide theoretical support regarding the separability of hard samples and coverage guarantees for proportion prediction, and demonstrate consistent performance gains on CIFAR-10/100-C, ImageNet-C, and multiple cross-domain datasets, as well as in combination with methods such as TPT, CTPT, and BAT.

**Strengths:**

1. Proposes a bilateral mechanism and uses dynamic proportion estimation to mitigate overfitting and sensitivity to sample selection.
2. Provides an analysis of the separability of hard samples and coverage guarantees for proportion prediction, enhancing the method’s interpretability and robustness.
3. Achieves consistent gains on CIFAR-10/100-C, ImageNet-C, and multiple cross-domain datasets.

**Weaknesses:**

The paper needs additional experiments to further demonstrate the effectiveness of the method.

**Questions:**

1. How is the linear relationship between the dynamic low-entropy proportion and the number of classes fitted? Which data points are used, what is the goodness of fit, and are scatter plots with regression lines on CIFAR-10-C, CIFAR-100-C and ImageNet-C, together with a small-scale sensitivity check, included?
2. Why is the high-entropy proportion fixed at 0.1? Is a brief sweep at 0.05, 0.10, 0.15, 0.20 on CIFAR-10-C and ImageNet-C (severity 5), reporting both Top-1 and ECE?
3. In the component ablations, how much gain is attributed to low-entropy learning only, high-entropy unlearning only, and both together? Are the independent contributions of each component in the bilateral mechanism quantified?
4. Does unlearning improve model uncertainty and risk control? Are ECE, NLL, and rejection AUROC reported based on existing outputs and compared with baselines?
5. On which corruption types are the gains primarily concentrated? Are results broken down by distortion type provided, in addition to the main table, to clarify performance differences across corruptions?

---

> ### Author Response · Authors · 2025-11-20
> **Rebuttal by Authors [1/2]**
>
> Dear Reviewer c2Yw,
>
> Thank you for your valuable comments. In response to your questions, we will address them point by point, and our replies are as follows:
>
> >1. How is the linear relationship between the dynamic low-entropy proportion and the number of classes fitted? Which data points are used, what is the goodness of fit, and are scatter plots with regression lines on CIFAR-10-C, CIFAR-100-C and ImageNet-C, together with a small-scale sensitivity check, included?
>
> Thank you for your question. As we illustrated in Section 3.3, **We extracted a subset of images from the training set to fit the coefficients (2000 images from each datasets)**, and applied the results to the subsequent testing process. In this way, we can avoid violating the core setup of TTA that test data should remain unseen.
>
> For the the goodness of fit, we have conducted a detailed analysis **in Appendix E.3 (RELIABILITY OF DYNAMICALLY SELECTION MODULE).** Excluding the images used for regression, we reselect approximately 2000 images from the training set of the ImageNet-C dataset as the calibration set, and based on the ViT-B-16 backbone calculate the error between the predicted selection ratio and the groud-truth ratio shown in Table6. Sort these errors, we further solved the quantile described in Theorem3.2, the result is 0.028 when $\alpha=0.1$, which is really small. That means, **the ground-truth selection ratio has 90% probability to fall within the difference range of the predicted value of 0.028.** This demonstrates the goodness of fit.
>
> We didn't show the scatter plots with regression lines, **instead, we visualize the predicted interval of Theorem3.2 in Figure 12**, as we believe it more vividly presents the prediction error and is more theoretically rigorous.
>
> >2. Why is the high-entropy proportion fixed at 0.1? Is a brief sweep at 0.05, 0.10, 0.15, 0.20 on CIFAR-10-C and ImageNet-C (severity 5), reporting both Top-1 and ECE?
>
> Regarding the proportion of high-entropy samples, **we have conducted ablation experiments in Section 4.3**, "Ablation on dynamic selection." **The right part of Figure 7 shows the model performance with the ratio varying from 5% to 30%.** It can be observed that the model achieves optimal performance when the ratio of high-entropy samples is 10%. We attribute this to the fact that, unlike the adaptation on low-entropy samples which require diverse high-quality samples to learn feature representations, a small number of high-entropy samples can already provide sufficient regularization signals. Therefore, we finally set the selection ratio of high-entropy samples to 10%.
>
> >3. In the component ablations, how much gain is attributed to low-entropy learning only, high-entropy unlearning only, and both together? Are the independent contributions of each component in the bilateral mechanism quantified?
>
> Thank you for your question. As elaborated in our Introduction, previous methods usually select a fixed ratio of low-entropy samples for optimization. Therefore, compared with the base model (e.g., ViT-B-16), the experimental results of various baseline methods (e.g., BAT, TPT) directly reflect the **performance gain brought by low-entropy learning**; while the results presented by the version with BITTA incorporated further demonstrate **the additional gain brought by the introduction of high-entropy unlearning**. Based on this, the experimental results in Table 1-3 have fully covered the independent contributions of each component.

---

> > ### Author Response · Authors · 2025-11-20
> > **Rebuttal by Authors [2/2]**
> >
> > >4. Does unlearning improve model uncertainty and risk control? Are ECE, NLL, and rejection AUROC reported based on existing outputs and compared with baselines?
> >
> > First, regarding the question of whether unlearning improves the model’s uncertainty and risk control, **we make elaboration in Figure 8(c).** Comparing with Figure 2 (results before applying BITTA), it can be found that the entropy of misclassified samples does not decrease—which indicates that **the introduce of unlearning prevent the model from being overconfident in misclassified samples, thereby greatly reducing the risk of overfitting.** Based on this, we believe that unlearning effectively improves the model’s uncertainty and risk control capabilities.
> >
> > Second, regarding other indicators. Considering Expected Calibration Error (ECE) is an important metric that quantifies the calibration of a model’s predicted probabilities. We supplement the following experiment on CIFAR-10-C using ViT-B-16 to analyse the ECE before and after applying BITTA. As can be seen, BITTA also bring stable reduction on Expected Calibration Error.
> >
> > Corruption|Gaussian|Shot|Impulse|Defocus|Glass|Motion|Zoom|Snow|Frost|Fog|Brightness|Constrast|Elastic|Pixelate|JPEG|Mean
> > |-|-|-|-|-|-|-|-|-|-|-|-|-|-|-|-|-|
> > BAT|24.04|20.11|24.17|10.99|30.22|9.86|9.43|8.60|8.15|9.42|5.11|7.96|17.59|24.35|18.71|15.25
> > BAT+BITTA|20.73|17.59|20.55|9.67|24.48|8.71|8.84|7.13|6.85|8.64|3.94|6.52|15.43|18.30|16.54|12.93
> > Drop|13.77%|12.53%|14.98%|12.01%|18.99%|11.66%|6.26%|17.09%|15.95%|8.28%|22.89%|18.09%|12.28%|24.85%|11.59%|15.21%
> >
> > The above results and those on ViT-b-32 and other datasets are already supplemented into our main text in the revised version, you can find them in our revised version, thank you for your advice.
> >
> > >5. On which corruption types are the gains primarily concentrated? Are results broken down by distortion type provided, in addition to the main table, to clarify performance differences across corruptions?
> >
> > To more clearly demonstrate the performance gains of BITTA across different corruption types, we conduct the following statistics on the CIFAR-10-C dataset using ViT-B-16, including four different corruption types: Noise(i.e., gaussian noise, shot noise, impulse noise), Blur(i.e., defocus blur,  glass blur, motion blur, zoom blur), Weather(i.e., snow, frost, fog), Digital Categories(i.e., brightness, contrast, elastic_transform, pixelate, jpeg_compression)
> >
> > |Corruption Type|Noise|Blur|Weather|Digital Categories|
> > |-|-|-|-|-|
> > |Improvement|+1.87|+1.27|+0.63|+1.78|
> >
> > As shown in the results, the gain of BITTA on weather-related corruptions (0.63) is slightly lower than that on other types of noise corruptions, **but it still maintains a positive improvement.** We believe the reason for this phenomenon may be that weather noise usually causes distortions of global image features—for example, fog reduces image contrast and details, while snow leads to global brightness deviations. Such changes destroy the normal semantic feature distribution learned during the CLIP pre-training phase, distorting the global statistical information (such as texture, edges, and color distribution) relied on by the model, which significantly increases the difficulty of semantic alignment. Therefore, the performance improvement of BITTA on such noise data is relatively weaker. Similar phenomenon is also found in other datasets and visual backbones.
> >
> > In addition to the above analysis, to further verify the effectiveness of BITTA across different corruption types, we supplemented experiments on extral corruption types on ImageNet-C and ImageNet-C-Bar with ViT-B-16. The results indicate that BITTA also achieves universal performance improvements on these corruption types.
> >
> > Corruption|ImageNet-C speckle noise|ImageNet-C gaussian blur|ImageNet-C spatter|ImageNet-C saturate|ImageNet-C-Bar blue noise sample|ImageNet-C-Bar brownish noise|ImageNet-C-Bar caustic refraction|
> > |-|-|-|-|-|-|-|-|
> > BAT|33.60%|22.64%|37.78%|49.52%|38.28%|50.32%|38.40%
> > BAT+BITTA|35.74%|25.58%|38.54%|50.96%|39.96%|51.22%|38.96%
> >
> > Finally, **we sincerely thank you again for your professional comments and the precious time you have devoted.** If you have any other questions or need further discussion, you are welcome to raise them **at any time**, and we will spare no effort to provide you with satisfactory answers.

---

> > > ### Author Response · Authors · 2025-11-26
> > > **[Invitation to rolling discussion] Need further clarification?**
> > >
> > > Dear Reviewer c2Yw,
> > >
> > > Given that the discussion deadline is approaching, we are eager to know if our responses have addressed your concerns. We look forward to further discuss with you. If you have any remaining questions, please feel free to raise them, we will try our best to address your concerns until the end of the discussion period.

---

### Official Review · Reviewer_kaur · 2025-11-10

**Soundness:** 3
**Presentation:** 4
**Contribution:** 3
**Rating:** 6
**Confidence:** 3

**Summary:**

The authors argue that previous methods have focused on improving the objective functions and have ignored the dataset perspective of test time adaptation. The standard paradigm revolves around picking up the most confident samples and applying TTA methods on these. This relies on an assumption that all low entropy samples are correct predictions. However, this can lead to memorization of atypical features as shown in figure 2 of their motivation where model becomes confident about its wrong predictions. Authors propose an interesting work around to this, they utilize high entropy samples and maximize the entropy on these samples. This leads to unlearning of such atypical features leading to higher performance across the board. Lastly, to improve the training process, authors propose that the optimal percentage for low entropy samples vary for each dataset.

**Strengths:**

- Paper introduce a novel data centric perspective to the test time adaptation.
- paper is well motivated.
- presentation is well done.
- Results are promising.

**Weaknesses:**

It is unclear which features are atypical. could it also be because of high confident incorrect predictions being part of the training mix? maybe some sort of attention map visualizations would be nice here as well.
How does it compare to other regularization techniques such as weight decay.

**Questions:**

- Comparison with other regularization techniques.
- visualization of attention maps over low confident misclassifications before TTA and high confident misclassifications after TTA.
- You could have high entropy correctly classified samples. Does something like this exist and if yes, then does it impact your method negatively?

---

> ### Author Response · Authors · 2025-11-20
> **Rebuttal by Authors [1/2]**
>
> Dear Reviewer kaur,
>
> Thank you for your valuable comments. In response to your questions, we will address them point by point, and our replies are as follows:
>
> >Q1. It is unclear which features are atypical. could it also be because of high confident incorrect predictions being part of the training mix?
>
> We sincerely appreciate your comments. We agree your image that **the noise information contained in the high confident incorrect predictions in the training mix is what we claim to be atypical feature.** For the misclassified images of low-entropy part, the model **mistakenly identifies some noise information as key features of the wrong category**. In the high-entropy part, due to the influence of noise, the key features that could originally be used to distinguish categories are masked, **leaving only indistinguishable noise information.** It can be seen that both the indiscriminability of high entropy samples and the misclassification of some low-entropy samples **stem from noise information.** Therefore, we can use the atypical feature from high-entropy part to eliminate the influence of the atypical feature learned from low-entropy part.
>
> >Q2. Comparison with other regularization techniques such as weight decay.
>
> Thank you for your question. Weight decay is indeed a direction worth exploring. To clearly compare the differences between our method and weight decay, we have supplemented the following experiment: On the CIFAR-10-C dataset, we first sort the prediction entropy of 200 images in each batch in descending order, then calculate the weight as $weight=1-\frac{(prediction\_entropy-min\_entropy)}{(max\_entropy - min\_entropy)}$, i.e., assign a weight between 0 and 1 to each image according to the magnitude of its prediction entropy. The experimental results of this scheme are shown below. It can be observed that although weight decay can achieve a certain effect, it is unable to prevent the model from learning atypical features and fails to realize more reasonable data selection (compared with our dynamic selection module), thus **its overall performance is weaker than that of our method.**
>
> Corruption|Gaussian|Shot|Impulse|Defocus|Glass|Motion|Zoom|Snow|Frost|Fog|Brightness|Constrast|Elastic|Pixelate|JPEG|Mean
> |-|-|-|-|-|-|-|-|-|-|-|-|-|-|-|-|-|
> BAT|60.47|65.48|63.40|80.09|52.34|80.40|82.00|83.05|83.55|81.25|89.62|82.39|67.54|60.52|68.06|**73.34**
> BAT+weight decay|61.14|65.97|63.94|80.23|52.28|80.49|82.26|82.99|84.06|82.09|90.09|82.86|67.71|61.57|69.15|**73.79**
> BAT+BITTA|62.59|67.42|64.97|80.63|56.25|80.74|82.29|83.52|84.29|81.92|89.90|83.25|69.13|65.50|69.26|**74.78**
>
> >Q3. visualization of attention maps over low confident misclassifications before TTA and high confident misclassifications after TTA
>
> Thank you for your valuable comment. We have indeed attempted to visualize the attention maps of the visual encoder, but found that they do not yield clear or insightful observations. We think that the core strength of the CLIP model lies in its cross-modal alignment capability, rather than its focus on unimodal (visual) local features. Therefore, examining the attention maps of the visual encoder in isolation, without considering its feature correlations with the text encoder, fails to effectively reflect CLIP’s inherent logic of feature capture and integration. For this reason, we have chosen not to forcibly include the visual attention maps in the main text, as we are concerned that this might introduce ambiguity and incur unnecessary questions and explanation burdens.
>
> Nevertheless, we fully appreciate your emphasis on visualization analysis before and after model training. To better demonstrate the model’s behavior, **we have already detailedly presented comparative analyses of multiple groups of images before and after model classification in Section H.5 of the supplementary materials.** We hope these results will be helpful to you.

---

> > ### Author Response · Authors · 2025-11-20
> > **Rebuttal by Authors [2/2]**
> >
> > >Q4. You could have high entropy correctly classified samples. Does something like this exist and if yes, then does it impact your method negatively?
> >
> > To verify that those high entropy correctly classified samples do not have an adverse impact on our method, we conducted a statistical analysis on the part of the samples in the CIFAR-10-C dataset with an entropy value greater than 2.0. Specifically, we counted the samples that were correctly classified without BITTA and still remained correctly classified with BITTA. Detailed results are as follows:
> >
> > Method|Gaussian|Shot|Impulse|Defocus|Glass|Motion|Zoom|Snow|Frost|Fog|Brightness|Constrast|Elastic|Pixelate|JPEG|Mean
> > |-|-|-|-|-|-|-|-|-|-|-|-|-|-|-|-|-|
> > BAT|77|82|79|61|83|58|61|59|60|61|57|58|76|76|75|68.2
> > BAT+BITTA|76|80|79|61|82|58|61|58|60|60|57|58|75|76|75|67.7
> >
> > It can be clearly seen from the statistical results that after applying BITTA, the vast majority of such samples still **maintain correct classification.** This fully demonstrates that the "unlearning" operation has not impaired the model's classification performance for these samples containing rare features. Therefore, **those high entropy correctly classified samples do not have an adverse impact on our method.**
> >
> > Finally, **we sincerely thank you again for your professional comments and the precious time you have devoted.** If you have any other questions or need further discussion, you are welcome to raise them **at any time**, and we will spare no effort to provide you with satisfactory answers.

---

> > > ### Author Response · Authors · 2025-11-26
> > > **[Invitation to rolling discussion] Need further clarification?**
> > >
> > > Dear Reviewer kaur,
> > >
> > > Given that the discussion deadline is approaching, we are eager to know if our responses have addressed your concerns. We look forward to further discuss with you. If you have any remaining questions, please feel free to raise them, we will try our best to address your concerns until the end of the discussion period.

---

### Meta-Review · Area_Chair_zEqz · 2026-01-02

**Summary:**

The reviewers' concerns centered primarily on the technical novelty, the empirical significance of the results, and the clarity of the underlying mechanisms. Several reviewers initially felt that the "learn from confident/unlearn from uncertain" paradigm was a relatively incremental addition to the Test-Time Adaptation (TTA) literature. There was a specific worry that the observed performance gains might fall within the margin of error or represent generic regularization (like weight decay) rather than a breakthrough in handling "atypical features." This led to requests for more exhaustive benchmarking against recent VLM-specific TTA methods like DiffTPT and BCA, as well as more rigorous analysis of model calibration and uncertainty using metrics.

Beyond the empirical results, a major conceptual hurdle was the definition and verification of "atypical features." Reviewers questioned whether the high-entropy samples truly shared the same noise patterns as misclassified low-entropy samples. While most reviewers were satisfied by the authors' qualitative explanations and entropy-dynamic visualizations, one reviewer maintained a strong objection, demanding a formal mathematical subspace definition of these features. Furthermore, concerns were raised regarding the method's sensitivity to batch composition and size, particularly in real-world scenarios where data might be non-i.i.d. or arrive in extremely small batches. The authors addressed these by demonstrating robustness to imbalanced class streams and providing additional statistics on computational overhead.

**Reviewer Concerns:**

The rebuttal successfully addressed the vast majority of empirical and comparative concerns raised by the reviewers. By providing extensive new results on CIFAR-100-C, ImageNet-C, and several out-of-distribution variants (ImageNet-A/R/V2), the authors effectively countered the critique that their evaluation was limited. They also satisfied the most critical request for baseline comparisons by testing against recent state-of-the-art methods such as BCA, TDA, and DMN-ZS, demonstrating that BITTA consistently outperforms or complements these existing frameworks. Furthermore, the inclusion of ECE statistics was a meaningful addition; it provided quantitative proof that the "unlearning" mechanism actually improves model reliability, moving the discussion from purely heuristic intuition to measurable performance gains.

However, a few conceptual and theoretical concerns remain technically outstanding. While the authors provided empirical evidence of "atypical features" through entropy dynamics, they did not provide the formal mathematical formalization (e.g., specific feature subspaces or robust vs. non-robust feature definitions) requested by Reviewer y1BL.

**Reviewer Scores:**

I considered Reviewer y1BL's outlying review less seriously, because this paper has five reviews even though I exclude his/her review.

Reviewer q3aD actually raised his/her rating.

Except Reviewer y1BL, at least two other reviewers would raise his/her rating because I think that the rebuttals are very thorough.

Therefore, except Reviewer y1BL, the overall rating would approach 6, which is reasonable to recommend accept (poster).

---

### Decision · Program_Chairs · 2026-01-26

Accept (Poster)